# The *ALF* (*A*lgorithms for *L*attice *F*ermions) project release 2.1
## Documentation for the auxiliary-field quantum Monte Carlo code

The **ALF Collaboration**[*]: F. F. Assaad[1,4], M. Bercx[1], F. Goth[1], A. Götz[1],
J. S. Hofmann[2], E. Huffman[3], Z. Liu[1], F. Parisen Toldin[1], J. S. E. Portela[1], J. Schwab[1]

**1** Institut für Theoretische Physik und Astrophysik, Universität Würzburg,
97074 Würzburg, Germany
**2** Department of Condensed Matter Physics, Weizmann Institute of Science,
Rehovot, 76100, Israel
**3** Perimeter Institute for Theoretical Physics,
Waterloo, Ontario N2L 2Y5, Canada
**4** Würzburg-Dresden Cluster of Excellence ct.qmat,
Am Hubland, 97074 Würzburg, Germany
[*]alf@physik.uni-wuerzburg.de

December 7, 2022

## Abstract

The *Algorithms for Lattice Fermions* package provides a general code for the
finite-temperature and projective auxiliary-field quantum Monte Carlo algorithm.
The code is engineered to be able to simulate any model that can be written
in terms of sums of single-body operators, of squares of single-body operators
and single-body operators coupled to a bosonic field with given dynamics. The
package includes five pre-defined model classes: SU(N) Kondo, SU(N) Hubbard,
SU(N) t-V and SU(N) models with long range Coulomb repulsion on honeycomb,
square and N-leg lattices, as well as $Z_2$ unconstrained lattice gauge theories cou-
pled to fermionic and $Z_2$ matter. An implementation of the stochastic Maximum
Entropy method is also provided. One can download the code from our Git in-
stance at https://git.physik.uni-wuerzburg.de/ALF/ALF/-/tree/ALF-2.1 and sign
in to file issues.

# 1  Introduction

## 1.1  Motivation

The aim of the ALF project is to provide a general formulation of the auxiliary-field QMC method that enables one to promptly play with different model Hamiltonians at minimal programming cost. The package also comes with a number of predefined Hamiltonians aimed at producing benchmark results.

The auxiliary-field quantum Monte Carlo (QMC) approach is the algorithm of choice to simulate thermodynamic properties of a variety of correlated electron systems in the solid state and beyond [1–6]. Apart from the physics of the canonical Hubbard model [7,8], the topics one can investigate in detail include correlation effects in the bulk and on surfaces of topological

insulators [9–12], quantum phase transitions between Dirac fermions and insulators [13–20], deconfined quantum critical points [18, 21–24], constrained and unconstrained lattice gauge theories [21, 25–30], heavy fermion systems [31–36], nematic [37, 38] and magnetic [39, 40] quantum phase transitions in metals, antiferromagnetism in metals [41], superconductivity in spin-orbit split and in topological flat bands [42–44], SU(N) symmetric models [45–50], long-ranged Coulomb interactions in graphene systems [51–55], cold atomic gases [56], low energy nuclear physics [57] that may require formulations in the canonical ensemble [58, 59], entanglement entropies and spectra [60–66], electron-phonon systems [67–69], Landau level regularization of continuum theories [70, 71], Yukawa SYK models [72] and even spin systems [73] among others. This ever-growing list of topics is based on algorithmic progress and on recent symmetry-related insights [74–77] that lead to formulations free of the negative sign problem for a number of model systems with very rich phase diagrams.

Auxiliary-field methods can be formulated in a number of very different ways. The fields define the configuration space $\mathcal{C}$. They can stem from the Hubbard-Stratonovich (HS) [78] transformation required to decouple the many-body interacting term into a sum of non-interacting problems, or they can correspond to bosonic modes with predefined dynamics such as phonons or gauge fields. In all cases, the result is that the grand-canonical partition function takes the form

$$Z = \text{Tr}\left(e^{-\beta\hat{\mathcal{H}}}\right) = \sum_{\mathcal{C}} e^{-S(\mathcal{C})}, \tag{1}$$

where $\beta$ corresponds to the inverse temperature and $S$ is the action of non-interacting fermions subject to a space-time fluctuating auxiliary field. The high-dimensional integration over the fields is carried out stochastically. In this formulation of many-body quantum systems, there is no reason for the action to be a real number. Thereby $e^{-S(\mathcal{C})}$ cannot be interpreted as a weight. To circumvent this problem one can adopt re-weighting schemes and sample $|e^{-S(\mathcal{C})}|$. This invariably leads to the so-called *negative sign problem*, with the associated exponential computational scaling in system size and inverse temperature [79]. The sign problem is formulation dependent and, as mentioned above, there has been tremendous progress at identifying an increasing number of models not affected by the negative sign problem which cover a rich domain of collective emergent phenomena. For continuous fields, the stochastic integrations can be carried out with Langevin dynamics or hybrid methods [80]. However, for many problems one can get away with discrete fields [81]. In this case, Monte Carlo importance sampling will often be put to use [82]. We note that due to the non-locality of the fermion determinant (see below), cluster updates, such as in the loop or stochastic series expansion algorithms for quantum spin systems [83–85], are hard to formulate for this class of problems. The search for efficient updating schemes that quickly wander through the configuration space defines ongoing challenges.

Formulations differ not only in the choice of the fields, continuous or discrete, and sampling strategy, but also by the formulation of the action itself. For a given field configuration, integrating out fermionic degrees of freedom generically leads to a fermionic determinant of dimension $\beta N$ where $N$ is the volume of the system. Working with this determinant leads to the Hirsch-Fye approach [86] and the computational effort scales[1] as $\mathcal{O}\left(\beta N\right)^3$. The Hirsch-Fye algorithm is the method of choice for impurity problems, but has in general been outperformed by a class of so-called continuous-time quantum Monte Carlo approaches [87–89]. One key advantage of continuous-time methods is being action based, allowing one to better handle

---

[1] Here we implicitly assume the absence of negative sign problem.

the retarded interactions obtained when integrating out fermion or boson baths. However, in high dimensions or at low temperatures, the cubic scaling originating from the fermionic determinant is expensive. To circumvent this, the hybrid Monte-Carlo approach [5, 90, 91] expresses the fermionic determinant in terms of a Gaussian integral thereby introducing a new variable in the Monte Carlo integration. The resulting algorithm is the method of choice for lattice gauge theories in 3+1 dimensions and has been used to provide *ab initio* estimates of light hadron masses starting from quantum chromodynamics [92].

The approach we adopt lies between the above two *extremes.* We keep the fermionic determinant, but formulate the problem so as to work only with $N \times N$ matrices. This Blankenbecler, Scalapino, Sugar (BSS) algorithm scales linearly in imaginary time $\beta$, but remains cubic in the volume $N$. Furthermore, the algorithm can be formulated either in a projective manner [3, 4], adequate to obtain zero temperature properties in the canonical ensemble, or at finite temperatures, in the grand-canonical ensemble [2]. In this documentation we summarize the essential aspects of the auxiliary-field QMC approach, and refer the reader to Refs. [6, 93] for complete reviews.

## 1.2 Definition of the Hamiltonian

The first and most fundamental part of the project is to define a general Hamiltonian which can accommodate a large class of models. Our approach is to express the model as a sum of one-body terms, a sum of two-body terms each written as a perfect square of a one body term, as well as a one-body term coupled to a bosonic field with dynamics to be specified by the user. Writing the interaction in terms of sums of perfect squares allows us to use generic forms of discrete approximations to the HS transformation [94, 95]. Symmetry considerations are imperative to increase the speed of the code. We therefore include a *color* index reflecting an underlying SU(N) color symmetry as well as a *flavor* index reflecting the fact that after the HS transformation, the fermionic determinant is block diagonal in this index.

The class of solvable models includes Hamiltonians $\hat{\mathcal{H}}$ that have the following general form:

$$\hat{\mathcal{H}} = \hat{\mathcal{H}}_T + \hat{\mathcal{H}}_V + \hat{\mathcal{H}}_I + \hat{\mathcal{H}}_{0,I} \,, \text{ where} \tag{2}$$

$$\hat{\mathcal{H}}_T = \sum_{k=1}^{M_T} \sum_{\sigma=1}^{N_{\mathrm{col}}} \sum_{s=1}^{N_{\mathrm{fl}}} \sum_{x,y}^{N_{\mathrm{dim}}} \hat{c}_{x\sigma s}^\dagger T_{xy}^{(ks)} \hat{c}_{y\sigma s} \equiv \sum_{k=1}^{M_T} \hat{T}^{(k)} \,, \tag{3}$$

$$\hat{\mathcal{H}}_V = \sum_{k=1}^{M_V} U_k \left\{ \sum_{\sigma=1}^{N_{\mathrm{col}}} \sum_{s=1}^{N_{\mathrm{fl}}} \left[ \left( \sum_{x,y}^{N_{\mathrm{dim}}} \hat{c}_{x\sigma s}^\dagger V_{xy}^{(ks)} \hat{c}_{y\sigma s} \right) + \alpha_{ks} \right] \right\}^2 \equiv \sum_{k=1}^{M_V} U_k \left( \hat{V}^{(k)} \right)^2 \,, \tag{4}$$

$$\hat{\mathcal{H}}_I = \sum_{k=1}^{M_I} \hat{Z}_k \left( \sum_{\sigma=1}^{N_{\mathrm{col}}} \sum_{s=1}^{N_{\mathrm{fl}}} \sum_{x,y}^{N_{\mathrm{dim}}} \hat{c}_{x\sigma s}^\dagger I_{xy}^{(ks)} \hat{c}_{y\sigma s} \right) \equiv \sum_{k=1}^{M_I} \hat{Z}_k \hat{I}^{(k)} \,. \tag{5}$$

The indices and symbols used above have the following meaning:

- The number of fermion *flavors* is set by $N_{\mathrm{fl}}$. After the HS transformation, the action will be block diagonal in the flavor index.

- The number of fermion *colors* is set[2] by $N_{\mathrm{col}}$. The Hamiltonian is invariant under SU($N_{\mathrm{col}}$) rotations.

---

[2]Note that in the code $N_{\mathrm{col}} \equiv$ `N_SUN`.

- $N_{\text{dim}}$ is the total number of spacial vertices: $N_{\text{dim}} = N_{\text{unit-cell}}N_{\text{orbital}}$, where $N_{\text{unit-cell}}$ is the number of unit cells of the underlying Bravais lattice and $N_{\text{orbital}}$ is the number of orbitals per unit cell.

- The indices $x$ and $y$ label lattice sites where $x, y = 1, \cdots, N_{\text{dim}}$.

- Therefore, the matrices $\boldsymbol{T}^{(ks)}$, $\boldsymbol{V}^{(ks)}$ and $\boldsymbol{I}^{(ks)}$ are of dimension $N_{\text{dim}} \times N_{\text{dim}}$.

- The number of interaction terms is labeled by $M_V$ and $M_I$. $M_T > 1$ would allow for a checkerboard decomposition.

- $\hat{c}^{\dagger}_{y\sigma s}$ is a second-quantized operator that creates an electron in a Wannier state centered around lattice site $y$, with color $\sigma$, and flavor index $s$. The operators satisfy the anti-commutation relations:

$$\left\{ \hat{c}^{\dagger}_{y\sigma s}, \hat{c}_{y'\sigma' s'} \right\} = \delta_{yy'}\delta_{ss'}\delta_{\sigma\sigma'}, \quad \text{and} \quad \left\{ \hat{c}_{y\sigma s}, \hat{c}_{y'\sigma' s'} \right\} = 0. \tag{6}$$

- $\alpha_{ks}$ is a complex number.

The bosonic part of the general Hamiltonian (2) is $\hat{\mathcal{H}}_{0,I} + \hat{\mathcal{H}}_I$ and has the following properties:

- $\hat{Z}_k$ couples to a general one-body term. We will work in a basis where this operator is diagonal: $\hat{Z}_k|\phi\rangle = \phi_k|\phi\rangle$. $\phi_k$ is a real number or an Ising variable. Hence $\hat{Z}_k$ can correspond to the Pauli matrix $\hat{\sigma}_z$ or to the position operator.

- The dynamics of the bosonic field is given by $\hat{\mathcal{H}}_{0,I}$. This term is not specified here; it has to be specified by the user and becomes relevant when the Monte Carlo update probability is computed in the code

Note that the matrices $\boldsymbol{T}^{(ks)}$, $\boldsymbol{V}^{(ks)}$ and $\boldsymbol{I}^{(ks)}$ explicitly depend on the flavor index $s$ but not on the color index $\sigma$. The color index $\sigma$ only appears in the second quantized operators such that the Hamiltonian is manifestly $\text{SU}(N_{\text{col}})$ symmetric. We also require the matrices $\boldsymbol{T}^{(ks)}$, $\boldsymbol{V}^{(ks)}$ and $\boldsymbol{I}^{(ks)}$ to be Hermitian.

## 1.3   Outline and What is new

In order to use the program, a minimal understanding of the algorithm is necessary. Its code is written in Fortran, according to the 2008 standard, and natively uses MPI (MPI 3.0 compliant implementation needed) for parallel runs on supercomputing systems. In this documentation we aim to present in enough detail both the algorithm and its implementation to allow the user to confidently use and modify the program.

ALF 2.1 introduces a change that is **not backwards compatible** with ALF 2.0: Hamiltonians are now written as submodules of the module `Hamiltonian_main`. Sec. 5.6 details the new implementation and should ease adjusting existing code to ALF 2.1, which takes 2 steps:

- restructuring your Hamiltonian as a submodule,

- adding your Hamiltonian name to the file `Prog/Hamiltonians.list`.

In Sec. 2, we summarize the steps required to formulate the many-body, imaginary-time propagation in terms of a sum over HS and bosonic fields of one-body, imaginary-time propagators. To simulate a model not already included in ALF, the user has to provide this one-body, imaginary-time propagator for a given configuration of HS and bosonic fields. In

this section we also touch on how to compute observables and on how we deal with the negative sign problem. Since version 2.0 ALF has a number of new updating schemes. The package comes with the possibility to implement global updates in space and time or only in space. We provide parallel-tempering and Langevin dynamics options. Another possibility is to implement symmetric Trotter decompositions. At the end of the section we comment on the issue of stabilization for the finite temperature code.

In Sec. 3, we describe the projective version of the algorithm, constructed to produce ground state properties. One can very easily switch between projective and finite temperature codes, but a trial wave function must be provided.

One of the key challenges in Monte Carlo methods is to adequately evaluate the stochastic error. In Sec. 4 we provide an explicit example of how to correctly estimate the error.

Section 5 is devoted to the data structures that are needed to implement the model, as well as to the input and output file structure. The data structures include an `Operator` type to optimally work with sparse Hermitian matrices, a `Lattice` type to define one- and two-dimensional Bravais lattices, a generic `Fields` type for the auxiliary fields, two `Observable` types to handle scalar observables (e.g., total energy) and equal-time or time-displaced two-point correlation functions (e.g., spin-spin correlations) and finally a `Wavefunction` type to define the trial wave function in the projective code. At the end of this section we comment on the file structure.

In Sec. 6 we provide details on running the code using the shell. As an alternative the user can download a separate project, pyALF that provides a convenient python interface as well as Jupyter notebooks.

The package has a set of predefined structures that allow easy reuse of lattices, observables, interactions and trial wave functions. Although convenient, this extra layer of abstraction might render ALF harder to modify. To circumvent this we make available an implementation of a plain vanilla Hubbard model on the square lattice (see Sec. 7) that shows explicitly how to implement this basic model without making use of predefined structures. We believe that this is a good starting point to modify a Hamiltonian from scratch, as exemplified in the package's Tutorial.

Sec. 8 introduces the sets of predefined lattices, hopping matrices, interactions, observables and trial wave functions available. The goal here is to provide a library so as to facilitate implementation of new Hamiltonians.

The package comes with as set of Hamiltonians, described in Sec. 9, which includes: (i) SU(N) Hubbard models, (ii) SU(N) t-V models, (iii) SU(N) Kondo lattice models, (iv) Models with long ranged coulomb interactions, and (v) Generic $\mathbf{Z}_2$ lattice gauge theories coupled to $\mathbf{Z}_2$ matter and fermions. These model classes are built on the predefined structures.

In Sec. 10 we describe how to use our implementation of the stochastic analytical continuation [96, 97].

Finally, in Sec. 11 we list a number of features being considered for future releases of the ALF package.

## 2  Auxiliary Field Quantum Monte Carlo: finite temperature

We start this section by deriving the detailed form of the partition function and outlining the computation of observables (Sec. 2.1.1 - 2.1.3). Next, we present a number of update

strategies, namely local updates, global updates, parallel tempering and Langevin dynamics (Sec. 2.2). We then discuss the Trotter error, both for symmetric and asymmetric decompositions (Sec. 2.3) and, finally, we describe the measures we have implemented to make the code numerically stable (Sec. 2.4).

## 2.1 Formulation of the method

Our aim is to compute observables for the general Hamiltonian (2) in thermodynamic equilibrium as described by the grand-canonical ensemble. We show below how the grand-canonical partition function can be rewritten as

$$Z = \text{Tr}\left(e^{-\beta\hat{\mathcal{H}}}\right) = \sum_C e^{-S(C)} + \mathcal{O}(\Delta\tau^2), \tag{7}$$

and define the space of configurations $C$. Note that the chemical potential term is already included in the definition of the one-body term $\hat{\mathcal{H}}_T$, see Eq. (3), of the general Hamiltonian. The essential ingredients of the auxiliary-field quantum Monte Carlo implementation in the ALF package are the following:

- We discretize the imaginary time propagation: $\beta = \Delta\tau L_{\text{Trotter}}$. Generically this introduces a systematic Trotter error of $\mathcal{O}(\Delta\tau)^2$ [98]. We note that there has been considerable effort at getting rid of the Trotter systematic error and to formulate a genuine continuous-time BSS algorithm [99]. To date, efforts in this direction that are based on a CT-AUX type formulation [100, 101] face two issues. The first one is that they are restricted to a class of models with Hubbard-type interactions

$$(\hat{n}_i - 1)^2 = (\hat{n}_i - 1)^4, \tag{8}$$

in order for the basic CT-AUX equation [102],

$$1 + \frac{U}{K}(\hat{n}_i - 1)^2 = \frac{1}{2}\sum_{s=\pm 1} e^{\alpha s(\hat{n}_i - 1)} \ \text{ with } \ \frac{U}{K} = \cosh(\alpha) - 1 \ \text{ and } \ K \in \mathbb{R}, \tag{9}$$

to hold. The second issue is that it is hard to formulate a computationally efficient algorithm. Given this situation, if eliminating the Trotter systematic error is required, it turns out that extrapolating to small imaginary-time steps using the multi-grid method [103–105] is a more efficient scheme.

There has also been progress in efficient continuous-time methods using techniques that draw from the Stochastic Series Expansion [106] which can be combined with fermion bag ideas [107]. However, these techniques are even more restricted to a specific class of Hamiltonians, those that can be expressed as sums of exponentiated fermionic bilinear terms $\hat{H} = \sum_i \hat{h}^{(i)}$, where

$$\hat{h}^{(i)} = -\gamma^{(i)} e^{\sum_{jk} \alpha_{jk}^{(i)} \hat{c}_j^\dagger \hat{c}_k + \text{H.c.}} \tag{10}$$

Stabilization can also be costly depending on the parameters, particularly for large $\alpha$ values [108].

- Having isolated the two-body term, we apply Gauß-Hermite quadrature [109] to the continuous HS transform and obtain the discrete HS transformation [94, 95]:

$$e^{\Delta\tau\lambda\hat{A}^2} = \frac{1}{4}\sum_{l=\pm1,\pm2}\gamma(l)e^{\sqrt{\Delta\tau\lambda}\eta(l)\hat{A}} + \mathcal{O}\left((\Delta\tau\lambda)^4\right) , \qquad (11)$$

where the fields $\eta$ and $\gamma$ take the values:

$$
\begin{aligned}
\gamma(\pm1) &= 1 + \sqrt{6}/3, & \eta(\pm1) &= \pm\sqrt{2\left(3-\sqrt{6}\right)}, \\
\gamma(\pm2) &= 1 - \sqrt{6}/3, & \eta(\pm2) &= \pm\sqrt{2\left(3+\sqrt{6}\right)}.
\end{aligned}
\qquad (12)
$$

Since the Trotter error is already of order $(\Delta\tau^2)$ per time slice, this transformation is next to exact. One can relate the expectation value of the field $\eta(l)$ to the operator $\hat{A}$ by noting that:

$$\frac{1}{4}\sum_{l=\pm1,\pm2}\gamma(l)e^{\sqrt{\Delta\tau\lambda}\eta(l)\hat{A}}\left(\frac{\eta(l)}{-2\sqrt{\Delta\tau\lambda}}\right) = e^{\Delta\tau\lambda\hat{A}^2}\hat{A} + \mathcal{O}\left((\Delta\tau\lambda)^3\right) \text{ and}$$

$$\frac{1}{4}\sum_{l=\pm1,\pm2}\gamma(l)e^{\sqrt{\Delta\tau\lambda}\eta(l)\hat{A}}\left(\frac{(\eta(l))^2-2}{4\Delta\tau\lambda}\right) = e^{\Delta\tau\lambda\hat{A}^2}\hat{A}^2 + \mathcal{O}\left((\Delta\tau\lambda)^2\right). \qquad (13)$$

- $\hat{Z}_k$ in Eq. (5) can stand for a variety of operators, such as the Pauli matrix $\hat{\sigma}_z$ – in which case the Ising spins take the values $s_k = \pm1$ – or the position operator – such that $\hat{Z}_k|\phi\rangle = \phi_k|\phi\rangle$, with $\phi_k$ a real number.

- From the above it follows that the Monte Carlo configuration space $C$ is given by the combined spaces of bosonic configurations and of HS discrete field configurations:

$$C = \{\phi_{i,\tau}, l_{j,\tau} \text{ with } i = 1\cdots M_I, \ j = 1\cdots M_V, \ \tau = 1\cdots L_{\text{Trotter}}\}. \qquad (14)$$

Here, the HS fields take the values $l_{j,\tau} = \pm2, \pm1$ and $\phi_{i,\tau}$ may, for instance, be a continuous real field or, if $\hat{Z}_k = \hat{\sigma}_z$, be restricted to $\pm1$.

### 2.1.1 The partition function

With the above, the partition function of the model (2) can be written as follows.

$$
\begin{aligned}
Z &= \text{Tr}\left(e^{-\beta\hat{\mathcal{H}}}\right) \\
&= \text{Tr}\left[e^{-\Delta\tau\hat{\mathcal{H}}_{0,I}}\prod_{k=1}^{M_V}e^{-\Delta\tau U_k\left(\hat{V}^{(k)}\right)^2}\prod_{k=1}^{M_I}e^{-\Delta\tau\hat{\sigma}_k\hat{I}^{(k)}}\prod_{k=1}^{M_T}e^{-\Delta\tau\hat{T}^{(k)}}\right]^{L_{\text{Trotter}}} + \mathcal{O}(\Delta\tau^2) \\
&= \sum_C\left(\prod_{k=1}^{M_V}\prod_{\tau=1}^{L_{\text{Trotter}}}\gamma_{k,\tau}\right)e^{-S_0(\{s_{i,\tau}\})}\times \\
&\quad \text{Tr}_F\left\{\prod_{\tau=1}^{L_{\text{Trotter}}}\left[\prod_{k=1}^{M_V}e^{\sqrt{-\Delta\tau U_k}\eta_{k,\tau}\hat{V}^{(k)}}\prod_{k=1}^{M_I}e^{-\Delta\tau s_{k,\tau}\hat{I}^{(k)}}\prod_{k=1}^{M_T}e^{-\Delta\tau\hat{T}^{(k)}}\right]\right\} + \mathcal{O}(\Delta\tau^2). \qquad (15)
\end{aligned}
$$

In the above, the trace Tr runs over the bosonic and fermionic degrees of freedom, and $\mathrm{Tr_F}$ only over the fermionic Fock space. $S_0(\{s_{i,\tau}\})$ is the action corresponding to the bosonic Hamiltonian, and is only dependent on the bosonic fields so that it can be pulled out of the fermionic trace. We have adopted the shorthand notation $\eta_{k,\tau} \equiv \eta(l_{k,\tau})$ and $\gamma_{k,\tau} \equiv \gamma(l_{k,\tau})$. At this point, and since for a given configuration $C$ we are dealing with a free propagation, we can integrate out the fermions to obtain a determinant:

$$
\mathrm{Tr_F} \left\{ \prod_{\tau=1}^{L_{\mathrm{Trotter}}} \left[ \prod_{k=1}^{M_V} e^{\sqrt{-\Delta\tau U_k}\eta_{k,\tau}\hat{V}^{(k)}} \prod_{k=1}^{M_I} e^{-\Delta\tau s_{k,\tau}\hat{I}^{(k)}} \prod_{k=1}^{M_T} e^{-\Delta\tau\hat{T}^{(k)}} \right] \right\} =
$$

$$
\prod_{s=1}^{N_{\mathrm{fl}}} \left[ e^{\sum_{k=1}^{M_V} \sum_{\tau=1}^{L_{\mathrm{Trotter}}} \sqrt{-\Delta\tau U_k}\alpha_{k,s}\eta_{k,\tau}} \right]^{N_{\mathrm{col}}} \times
$$

$$
\prod_{s=1}^{N_{\mathrm{fl}}} \left[ \det \left( \mathbb{1} + \prod_{\tau=1}^{L_{\mathrm{Trotter}}} \prod_{k=1}^{M_V} e^{\sqrt{-\Delta\tau U_k}\eta_{k,\tau}\boldsymbol{V}^{(ks)}} \prod_{k=1}^{M_I} e^{-\Delta\tau s_{k,\tau}\boldsymbol{I}^{(ks)}} \prod_{k=1}^{M_T} e^{-\Delta\tau\boldsymbol{T}^{(ks)}} \right) \right]^{N_{\mathrm{col}}}, \tag{16}
$$

where the matrices $\boldsymbol{T}^{(ks)}$, $\boldsymbol{V}^{(ks)}$, and $\boldsymbol{I}^{(ks)}$ define the Hamiltonian [Eq. (2) - (5)]. All in all, the partition function is given by:

$$
Z = \sum_C e^{-S_0(\{s_{i,\tau}\})} \left( \prod_{k=1}^{M_V} \prod_{\tau=1}^{L_{\mathrm{Trotter}}} \gamma_{k,\tau} \right) e^{N_{\mathrm{col}} \sum_{s=1}^{N_{\mathrm{fl}}} \sum_{k=1}^{M_V} \sum_{\tau=1}^{L_{\mathrm{Trotter}}} \sqrt{-\Delta\tau U_k}\alpha_{k,s}\eta_{k,\tau}} \times \prod_{s=1}^{N_{\mathrm{fl}}} \left[ \det \left( \mathbb{1} \right. \right.
$$

$$
\left. \left. + \prod_{\tau=1}^{L_{\mathrm{Trotter}}} \prod_{k=1}^{M_V} e^{\sqrt{-\Delta\tau U_k}\eta_{k,\tau}\boldsymbol{V}^{(ks)}} \prod_{k=1}^{M_I} e^{-\Delta\tau s_{k,\tau}\boldsymbol{I}^{(ks)}} \prod_{k=1}^{M_T} e^{-\Delta\tau\boldsymbol{T}^{(ks)}} \right) \right]^{N_{\mathrm{col}}} + \mathcal{O}(\Delta\tau^2)
$$

$$
\equiv \sum_C e^{-S(C)} + \mathcal{O}(\Delta\tau^2). \tag{17}
$$

In the above, one notices that the weight factorizes in the flavor index. The color index raises the determinant to the power $N_{\mathrm{col}}$. This corresponds to an explicit $\mathrm{SU}(N_{\mathrm{col}})$ symmetry for each configuration. This symmetry is manifest in the fact that the single particle Green functions are color independent, again for each given configuration $C$.

### 2.1.2 Observables

In the auxiliary-field QMC approach, the single-particle Green function plays a crucial role. It determines the Monte Carlo dynamics and is used to compute observables. Consider the observable:

$$
\langle \hat{O} \rangle = \frac{\mathrm{Tr} \left[ e^{-\beta\hat{H}}\hat{O} \right]}{\mathrm{Tr} \left[ e^{-\beta\hat{H}} \right]} = \sum_C P(C)\langle\langle\hat{O}\rangle\rangle_{(C)}, \quad \text{where } P(C) = \frac{e^{-S(C)}}{\sum_C e^{-S(C)}} \tag{18}
$$

and $\langle\langle\hat{O}\rangle\rangle_{(C)}$ denotes the observed value of $\hat{O}$ for a given configuration $C$. For a given configuration $C$ one can use Wick's theorem to compute $O(C)$ from the knowledge of the single-particle Green function:

$$
G(x,\sigma,s,\tau|x',\sigma',s',\tau') = \langle\langle \mathcal{T}\hat{c}_{x\sigma s}(\tau)\hat{c}_{x'\sigma's'}^{\dagger}(\tau') \rangle\rangle_C, \tag{19}
$$

where $\mathcal{T}$ denotes the imaginary-time ordering operator. The corresponding equal-time quantity reads

$$G(x, \sigma, s, \tau | x', \sigma', s', \tau) = \langle\langle \hat{c}_{x\sigma s}(\tau) \hat{c}^\dagger_{x'\sigma's'}(\tau)\rangle\rangle_C. \tag{20}$$

Since, for a given HS field, translation invariance in imaginary-time is broken, the Green function has an explicit $\tau$ and $\tau'$ dependence. On the other hand it is diagonal in the flavor index, and independent of the color index. The latter reflects the explicit SU(N) color symmetry present at the level of individual HS configurations. As an example, one can show that the equal-time Green function at $\tau = 0$ reads [6]:

$$G(x, \sigma, s, 0 | x', \sigma, s, 0) = \left( \mathbb{1} + \prod_{\tau=1}^{L_{\text{Trotter}}} \boldsymbol{B}_\tau^{(s)} \right)^{-1}_{x,x'} \tag{21}$$

with

$$\boldsymbol{B}_\tau^{(s)} = \prod_{k=1}^{M_V} e^{\sqrt{-\Delta\tau U_k}\eta_{k,\tau}\boldsymbol{V}^{(ks)}} \prod_{k=1}^{M_I} e^{-\Delta\tau s_{k,\tau}\boldsymbol{I}^{(ks)}} \prod_{k=1}^{M_T} e^{-\Delta\tau \boldsymbol{T}^{(ks)}}. \tag{22}$$

To compute equal-time, as well as time-displaced observables, one can make use of Wick's theorem. A convenient formulation of this theorem for QMC simulations reads:

$$\langle\langle \mathcal{T} \hat{c}^\dagger_{\underline{x}_1}(\tau_1) \hat{c}_{\underline{x}'_1}(\tau'_1) \cdots \hat{c}^\dagger_{\underline{x}_n}(\tau_n) \hat{c}_{\underline{x}'_n}(\tau'_n)\rangle\rangle_C =$$

$$\det \begin{bmatrix} \langle\langle \mathcal{T} \hat{c}^\dagger_{\underline{x}_1}(\tau_1) \hat{c}_{\underline{x}'_1}(\tau'_1)\rangle\rangle_C & \langle\langle \mathcal{T} \hat{c}^\dagger_{\underline{x}_1}(\tau_1) \hat{c}_{\underline{x}'_2}(\tau'_2)\rangle\rangle_C & \cdots & \langle\langle \mathcal{T} \hat{c}^\dagger_{\underline{x}_1}(\tau_1) \hat{c}_{\underline{x}'_n}(\tau'_n)\rangle\rangle_C \\ \langle\langle \mathcal{T} \hat{c}^\dagger_{\underline{x}_2}(\tau_2) \hat{c}_{\underline{x}'_1}(\tau'_1)\rangle\rangle_C & \langle\langle \mathcal{T} \hat{c}^\dagger_{\underline{x}_2}(\tau_2) \hat{c}_{\underline{x}'_2}(\tau'_2)\rangle\rangle_C & \cdots & \langle\langle \mathcal{T} \hat{c}^\dagger_{\underline{x}_2}(\tau_2) \hat{c}_{\underline{x}'_n}(\tau'_n)\rangle\rangle_C \\ \vdots & \vdots & \ddots & \vdots \\ \langle\langle \mathcal{T} \hat{c}^\dagger_{\underline{x}_n}(\tau_n) \hat{c}_{\underline{x}'_1}(\tau'_1)\rangle\rangle_C & \langle\langle \mathcal{T} \hat{c}^\dagger_{\underline{x}_n}(\tau_n) \hat{c}_{\underline{x}'_2}(\tau'_2)\rangle\rangle_C & \cdots & \langle\langle \mathcal{T} \hat{c}^\dagger_{\underline{x}_n}(\tau_n) \hat{c}_{\underline{x}'_n}(\tau'_n)\rangle\rangle_C \end{bmatrix}. \tag{23}$$

Here, we have defined the super-index $\underline{x} = \{x, \sigma, s\}$.

Wick's theorem can be also used to express a reduced density matrix, i.e., the density matrix for a subsystem, in terms of its correlations [110]. Within the framework of Auxiliary-Field QMC, this allows to express a reduced density matrix $\hat{\rho}_A$ for a subsystem $A$ as [60]

$$\hat{\rho}_A = \sum_C P(C) \det(\mathbb{1} - G_A(\tau_0; C)) e^{-c^\dagger_{\underline{x}} H^{(A)}_{\underline{x},\underline{x}'} c_{\underline{x}'}}, \qquad H^{(A)} \equiv \ln\left\{ \left[ (G_A(\tau_0; C))^T \right]^{-1} - \mathbb{1} \right\}, \tag{24}$$

where $G_A(\tau_0; C)$ is the equal-time Green's function matrix restricted on the subsystem $A$ and at a given time-slice $\tau_0$. In Eq. (24) an implicit summation over repeated indexes $\underline{x}, \underline{x}' \in A$ is assumed. Interestingly, Eq. (24) holds also when $A$ is the entire system: in this case, it provides an alternative expression for the density matrix, or the (normalized) partition function, as a superposition of Gaussian operators. Eq. (24) is the starting point for computing the entanglement Hamiltonian [64] and the Rényi entropies [60, 62, 63]. A short review on various computational approaches to quantum entanglement in interacting fermionic models can be found in Ref. [66]. ALF provides predefined observables to compute the second Rényi entropy and its associated mutual information, see Sec. 8.4.11.

In Sec. 8.4 we describe the equal-time and time-displaced correlation functions that come predefined in ALF. Using the above formulation of Wick's theorem, arbitrary correlation functions can be computed (see Appendix A). We note, however, that the program is limited to the calculation of observables that contain only two different imaginary times.

### 2.1.3   Reweighting and the sign problem

In general, the action $S(C)$ will be complex, thereby inhibiting a direct Monte Carlo sampling of $P(C)$. This leads to the infamous sign problem. The sign problem is formulation dependent and as noted above, much progress has been made at understanding the class of models that can be formulated without encountering this problem [74–77]. When the average sign is not too small, we can nevertheless compute observables within a reweighting scheme. Here we adopt the following scheme. First note that the partition function is real such that:

$$Z = \sum_C e^{-S(C)} = \sum_C \overline{e^{-S(C)}} = \sum_C \mathrm{Re}\left[e^{-S(C)}\right].$$   (25)

Thereby[3] and with the definition

$$\mathrm{sgn}(C) = \frac{\mathrm{Re}\left[e^{-S(C)}\right]}{\left|\mathrm{Re}\left[e^{-S(C)}\right]\right|},$$   (26)

the computation of the observable [Eq. (18)] is re-expressed as follows:

$$
\begin{aligned}
\langle \hat{O} \rangle &= \frac{\sum_C e^{-S(C)} \langle\!\langle \hat{O} \rangle\!\rangle_{(C)}}{\sum_C e^{-S(C)}} \\
&= \frac{\sum_C \mathrm{Re}\left[e^{-S(C)}\right] \frac{e^{-S(C)}}{\mathrm{Re}[e^{-S(C)}]} \langle\!\langle \hat{O} \rangle\!\rangle_{(C)}}{\sum_C \mathrm{Re}\left[e^{-S(C)}\right]} \\
&= \frac{\left\{\sum_C \left|\mathrm{Re}\left[e^{-S(C)}\right]\right| \mathrm{sgn}(C) \frac{e^{-S(C)}}{\mathrm{Re}[e^{-S(C)}]} \langle\!\langle \hat{O} \rangle\!\rangle_{(C)}\right\} / \sum_C \left|\mathrm{Re}\left[e^{-S(C)}\right]\right|}{\left\{\sum_C \left|\mathrm{Re}\left[e^{-S(C)}\right]\right| \mathrm{sgn}(C)\right\} / \sum_C \left|\mathrm{Re}\left[e^{-S(C)}\right]\right|} \\
&= \frac{\left\langle \mathrm{sgn} \frac{e^{-S}}{\mathrm{Re}[e^{-S}]} \langle\!\langle \hat{O} \rangle\!\rangle \right\rangle_{\overline{P}}}{\langle \mathrm{sgn} \rangle_{\overline{P}}}.
\end{aligned}
$$   (27)

The average sign is

$$\langle \mathrm{sgn} \rangle_{\overline{P}} = \frac{\sum_C \left|\mathrm{Re}\left[e^{-S(C)}\right]\right| \mathrm{sgn}(C)}{\sum_C \left|\mathrm{Re}\left[e^{-S(C)}\right]\right|},$$   (28)

and we have $\langle \mathrm{sgn} \rangle_{\overline{P}} \in \mathbb{R}$ per definition. The Monte Carlo simulation samples the probability distribution

$$\overline{P}(C) = \frac{\left|\mathrm{Re}\left[e^{-S(C)}\right]\right|}{\sum_C \left|\mathrm{Re}\left[e^{-S(C)}\right]\right|}.$$   (29)

such that the nominator and denominator of Eq. (27) can be computed.

Notice that, for the Langevin updating scheme with variable Langevin time step, a straightforward generalization of the equations above is used, see Sec. 2.2.6.

The negative sign problem is still an issue because the average sign is a ratio of two partition functions and one can argue that

$$\langle \mathrm{sgn} \rangle_{\overline{P}} \propto e^{-\Delta N \beta},$$   (30)

---

[3]The attentive reader will have noticed that for arbitrary Trotter decompositions, the imaginary time propagator is not necessarily Hermitian. Thereby, the above equation is correct only up to corrections stemming from the controlled Trotter systematic error.

where $\Delta$ is an intensive positive quantity and $N\beta$ denotes the Euclidean volume. In a Monte Carlo simulation the error scales as $1/\sqrt{T_{\text{CPU}}}$ where $T_{\text{CPU}}$ corresponds to the computational time. Since the error on the average sign has to be much smaller than the average sign itself, one sees that:

$$T_{\text{CPU}} \gg e^{2\Delta N\beta}. \tag{31}$$

Two comments are in order. First, the presence of a sign problem invariably leads to an exponential increase of CPU time as a function of the Euclidean volume. And second, $\Delta$ is formulation dependent. For instance, at finite doping, the SU(2) invariant formulation of the Hubbard model presented in Sec. 9.1 has a much more severe sign problem than the formulation (presented in the same section) where the HS field couples to the $z$-component of the magnetization. Optimization schemes minimize $\Delta$ have been put forward in [111, 112].

## 2.2 Updating schemes

The program allows for different types of updating schemes, which are described below and summarized in Tab. 1. With the exception of Langevin dynamics, for a given configuration $C$, we propose a new one, $C'$, with a given probability $T_0(C \to C')$ and accept it according to the Metropolis-Hastings acceptance-rejection probability,

$$P(C \to C') = \min\left(1, \frac{T_0(C' \to C)W(C')}{T_0(C \to C')W(C)}\right), \tag{32}$$

so as to guarantee the stationarity condition. Here, $W(C) = \left|\text{Re}\left[e^{-S(C)}\right]\right|$.

Predicting how efficient a certain Monte Carlo update scheme will turn out to be for a given simulation is very hard, so one must typically resort to testing to find out which option produces best results. Methods to optimize the acceptance of global moves include Hybrid Monte Carlo [80] as well as self-learning techniques [113, 114]. Langevin dynamics stands apart, and as we will see does not depend on the Metropolis-Hastings acceptance-rejection scheme.

### 2.2.1 Sequential single spin flips

The program adopts per default a sequential, single spin-flip strategy. It will visit sequentially each HS field in the space-time operator list and propose a spin flip. Consider the Ising spin $s_{i,\tau}$. By default (`Propose_S0=.false.`), we will flip it with probability 1, such that for this local move the proposal matrix is symmetric. If we are considering the HS field $l_{i,\tau}$ we will propose with probability 1/3 one of the other three possible fields. For a continuous field, we modify it with a box distribution of width `Amplitude` centered around the origin. The default value of `Amplitude` is set to unity. These updating rules are defined in the `Fields_mod.F90` module (see Sec. 5.2). Again, for these local moves, the proposal matrix is symmetric. Hence in all cases we will accept or reject the move according to

$$P(C \to C') = \min\left(1, \frac{W(C')}{W(C)}\right). \tag{33}$$

This default updating scheme can be overruled by, e.g., setting `Global_tau_moves` to `.true.` and not setting `Nt_sequential_start` and `Nt_sequential_end` (see Sec. 5.7.1). It is also worth noting that this type of sequential spin-flip updating does not satisfy detailed balance, but rather the more fundamental stationarity condition [82].

| Updating schemes | Type | Description |
|---|---|---|
| `Sequential` | logical | (internal variable) If true, the configurations moves through sequential, single spin flips |
| `Propose_S0` | logical | If true, proposes sequential local moves according to the probability $e^{-S_0}$, where $S_0$ is the free Ising action. This option only works for `type=1` operator where the field corresponds to an Ising variable |
| `Global_tau_moves` | logical | Whether to carry out global moves on a single time slice. For a given time slice the user can define which part of the operator string is to be computed sequentially. This is specified by the variable `N_sequential_start` and `N_sequential_end`. A number of `N_tau_Global` user-defined global moves on the given time slice will then be carried out |
| `Global_moves` | logical | If true, allows for global moves in space and time. A user-defined number `N_Global` of global moves in space and time will be carried out at the end of each sweep |
| `Langevin` | logical | If true, Langevin dynamics is used exclusively (i.e., can only be used in association with tempering) |
| `Tempering` | Compiling option | Requires MPI and runs the code in a parallel tempering mode, also see Sec. 2.2.5, 6.2 |

Table 1: Variables required to control the updating scheme. Per default the program carries out sequential, single spin-flip sweeps, and logical variables are set to `.false.`.

## 2.2.2 Sampling of $e^{-S_0}$

The package can also propose single spin-flip updates according to a non-vanishing free bosonic action $S_0(C)$. This sampling scheme is used if the logical variable `Propose_S0` is set to `.true.`. As mentioned previously, this option only holds for Ising variables.

Consider an Ising spin at space-time $i, \tau$ in the configuration $C$. Flipping this spin generates the configuration $C'$ and we propose this move according to

$$T_0(C \to C') = \frac{e^{-S_0(C')}}{e^{-S_0(C')} + e^{-S_0(C)}} = 1 - \frac{1}{1 + e^{-S_0(C')}/e^{-S_0(C)}}. \tag{34}$$

Note that the function `S0` in the `Hamiltonian_Hubbard_include.h` module computes precisely the ratio
$e^{-S_0(C')}/e^{-S_0(C)}$, therefore $T_0(C \to C')$ is obtained without any additional calculation. The proposed move is accepted with the probability:

$$P(C \to C') = \min\left(1, \frac{e^{-S_0(C)}W(C')}{e^{-S_0(C')}W(C)}\right). \tag{35}$$

Note that, as can be seen from Eq. (17), the bare action $S_0(C)$ determining the dynamics of the bosonic configuration in the absence of coupling to the fermions does not enter the Metropolis acceptance-rejection step.

### 2.2.3   Global updates in space

This option allows one to carry out user-defined global moves on a single time slice. This option is enabled by setting the logical variable `Global_tau_moves` to `.true.`. Recall that the propagation over a time step $\Delta\tau$ (see Eq. 22) can be written as:

$$e^{-V_{M_I+M_V}(s_{M_I+M_V,\tau})}\cdots e^{-V_1(s_{1,\tau})}\prod_{k=1}^{M_T} e^{-\Delta\tau \boldsymbol{T}^{(k)}}, \tag{36}$$

where $e^{-V_n(s_n)}$ denotes one element of the operator list containing the HS fields. One can provide an interval of indices, [`Nt_sequential_start, Nt_sequential_end`], in which the operators will be updated sequentially. Setting `Nt_sequential_start` $= 1$ and `Nt_sequential_end` $= M_I + M_V$ reproduces the sequential single spin flip strategy of the above section.

The variable `N_tau_Global` sets the number of global moves carried out on each time slice `ntau`. Each global move is generated in the routine `Global_move_tau`, which is provided by the user in the Hamiltonian file. In order to define this move, one specifies the following variables:

- `Flip_length`: An integer stipulating the number of spins to be flipped.

- `Flip_list(1:Flip_length)`: Integer array containing the indices of the operators to be flipped.

- `Flip_value(1:Flip_length)`: `Flip_value(n)` is an integer containing the new value of the HS field for the operator `Flip_list(n)`.

- `T0_Proposal_ratio`: Real number containing the quotient

$$\frac{T_0(C' \to C)}{T_0(C \to C')}, \tag{37}$$

where $C'$ denotes the new configuration obtained by flipping the spins specified in the `Flip_list` array. Since we allow for a stochastic generation of the global move, it may very well be that no change is proposed. In this case, `T0_Proposal_ratio` takes the value 0 upon exit of the routine `Global_move_tau` and no update is carried out.

- `S0_ratio`: Real number containing the ratio $e^{-S_0(C')}/e^{-S_0(C)}$.

### 2.2.4   Global updates in time and space

The code allows for global updates as well. The user must then provide two additional functions (see `Hamiltonian_Hubbard_include.h`): `Global_move` and `Delta_S0_global(Nsigma_old)`.

The subroutine `Global_move(T0_Proposal_ratio,nsigma_old,size_clust)` proposes a global move. Its single input is the variable `nsigma_old` of type `Field` (see Section 5.2) that contains the full configuration $C$ stored in `nsigma_old%f(M_V + M_I, Ltrot)`. On output, the new configuration $C'$, determined by the user, is stored in the two-dimensional array `nsigma`, which is a global variable declared in the Hamiltonian module. Like for the global move in space (Sec. 2.2.3), `T0_Proposal_ratio` contains the proposal ratio $T_0(C' \to C)/T_0(C \to C')$. Since we allow for a stochastic generation of the global move, it may very well be that

no change is proposed. In this case, `T0_Proposal_ratio` takes the value 0 upon exit, and `nsigma = nsigma_old`. The real-valued `size_clust` gives the size of the proposed move (e.g., $\frac{\text{Number of flipped spins}}{\text{Total number of spins}}$). This is used to calculate the average sizes of proposed and accepted moves, which are printed in the `info` file. The variable `size_clust` is not necessary for the simulation, but may help the user to estimate the effectiveness of the global update.

In order to compute the acceptance-rejection ratio, the user must also provide a function `Delta_S0_global(nsigma_old)` that computes the ratio $e^{-S_0(C')}/e^{-S_0(C)}$. Again, the configuration $C'$ is given by the field `nsigma`.

The variable `N_Global` determines the number of global updates performed per sweep. Note that global updates are expensive, since they require a complete recalculation of the weight.

### 2.2.5 Parallel tempering

Exchange Monte Carlo [115], or parallel tempering [116], is a possible route to overcome sampling issues in parts of the parameter space. Let $h$ be a parameter which one can vary without altering the configuration space $\{C\}$ and let us assume that for some values of $h$ one encounters sampling problems. For example, in the realm of spin glasses, $h$ could correspond to the inverse temperature. Here at high temperatures the phase space is easily sampled, but at low temperatures simulations get stuck in local minima. For quantum systems, $h$ could trigger a quantum phase transition where sampling issues are encountered, for example, in the ordered phase and not in the disordered one. As its name suggests, parallel tempering carries out in parallel simulations at consecutive values of $h$: $h_1, h_2, \cdots h_n$, with $h_1 < h_2 < \cdots < h_n$. One will sample the extended ensemble:

$$P([h_1, C_1], [h_2, C_2], \cdots, [h_n, C_n]) = \frac{W(h_1, C_1)W(h_2, C_2)\cdots W(h_n, C_n)}{\sum_{C_1, C_2, \cdots, C_n} W(h_1, C_1)W(h_2, C_2)\cdots W(h_n, C_n)}, \quad (38)$$

where $W(h, C)$ corresponds to the weight for a given value of $h$ and configuration C. Clearly, one can sample $P([h_1, C_1], [h_2, C_2], \cdots, [h_n, C_n])$ by carrying out $n$ independent runs. However, parallel tempering includes the following exchange step:

$$[h_1, C_1], \cdots, [h_i, C_i], [h_{i+1}, C_{i+1}], \cdots, [h_n, C_n] \rightarrow$$
$$[h_1, C_1], \cdots, [h_i, C_{i+1}], [h_{i+1}, C_i], \cdots, [h_n, C_n] \quad (39)$$

which, for a symmetric proposal matrix, will be accepted with probability

$$\min\left(1, \frac{W(h_i, C_{i+1})W(h_{i+1}, C_i)}{W(h_i, C_i)W(h_{i+1}, C_{i+1})}\right). \quad (40)$$

In this way a configuration can meander in parameter space $h$ and explore regions where ergodicity is not an issue. In the context of spin-glasses, a low temperature configuration, stuck in a local minima, can heat up, overcome the potential barrier and then cool down again.

A judicious choice of the values $h_i$ is important to obtain a good acceptance rate for the exchange step. With $W(h, C) = e^{-S(h, C)}$, the distribution of the action $S$ reads:

$$\mathcal{P}(h, S) = \sum_C P(h, C)\delta(S(h, C) - S). \quad (41)$$

A given exchange step can only be accepted if the distributions $\mathcal{P}(h, S)$ and $\mathcal{P}(h + \Delta h, S)$ overlap. For $\langle S \rangle_h < \langle S \rangle_{h+\Delta h}$ one can formulate this requirement as:

$$\langle S \rangle_h + \langle \Delta S \rangle_h \simeq \langle S \rangle_{h+\Delta h} - \langle \Delta S \rangle_{h+\Delta h}, \text{ with } \langle \Delta S \rangle_h = \sqrt{\langle (S - \langle S \rangle_h)^2 \rangle_h}. \tag{42}$$

Assuming $\langle \Delta S \rangle_{h+\Delta h} \simeq \langle \Delta S \rangle_h$ and expanding in $\Delta h$ one obtains:

$$\Delta h \simeq \frac{2 \langle \Delta S \rangle_h}{\partial \langle S \rangle_h / \partial h}. \tag{43}$$

The above equation becomes transparent for classical systems with $S(h, C) = hH(C)$. In this case, the above equation reads:

$$\Delta h \simeq 2h \frac{\sqrt{c}}{c + h \langle H \rangle_h}, \text{ with } c = h^2 \langle (H - \langle H \rangle_h)^2 \rangle_h. \tag{44}$$

Several comments are in order:

i) Let us identify $h$ with the inverse temperature such that $c$ corresponds to the specific heat. This quantity is extensive, as well as the energy, such that $\Delta h \simeq 1/\sqrt{N}$ where $N$ is the system size.

ii) Near a phase transition the specific heat can diverge, and $h$ must be chosen with particular care.

iii) Since the action is formulation dependent, also the acceptance rate of the exchange move equally depend upon the formulation.

The quantum Monte Carlo code in the ALF project carries out parallel-tempering runs when the script `configure.sh` is called with the argument `Tempering` before compilation, see Sec. 6.2.

### 2.2.6 Langevin dynamics

For models that include continuous real fields $\boldsymbol{s} \equiv \{s_{k,\tau}\}$ there is the option of using Langevin dynamics for the updating scheme, by setting the variable `Langevin` to `.true.`. This corresponds to a stochastic differential equation for the fields. They acquire a discrete Langevin time $t_l$ with step width $\delta t_l$ and satisfy the stochastic differential equation

$$\boldsymbol{s}(t_l + \delta t_l) = \boldsymbol{s}(t_l) - Q \frac{\partial S(\boldsymbol{s}(t_l))}{\partial \boldsymbol{s}(t_l)} \delta t_l + \sqrt{2 \delta t_l Q} \boldsymbol{\eta}(t_l). \tag{45}$$

Here, $\boldsymbol{\eta}(t_l)$ are independent Gaussian stochastic variables satisfying:

$$\langle \eta_{k,\tau}(t_l) \rangle_\eta = 0 \quad \text{and} \quad \langle \eta_{k,\tau}(t_l) \eta_{k',\tau'}(t_l') \rangle_\eta = \delta_{k,k'} \delta_{\tau,\tau'} \delta_{t_l,t_l'}, \tag{46}$$

$S(\boldsymbol{s}(t_l))$ is an arbitrary real action and $Q$ is an arbitrary positive definite matrix. By default $Q$ is equal to the identity matrix, but a proper choice can help accelerate the update scheme, as we discuss below. We refer the reader to Ref. [117] for an in-depth introduction to stochastic differential equations. To see that the above indeed produces the desired probability distribution in the long Langevin time limit, we can transform the Langevin equation into the

corresponding Fokker-Plank one. Let $P(\boldsymbol{s}, t_l)$ be the distribution of fields at Langevin time $t_l$. Then,

$$P(\boldsymbol{s}, t_l + \delta t_l) = \int D\boldsymbol{s}' P(\boldsymbol{s}', t_l) \left\langle \delta \left( \boldsymbol{s} - \left[ \boldsymbol{s}' - Q \frac{\partial S(\boldsymbol{s}')}{\partial \boldsymbol{s}'} \delta t_l + \sqrt{2 \delta t_l Q} \boldsymbol{\eta}(t_l) \right] \right) \right\rangle_\eta, \qquad (47)$$

where $\delta$ corresponds to the $L_{\text{trotter}} M_I$ dimensional Dirac $\delta$-function. Taylor expanding up to order $\delta t_l$ and averaging over the stochastic variable yields:

$$P(\boldsymbol{s}, t_l + \delta t_l) = \int D\boldsymbol{s}' P(\boldsymbol{s}', t_l) \left( \delta \left( \boldsymbol{s}' - \boldsymbol{s} \right) - \frac{\partial}{\partial \boldsymbol{s}'} \delta \left( \boldsymbol{s}' - \boldsymbol{s} \right) Q \frac{\partial S(\boldsymbol{s}')}{\partial \boldsymbol{s}'} \delta t_l \right.$$
$$\left. + \frac{\partial}{\partial \boldsymbol{s}'} Q \frac{\partial}{\partial \boldsymbol{s}'} \delta \left( \boldsymbol{s}' - \boldsymbol{s} \right) \delta t_l \right) + \mathcal{O} \left( \delta t_l^2 \right). \quad (48)$$

Partial integration and taking the limit of infinitesimal time steps gives the Fokker-Plank equation

$$\frac{\partial}{\partial t_l} P(\boldsymbol{s}, t_l) = \frac{\partial}{\partial \boldsymbol{s}} \left( P(\boldsymbol{s}, t_l) Q \frac{\partial S(\boldsymbol{s})}{\partial \boldsymbol{s}} + Q \frac{\partial P(\boldsymbol{s}, t_l)}{\partial \boldsymbol{s}} \right). \qquad (49)$$

The stationary, $\frac{\partial}{\partial t_l} P(\boldsymbol{s}, t_l) = 0$, normalizable, solution to the above equation corresponds to the desired probability distribution:

$$P(\boldsymbol{s}) = \frac{e^{-S(\boldsymbol{s})}}{\int D\boldsymbol{s} e^{-S(\boldsymbol{s})}}. \qquad (50)$$

Taking into account a potential negative sign problem, the action for our general model reads:

$$\overline{S}(C) = -\ln \left| \text{Re} \left\{ e^{-S(C)} \right\} \right| \qquad (51)$$

where $S(C)$ is defined in Eq. (17). Hence,

$$\frac{\partial \overline{S}(C)}{\partial s_{k,\tau}} = \frac{1}{\text{Re} \left\{ e^{i\phi(C)} \right\}} \text{Re} \left\{ e^{i\phi(C)} \frac{\partial S(C)}{\partial s_{k,\tau}} \right\} \qquad (52)$$

with

$$e^{i\phi(C)} = \frac{e^{-S(C)}}{|e^{-S(C)}|} \qquad (53)$$

corresponding to the variable `PHASE` in the ALF-package.

Therefore, to formulate the Langevin dynamics we need to estimate the forces:

$$\frac{\partial S(C)}{\partial s_{k,\tau}} = \frac{\partial S_0(C)}{\partial s_{k,\tau}} + \frac{\partial S^F(C)}{\partial s_{k,\tau}}, \qquad (54)$$

with the fermionic part of the action being

$$S^F(C) = -\ln \left\{ \left( \prod_{k=1}^{M_V} \prod_{\tau=1}^{L_{\text{Trotter}}} \gamma_{k,\tau} \right) e^{N_{\text{col}} \sum_{s=1}^{N_{\text{fl}}} \sum_{k=1}^{M_V} \sum_{\tau=1}^{L_{\text{Trotter}}} \sqrt{-\Delta \tau U_k} \alpha_{k,s} \eta_{k,\tau}} \right.$$

$$\left. \times \prod_{s=1}^{N_{\text{fl}}} \left[ \det \left( \mathbb{1} + \prod_{\tau=1}^{L_{\text{Trotter}}} \prod_{k=1}^{M_V} e^{\sqrt{-\Delta \tau U_k} \eta_{k,\tau} \boldsymbol{V}^{(ks)}} \prod_{k=1}^{M_I} e^{-\Delta \tau s_{k,\tau} \boldsymbol{I}^{(ks)}} \prod_{k=1}^{M_T} e^{-\Delta \tau \boldsymbol{T}^{(ks)}} \right) \right]^{N_{\text{col}}} \right\}. \quad (55)$$

The forces must be bounded for Langevin dynamics to work well. If this condition is violated the results produced by the code are *not reliable*.

One possible source of divergence is the determinant in the fermionic action. Zeros lead to unbounded forces and, in order to mitigate this problem, we adopt a variable time step. The user provides an upper bound to the fermion force, `Max_Force` and, if the maximal force in a configuration, `Max_Force_Conf`, is larger than `Max_Force`, then the time step is rescaled as

$$\tilde{\delta t}_l = \frac{\texttt{Max\_Force}}{\texttt{Max\_Force\_Conf}} * \delta t_l. \tag{56}$$

With the adaptive time step, averages are computed as:

$$\langle \hat{O} \rangle = \frac{\sum_n (\tilde{\delta t}_l)_n \operatorname{sgn}(C_n) \frac{e^{-S(C_n)}}{\operatorname{Re}[e^{-S(C_n)}]} \langle\langle \hat{O} \rangle\rangle_{(C_n)}}{\sum_n (\tilde{\delta t}_l)_n \operatorname{sgn}(C_n)}. \tag{57}$$

where $\operatorname{sgn}(C_n)$ is defined in Eq. (26). In this context the adaptive time step corresponds to the variable `Mc_step_weight` required for the measurement routines (see Sec. 5.4).

A possible way to reduce autocorrelation times is to employ Fourier acceleration [118,119]. As we see from Eq. (50), the choice of the matrix $Q$ does not alter the probability distribution obtained from the Langevin equation. The main idea of Fourier acceleration is to exploit this freedom and use $Q$ to enhance (reduce) the Langevin time step $\delta t_l$ of slow (fast) modes of the fields $s$ [120]. The modified Langevin equation reads:

$$s(t_l + \delta t_l) = s(t_l) - \hat{F}^{-1} \left[ Q\hat{F} \left[ \frac{\partial S(s(t_l))}{\partial s(t_l)} \right] \delta t_l + \sqrt{2\delta t_l Q} \hat{F} \left[ \eta(t_l) \right] \right], \tag{58}$$

with $\hat{F}$ being a transformation to independent modes of the field. This generically corresponds to a Fourier transform, thus the notation. Currently, Fourier acceleration is not implemented in ALF, but can be included by the user.

In order to use Langevin dynamics the user also has to provide the Langevin time step `Delta_t_Langevin_HMC`, the maximal force `Max_Force`, set `Global_update_scheme = Langevin` in the `parameter` file. Furthermore, the forces $\frac{\partial S_0(C)}{\partial s_{k,\tau}}$ are to be specified in the routine `Ham_Langevin_HMC_S0` of the Hamiltonian files. The Langevin update for a general Hamiltonian is carried out in the module `Langevin_HMC_mod.F90`. In particular the fermion forces,

$$\frac{\partial S^F(C)}{\partial s_{k,\tau}} = \Delta\tau N_{\mathrm{col}} \sum_{s=1}^{N_{\mathrm{fl}}} \operatorname{Tr} \left[ \boldsymbol{I}^{(ks)} \left( \mathbb{1} - \boldsymbol{G}^{(s)}(k,\tau) \right) \right], \tag{59}$$

are computed in this module. In the above, we introduce a Green function that depends on the time slice $\tau$ and the interaction term $k$ to which the corresponding field $s_{k,\tau}$ belongs:

$$G_{x,y}^{(s)}(k,\tau) = \frac{\operatorname{Tr} \left[ \hat{U}_{(s)}^{<}(k,\tau) \hat{c}_{x,s} \hat{c}_{y,s}^{\dagger} \hat{U}_{(s)}^{>}(k,\tau) \right]}{\operatorname{Tr} \left[ \hat{U}_{(s)}^{<}(k,\tau) \hat{U}_{(s)}^{>}(k,\tau) \right]}, \tag{60}$$

where the following definitions are used

$$\hat{U}_{(s)}^{<}(k',\tau') = \prod_{\tau=\tau'+1}^{L_{\text{Trotter}}} \left(\hat{U}_{(s)}(\tau)\right) \prod_{k=1}^{M_V} e^{\sqrt{-\Delta\tau U_k}\eta_{k,\tau'}\hat{\boldsymbol{c}}_s^\dagger \boldsymbol{V}^{(ks)}\hat{\boldsymbol{c}}_s} \prod_{k=k'+1}^{M_I} e^{-\Delta\tau s_{k,\tau'}\hat{\boldsymbol{c}}_s^\dagger \boldsymbol{I}^{(ks)}\hat{\boldsymbol{c}}_s}, \tag{61}$$

$$\hat{U}_{(s)}^{>}(k',\tau') = \prod_{k=1}^{k'} e^{-\Delta\tau s_{k,\tau'}\hat{\boldsymbol{c}}_s^\dagger \boldsymbol{I}^{(ks)}\hat{\boldsymbol{c}}_s} \prod_{k=1}^{M_T} e^{-\Delta\tau \hat{\boldsymbol{c}}_s^\dagger \boldsymbol{T}^{(ks)}\hat{\boldsymbol{c}}_s} \prod_{\tau=1}^{\tau'-1} \left(\hat{U}_{(s)}(\tau)\right), \tag{62}$$

$$\hat{U}_{(s)}(\tau) = \prod_{k=1}^{M_V} e^{\sqrt{-\Delta\tau U_k}\eta_{k,\tau}\hat{\boldsymbol{c}}_s^\dagger \boldsymbol{V}^{(ks)}\hat{\boldsymbol{c}}_s} \prod_{k=1}^{M_I} e^{-\Delta\tau s_{k,\tau}\hat{\boldsymbol{c}}_s^\dagger \boldsymbol{I}^{(ks)}\hat{\boldsymbol{c}}_s} \prod_{k=1}^{M_T} e^{-\Delta\tau \hat{\boldsymbol{c}}_s^\dagger \boldsymbol{T}^{(ks)}\hat{\boldsymbol{c}}_s}. \tag{63}$$

The vector $\hat{\boldsymbol{c}}_s^\dagger$ contains all fermionic operators $\hat{c}_{x,s}^\dagger$ of flavor $s$.

During each Langevin step, all fields are updated and the Langevin time is incremented by $\tilde{\delta t}_l$. At the end of a run, the mean and maximal forces encountered during the run are printed out in the info file.

The great advantage of the Langevin updating scheme is the absence of update rejection, meaning that all fields are updated at each step. As mentioned above, the price we pay for using Langevin dynamics is ensuring that forces show no singularities. Two other potential issues should be highlighted:

- Langevin dynamics is carried out at a finite Langevin time step, thereby introducing a further source of systematic error.

- The factor $\sqrt{2\delta t_l}$ multiplying the stochastic variable makes the noise dominant on short time scales. On these time scales Langevin dynamics essentially corresponds to a random walk. This has the advantage of allowing one to circumvent potential barriers, but may render the updating scheme less efficient than the hybrid molecular dynamics approach.

**Example - Hubbard chain at half-filling**

Let us consider a 6-site Hubbard chain at half-filling with $U/t = 4$ and $\beta t = 4$. The Hubbard interaction can be decoupled using a continuous HS transformation, where we introduce a real auxiliary field $s_{i,\tau}$ for every lattice site $i$ and time slice $\tau$. When the HS fields are coupled to the $z$-component of the magnetization (see Sec. 9.1), the partition function can be written as

$$Z = \int \left(\prod_{\tau=1}^{L_{\text{Trotter}}} \prod_{i=1}^{N_{\text{unit-cell}}} \frac{ds_{i,\tau}}{\sqrt{2\pi}} e^{-\frac{1}{2}s_{i,\tau}^2}\right)$$
$$\times \prod_{s=\uparrow,\downarrow} \det\left(\mathbb{1} + \prod_{\tau=1}^{L_{\text{Trotter}}} \prod_{i=1}^{N_{\text{unit-cell}}} \left(e^{-\sqrt{\Delta\tau U}s_{i,\tau}\boldsymbol{V}^{(is)}}\right)e^{-\Delta\tau\boldsymbol{T}}\right) + \mathcal{O}(\Delta\tau^2). \tag{64}$$

The flavor-dependent interaction matrices have only one non-vanishing entry each:

$$V_{x,y}^{(i,s=\uparrow)} = \delta_{x,y}\delta_{x,i} \quad \text{and} \quad V_{x,y}^{(i,s=\downarrow)} = -\delta_{x,y}\delta_{x,i}.$$

The forces of the Hubbard model are given by:

$$\frac{\partial S(C)}{\partial s_{i,\tau}} = s_{i,\tau} - \sqrt{\Delta\tau U} \sum_{s=\uparrow,\downarrow} \text{Tr}\left[\boldsymbol{V}^{(is)}\left(\mathbb{1} - \boldsymbol{G}^{(s)}(i,\tau)\right)\right], \tag{65}$$

where the Green function is defined by Eq. (60) with

$$\hat{U}^{<}_{(s)}(i',\tau') = \prod_{\tau=\tau'+1}^{L_{\text{Trotter}}} \left(\hat{U}_{(s)}(\tau)\right) \prod_{i=i'+1}^{N_{\text{unit-cell}}} e^{-\sqrt{\Delta\tau U}s_{i,\tau'}\hat{c}_s^\dagger V^{(is)}\hat{c}_s}, \tag{66}$$

$$\hat{U}^{>}_{(s)}(i',\tau') = \prod_{i=1}^{i'} \left(e^{-\sqrt{\Delta\tau U}s_{i,\tau'}\hat{c}_s^\dagger V^{(is)}\hat{c}_s}\right) e^{-\Delta\tau \hat{c}_s^\dagger T\hat{c}_s} \prod_{\tau=1}^{\tau'-1} \left(\hat{U}_{(s)}(\tau)\right), \tag{67}$$

$$\hat{U}_{(s)}(\tau) = \prod_{i=1}^{N_{\text{unit-cell}}} \left(e^{-\sqrt{\Delta\tau U}s_{i,\tau}\hat{c}_s^\dagger V^{(is)}\hat{c}_s}\right) e^{-\Delta\tau \hat{c}_s^\dagger T\hat{c}_s}. \tag{68}$$

One can show that for periodic boundary conditions the forces are not bounded and to make sure that the program does not crash we set `Max_Force = 1.5`.

The results are: the reference, discrete-variable code gives

$$\langle \hat{H} \rangle = -3.4684 \pm 0.0007, \tag{69}$$

while the Langevin code at $\delta t_l = 0.001$ yields

$$\langle \hat{H} \rangle = -3.457 \pm 0.010 \tag{70}$$

and at $\delta t_l = 0.01$

$$\langle \hat{H} \rangle = -3.495 \pm 0.007. \tag{71}$$

At $\delta t_l = 0.001$ the maximal force that occurred during the run was 112, whereas at $\delta t_l = 0.01$ it grew to 524. In both cases the average force was given by 0.45. For larger values of $\delta t_l$ the maximal force grows and the fluctuations on the energy become larger (for instance, $\langle \hat{H} \rangle = -3.718439 \pm 0.206469$ at $\delta t_l = 0.02$; for this parameter set the maximal force we encountered during the run was of 1658).

Controlling Langevin dynamics when the action has logarithmic divergences is a challenge, and it is not a given that the results are satisfactory. For our specific problem we can solve this issue by considering open boundary conditions. Following an argument put forward in [89], we can show, using world lines, that the determinant is always positive. In this case the action does not have logarithmic divergences and the Langevin dynamics works beautifully well, see Fig. 1.

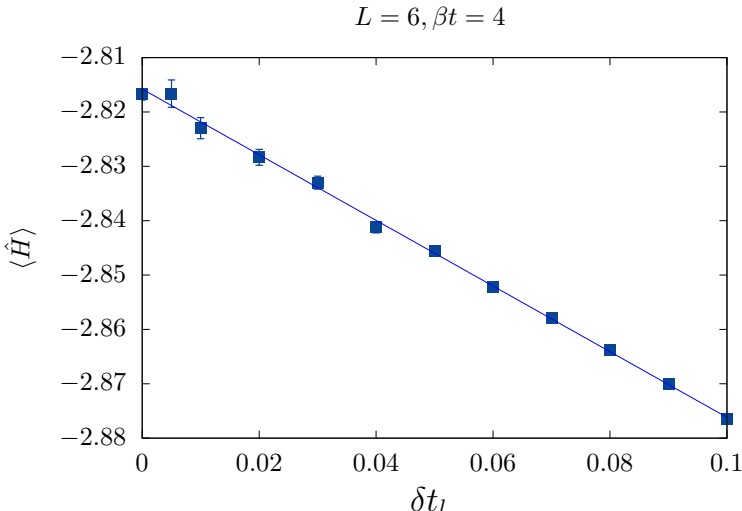

Figure 1: Total energy for the 6-site Hubbard chain at $U/t = 4$, $\beta t = 4$ and with open boundary conditions. For this system it can be shown that the determinant is always positive, so that no singularities occur in the action and, consequently, the Langevin dynamics works very well. The reference data point at $\delta t_l = 0$ comes from the discrete field code for the field coupled to the z-component of the magnetization and reads $-2.8169 \pm 0.0013$, while the extrapolated value is $-2.8176 \pm 0.0010$. Throughout the runs the maximal force remained bellow the threshold of 1.5. The displayed data has been produced by the pyALF script `Langevin.py`.

## 2.3 The Trotter error and checkerboard decomposition

### 2.3.1 Asymmetric Trotter decomposition

In practice, many applications are carried out at finite imaginary time steps, and it is important to understand the consequences of the Trotter error. How does it scale with system size and what symmetries does it break? In particular, when investigating a critical point, one should determine whether the potential symmetry breaking associated with the Trotter decomposition generates relevant operators.

To best describe the workings of the ALF code, we divide the Hamiltonian into hopping terms $\hat{\mathcal{H}}_T$ and interaction terms $\hat{\mathcal{H}}_V + \hat{\mathcal{H}}_I + \hat{\mathcal{H}}_{0,I}$. Let

$$\hat{\mathcal{H}}_T = \sum_{i=1}^{N_T} \sum_{k \in \mathcal{S}_i^T} \hat{T}^{(k)} \equiv \sum_{i=1}^{N_T} \hat{T}_i. \tag{72}$$

Here the decomposition follows the rule that if $k$ and $k'$ belong to the same set $\mathcal{S}_i^T$ then $\left[ \hat{T}^{(k)}, \hat{T}^{(k')} \right] = 0$. An important case to consider is that of the checkerboard decomposition. For the square lattice we can decouple the nearest neighbor hopping into $N_T = 4$ groups, each consisting of two site hopping processes. This type of checkerboard decomposition is

activated for a set of predefined lattices by setting the flag `Checkerboard` to `.true.`. We will carry out the same separation for the interaction:

$$\hat{\mathcal{H}}_V + \hat{\mathcal{H}}_I + \hat{\mathcal{H}}_{0,I} = \sum_{i=1}^{N_I} \hat{O}_i, \tag{73}$$

where each $\hat{O}_i$ contains a set of commuting terms. For instance, for the Hubbard model, the above reduces to $U \sum_{\boldsymbol{i}} \hat{n}_{\boldsymbol{i},\uparrow} \hat{n}_{\boldsymbol{i},\downarrow}$ such that $N_I = 1$ and $\hat{O}_1 = U \sum_{\boldsymbol{i}} \hat{n}_{\boldsymbol{i},\uparrow} \hat{n}_{\boldsymbol{i},\downarrow}$.

The default Trotter decomposition in the ALF code is based on the equation:

$$e^{-\Delta\tau(\hat{A}+\hat{B})} = e^{-\Delta\tau\hat{A}} e^{-\Delta\tau\hat{B}} + \frac{\Delta\tau^2}{2} \left[\hat{B}, \hat{A}\right] + \mathcal{O}\left(\Delta\tau^3\right). \tag{74}$$

Using iteratively the above the single time step is given by:

$$
\begin{aligned}
e^{-\Delta\tau\mathcal{H}} = &\prod_{i=1}^{N_O} e^{-\Delta\tau\hat{O}_i} \prod_{j=1}^{N_T} e^{-\Delta\tau\hat{T}_j} \\
&+ \frac{\Delta\tau^2}{2} \underbrace{\left( \sum_{i=1}^{N_O}\sum_{j=1}^{N_T} \left[\hat{T}_j, \hat{O}_i\right] + \sum_{j'}^{N_T-1} \left[\hat{T}_{j'}, \hat{T}_{j'}^{>}\right] + \sum_{i'=1}^{N_O-1} \left[\hat{O}_{i'}, \hat{O}_{i'}^{>}\right] \right)}_{\equiv \Delta\tau\hat{\lambda}_1} + \mathcal{O}\left(\Delta\tau^3\right).
\end{aligned} \tag{75}
$$

In the above, we have introduced the shorthand notation

$$\hat{T}_n^{>} = \sum_{j=n+1}^{N_T} \hat{T}_j \quad \text{and} \quad \hat{O}_n^{>} = \sum_{j=n+1}^{N_O} \hat{O}_j. \tag{76}$$

The full propagation then reads

$$
\begin{aligned}
\hat{U}_{\text{Approx}} &= \left( \prod_{i=1}^{N_O} e^{-\Delta\tau\hat{O}_i} \prod_{j=1}^{N_T} e^{-\Delta\tau\hat{T}_j} \right)^{L_{\text{Trotter}}} = e^{-\beta(\hat{H}+\hat{\lambda}_1)} + \mathcal{O}\left(\Delta\tau^2\right) \\
&= e^{-\beta\hat{H}} - \int_0^\beta d\tau\, e^{-(\beta-\tau)\hat{H}} \hat{\lambda}_1 e^{-\tau\hat{H}} + \mathcal{O}(\Delta\tau^2).
\end{aligned} \tag{77}
$$

The last step follows from time-dependent perturbation theory. The following comments are in order:

- The error is anti-Hermitian since $\hat{\lambda}_1^\dagger = -\hat{\lambda}_1$. As a consequence, if all the operators as well as the quantity being measured are simultaneously real representable, then the prefactor of the linear in $\Delta\tau$ error vanishes since it ultimately corresponds to computing the trace of an anti-symmetric matrix. This *lucky* cancellation was put forward in Ref. [98]. Hence, under this assumption – which is certainly valid for the Hubbard model considered in Fig. 2 – the systematic error is of order $\Delta\tau^2$.

- The biggest drawback of the above decomposition is that the imaginary-time propagation is not Hermitian. This can lead to acausal features in imaginary-time correlation functions [121]. To be more precise, the eigenvalues of $H_{\text{Approx}} = -\frac{1}{\beta} \log U_{\text{Approx}}$ need

not be real and thus imaginary-time displaced correlation functions may oscillate as a function of imaginary time. This is shown in Fig. 2(a) that plots the absolute value of local time-displaced Green function for the Honeycomb lattice at $U/t = 2$. Sign changes of this quantity involve zeros that, on the considered log-scale, correspond to negative divergences. As detailed in [109], using the non-symmetric Trotter decomposition leads to an additional non-hermitian second-order error in the measurement of observables $O$ that is proportional to $[T, [T, O]]$. As we see next, these issues can be solved by considering a symmetric Trotter decomposition.

### 2.3.2   Symmetric Trotter decomposition

To address the issue described above, the ALF package provides the possibility of using a symmetric Trotter decomposition,

$$e^{-\Delta\tau(\hat{A}+\hat{B})} = e^{-\Delta\tau\hat{A}/2}e^{-\Delta\tau\hat{B}}e^{-\Delta\tau\hat{A}/2} + \frac{\Delta\tau^3}{12}\left[2\hat{A} + \hat{B}, \left[\hat{B}, \hat{A}\right]\right] + \mathcal{O}\left(\Delta\tau^5\right), \qquad (78)$$

by setting the `Symm` flag to `.true.`. Before we apply the expression above to a time step, let us write

$$e^{-\Delta\tau\mathcal{H}} = e^{-\frac{\Delta\tau}{2}\sum_{j=1}^{N_T}\hat{T}_j}e^{-\Delta\tau\sum_{i=1}^{N_I}\hat{O}_i}e^{-\frac{\Delta\tau}{2}\sum_{j=1}^{N_T}\hat{T}_j} + \underbrace{\frac{\Delta\tau^3}{12}\left[2\hat{T}_0^> + \hat{O}_0^>, \left[\hat{O}_0^>, \hat{T}_0^>\right]\right]}_{\equiv \Delta\tau\hat{\lambda}_{TO}} + \mathcal{O}\left(\Delta\tau^5\right).$$

$$(79)$$

Then,

$$e^{-\Delta\tau\sum_i^{N_I}\hat{O}_i} = \left(\prod_{i=1}^{N_O-1}e^{-\frac{\Delta\tau}{2}\hat{O}_i}\right)e^{-\Delta\tau\hat{O}_{N_O}}\left(\prod_{i=N_O-1}^{1}e^{-\frac{\Delta\tau}{2}\hat{O}_i}\right)$$

$$+ \underbrace{\frac{\Delta\tau^3}{12}\sum_{i=1}^{N_0-1}\left[2\hat{O}_i + \hat{O}_i^>, \left[\hat{O}_i^>, \hat{O}_i\right]\right]}_{\equiv \Delta\tau\hat{\lambda}_O} + \mathcal{O}\left(\Delta\tau^5\right), \quad (80)$$

$$e^{-\frac{\Delta\tau}{2}\sum_j^{N_T}\hat{T}_j} = \left(\prod_{j=1}^{N_T-1}e^{-\frac{\Delta\tau}{4}\hat{T}_j}\right)e^{-\frac{\Delta\tau}{2}\hat{T}_{N_T}}\left(\prod_{j=N_T-1}^{1}e^{-\frac{\Delta\tau}{4}\hat{T}_j}\right)$$

$$+ \underbrace{\frac{\Delta\tau^3}{96}\sum_{j=1}^{N_T-1}\left[2\hat{T}_j + \hat{T}_j^>, \left[\hat{T}_j^>, \hat{T}_j\right]\right]}_{\equiv \Delta\tau\hat{\lambda}_T} + \mathcal{O}\left(\Delta\tau^5\right) \quad (81)$$

and we can derive a closed equation for the free energy density:

$$
f_{\text{Approx}} = -\frac{1}{\beta V} \log \text{Tr} \left[ \left( \prod_{j=1}^{N_T-1} e^{-\frac{\Delta\tau}{4}\hat{T}_j} \right) e^{-\frac{\Delta\tau}{2}\hat{T}_{N_T}} \left( \prod_{j=N_T-1}^{1} e^{-\frac{\Delta\tau}{4}\hat{T}_j} \right) \times \right.
$$
$$
\left( \prod_{i=1}^{N_O-1} e^{-\frac{\Delta\tau}{2}\hat{O}_i} \right) e^{-\Delta\tau\hat{O}_{N_O}} \left( \prod_{i=N_O-1}^{1} e^{-\frac{\Delta\tau}{2}\hat{O}_i} \right) \times
$$
$$
\left. \left( \prod_{j=1}^{N_T-1} e^{-\frac{\Delta\tau}{4}\hat{T}_j} \right) e^{-\frac{\Delta\tau}{2}\hat{T}_{N_T}} \left( \prod_{j=N_T-1}^{1} e^{-\frac{\Delta\tau}{4}\hat{T}_j} \right) \right]^{L_{\text{Trotter}}}
$$
$$
= f - \frac{1}{V}\langle \hat{\lambda}_{TO} + \hat{\lambda}_O + 2\hat{\lambda}_T \rangle + \mathcal{O}(\Delta\tau^4). \tag{82}
$$

The following comments are in order:

- The approximate imaginary-time propagation from which the $f_{\text{Approx}}$ is derived is Hermitian. Hence no spurious effects in imaginary-time correlation functions are to be expected. This is clearly shown in Fig. 2(a).

- In Fig. 2(b) we have used the ALF-library with `Symm=.true.` with and without checkerboard decomposition. We still expect the systematic error to be of order $\Delta\tau^2$. However its prefactor is much smaller than that of the aforementioned anti-symmetric decomposition.

- We have taken the burden to evaluate explicitly the prefactor of the $\Delta\tau^2$ error on the free energy density. One can see that for Hamiltonians that are sums of local operators, the quantity $\langle \hat{\lambda}_{TO} + \hat{\lambda}_O + 2\hat{\lambda}_T \rangle$ scales as the volume $V$ of the system, such that the systematic error on the free energy density (and on correlation functions that can be computed by adding source terms) will be volume independent. For model Hamiltonians that are not sums of local terms, care must be taken. A conservative upper bound on the error is $\langle \hat{\lambda}_{TO} + \hat{\lambda}_O + 2\hat{\lambda}_T \rangle \propto \Delta\tau^2 V^3$, which means that, in order to maintain a constant systematic error for the free energy density, we have to keep $\Delta\tau V$ constant. Such a situation has been observed in Ref. [71].

Alternative symmetric second order methods as well as the issues with decompositions of higher order have been detailed in [109].

### 2.3.3 The `Symm` flag

If the `Symm` flag is set to true, then the program will automatically – for the set of predefined lattices and models – use the symmetric $\Delta\tau$ time step of the interaction and hopping terms.

To save CPU time when the `Symm` flag is on we carry out the following approximation:

$$
\left[ \left( \prod_{j=1}^{N_T-1} e^{-\frac{\Delta\tau}{4}\hat{T}_j} \right) e^{-\frac{\Delta\tau}{2}\hat{T}_{N_T}} \left( \prod_{j=N_T-1}^{1} e^{-\frac{\Delta\tau}{4}\hat{T}_j} \right) \right]^2 \simeq
$$
$$
\left( \prod_{j=1}^{N_T-1} e^{-\frac{\Delta\tau}{2}\hat{T}_j} \right) e^{-\Delta\tau\hat{T}_{N_T}} \left( \prod_{j=N_T-1}^{1} e^{-\frac{\Delta\tau}{2}\hat{T}_j} \right). \tag{83}
$$

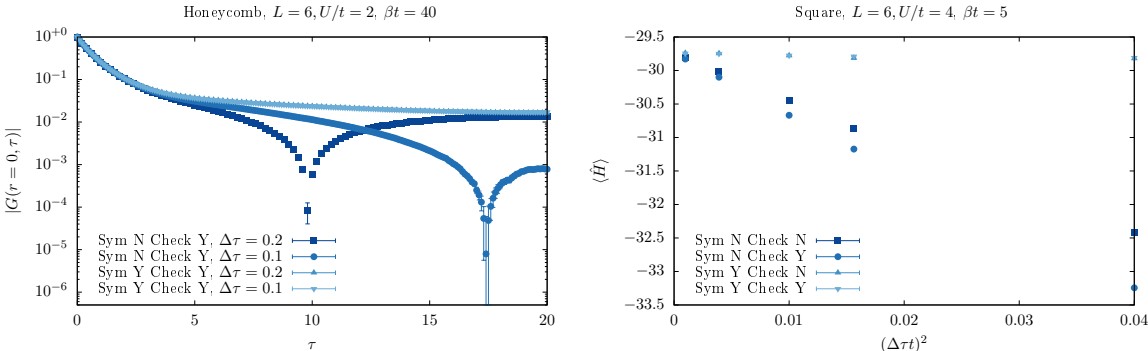

Figure 2: Analysis of Trotter systematic error. Left: We consider a $6 \times 6$ Hubbard model on the Honeycomb lattice, $U/t = 2$, half-band filling, inverse temperature $\beta t = 40$, and we have used an HS transformation that couples to the density. The figure plots the local-time displaced Green function. Right: Here we consider the $6 \times 6$ Hubbard model at $U/t = 4$, half-band filling, inverse temperature $\beta t = 5$, and we have used the HS transformation that couples to the $z$-component of spin. We provide data for the four combinations of the logical variables `Symm` and `Checkerboard`, where `Symm=.true.` (`.false.`) indicates a symmetric (asymmetric) Trotter decomposition has been used, and `Checkerboard=.true.` (`.false.`) that the checkerboard decomposition for the hopping matrix has (not) been used. The large deviations between different choices of `Symm` are here $\sim [T, [T, H]]$ as detailed in [109].

The above is consistent with the overall precision of the Trotter decomposition and more importantly conserves the Hermiticity of the propagation.

## 2.4 Stabilization - a peculiarity of the BSS algorithm

From the partition function in Eq. (17) it can be seen that, for the calculation of the Monte Carlo weight and of the observables, a long product of matrix exponentials has to be formed. In addition to that, we need to be able to extract the single-particle Green function for a given flavor index at, say, time slice $\tau = 0$. As mentioned above (cf. Eq. (21)), this quantity is given by:

$$\boldsymbol{G} = \left( \mathbb{1} + \prod_{\tau=1}^{L_{\text{Trotter}}} \boldsymbol{B}_\tau \right)^{-1}, \tag{84}$$

which can be recast as the more familiar linear algebra problem of finding a solution for the linear system

$$\left( \mathbb{1} + \prod_{\tau=1}^{L_{\text{Trotter}}} \boldsymbol{B}_\tau \right) x = b. \tag{85}$$

The matrices $\boldsymbol{B}_\tau \in \mathbb{C}^{n \times n}$ depend on the lattice size as well as other physical parameters that can be chosen such that a matrix norm of $\boldsymbol{B}_\tau$ can be unbound in magnitude. From standard perturbation theory for linear systems, the computed solution $\tilde{x}$ would contain a relative error

$$\frac{|\tilde{x} - x|}{|x|} = \mathcal{O}\left( \epsilon \kappa_p \left( \mathbb{1} + \prod_{\tau=1}^{L_{\text{Trotter}}} \boldsymbol{B}_\tau \right) \right), \tag{86}$$

where $\epsilon$ denotes the machine precision, which is $2^{-53}$ for IEEE double-precision numbers, and $\kappa_p(\boldsymbol{M})$ is the condition number of the matrix $\boldsymbol{M}$ with respect to the matrix $p$-norm. Due to $\prod_\tau \boldsymbol{B}_\tau$ containing exponentially large and small scales, as can be seen in Eq. (17), a straightforward inversion is completely ill-suited, in that its condition number would grow exponentially with increasing inverse temperature, rendering the computed solution $\tilde{x}$ meaningless.

In order to circumvent this, more sophisticated methods have to be employed. In the realm of the BSS algorithm there has been a long history [4, 93, 122–125] of using various matrix factorization techniques. The predominant techniques are either based on the singular value decomposition (SVD) or on techniques using the QR decomposition. The default stabilization strategy in the auxiliary-field QMC implementation of the ALF package is to form a product of QR-decompositions, which is proven to be weakly backwards stable [124]. While algorithms using the SVD can provide higher stability, though at a higher cost, we note that great care has to be taken in the choice of the algorithm, which has to achieve a high relative accuracy [126, 127].

As a first step we assume that, for a given integer `NWrap`, the multiplication of `NWrap` $\boldsymbol{B}$ matrices has an acceptable condition number and, for simplicity, that $L_{\text{Trotter}}$ is divisible by `NWrap`. We can then write:

$$\boldsymbol{G} = \left( \mathbb{1} + \prod_{i=1}^{\frac{L_{\text{Trotter}}}{\text{NWrap}}} \underbrace{\prod_{\tau=1}^{\text{NWrap}} \boldsymbol{B}_{(i-1)\cdot \text{NWrap}+\tau}}_{\equiv \mathcal{B}_i} \right)^{-1}. \tag{87}$$

The key idea is to efficiently separate the scales of a matrix from the orthogonal part of a matrix. This can be achieved with the QR decomposition of a matrix $\boldsymbol{A}$ in the form $\boldsymbol{A}_i = \boldsymbol{Q}_i \boldsymbol{R}_i$. The matrix $\boldsymbol{Q}_i$ is unitary and hence in the usual 2-norm it satisfies $\kappa_2(\boldsymbol{Q}_i) = 1$. To get a handle on the condition number of $\boldsymbol{R}_i$ we consider the diagonal matrix

$$(\boldsymbol{D}_i)_{n,n} = |(\boldsymbol{R}_i)_{n,n}| \tag{88}$$

and set $\tilde{\boldsymbol{R}}_i = \boldsymbol{D}_i^{-1} \boldsymbol{R}_i$, which gives the decomposition

$$\boldsymbol{A}_i = \boldsymbol{Q}_i \boldsymbol{D}_i \tilde{\boldsymbol{R}}_i. \tag{89}$$

The matrix $\boldsymbol{D}_i$ now contains the row norms of the original $\boldsymbol{R}_i$ matrix and hence attempts to separate off the total scales of the problem from $\boldsymbol{R}_i$. This is similar in spirit to the so-called matrix equilibration which tries to improve the condition number of a matrix through suitably chosen column and row scalings. Due to a theorem by van der Sluis [128] we know that the choice in Eq. (88) is almost optimal among all diagonal matrices $\boldsymbol{D}$ from the space of diagonal matrices $\mathcal{D}$, in the sense that

$$\kappa_p((\boldsymbol{D}_i)^{-1}\boldsymbol{R}_i) \leq n^{1/p} \min_{\boldsymbol{D}\in\mathcal{D}} \kappa_p(\boldsymbol{D}^{-1}\boldsymbol{R}_i).$$

Now, given an initial decomposition $\boldsymbol{A}_{j-1} = \prod_i \mathcal{B}_i = \boldsymbol{Q}_{j-1}\boldsymbol{D}_{j-1}\boldsymbol{T}_{j-1}$, an update $\mathcal{B}_j \boldsymbol{A}_{j-1}$ is formed in the following three steps:

    1. Form $\boldsymbol{M}_j = (\mathcal{B}_j \boldsymbol{Q}_{j-1})\boldsymbol{D}_{j-1}$. Note the parentheses.

2. Do a QR decomposition of $\boldsymbol{M}_j = \boldsymbol{Q}_j \boldsymbol{D}_j \boldsymbol{R}_j$. This gives the final $\boldsymbol{Q}_j$ and $\boldsymbol{D}_j$.

3. Form the updated $\boldsymbol{T}$ matrices $\boldsymbol{T}_j = \boldsymbol{R}_j \boldsymbol{T}_{j-1}$.

This is a stable but expensive method for calculating the matrix product. Here is where `NWrap` comes into play: it specifies the number of plain multiplications performed between the QR decompositions just described, so that `NWrap` $= 1$ corresponds to always performing QR decompositions whereas larger values define longer intervals where no QR decomposition will be performed. Whenever we perform a stabilization, we compare the old result (fast updates) with the new one (recalculated from the QR stabilized matrices). The difference is documented as the stability, both for the Green function and for the sign (of the determinant) The effectiveness of the stabilization *has* to be judged for every simulation from the output file `info` (Sec. 5.7.2). For most simulations there are two values to look out for:

- `Precision Green`

- `Precision Phase`

The Green function, as well as the average phase, are usually numbers with a magnitude of $\mathcal{O}(1)$. For that reason we recommend that `NWrap` is chosen such that the mean precision is of the order of $10^{-8}$ or better (for further recommendations see Sec. 6.4). We include typical values of `Precision Phase` and of the mean and the maximal values of `Precision Green` in the example simulations discussed in Sec. 7.7.

# 3 Auxiliary Field Quantum Monte Carlo: projective algorithm

The projective approach is the method of choice if one is interested in ground-state properties. The starting point is a pair of trial wave functions, $|\Psi_{T,L/R}\rangle$, that are not orthogonal to the ground state $|\Psi_0\rangle$:

$$\langle \Psi_{T,L/R} | \Psi_0 \rangle \neq 0. \tag{90}$$

The ground-state expectation value of any observable $\hat{O}$ can then be computed by propagation along the imaginary time axis:

$$\frac{\langle \Psi_0 | \hat{O} | \Psi_0 \rangle}{\langle \Psi_0 | \Psi_0 \rangle} = \lim_{\theta \to \infty} \frac{\langle \Psi_{T,L} | e^{-\theta \hat{H}} e^{-(\beta-\tau)\hat{H}} \hat{O} e^{-\tau \hat{H}} e^{-\theta \hat{H}} | \Psi_{T,R} \rangle}{\langle \Psi_{T,L} | e^{-(2\theta+\beta)\hat{H}} | \Psi_{T,R} \rangle}, \tag{91}$$

where $\beta$ defines the imaginary time range where observables (time displaced and equal time) are measured and $\tau$ varies from 0 to $\beta$ in the calculation of time-displace observables. The simulations are carried out at large but finite values of $\theta$ so as to guarantee convergence to the ground state within the statistical uncertainty. The trial wave functions are determined up to a phase, and the program uses this gauge choice to guarantee that

$$\langle \Psi_{T,L} | \Psi_{T,R} \rangle > 0. \tag{92}$$

In order to use the projective version of the code, the model's namespace in the `parameter` file must set `projector=.true.` and specify the value of the projection parameter `Theta`, as well as the imaginary time interval `Beta` in which observables are measured.

Note that time-displaced correlation functions are computed for a $\tau$ ranging from 0 to $\beta$. The implicit assumption in this formulation is that the projection parameter `Theta` suffices to reach the ground state. Since the computational time scales linearly with `Theta` large projections parameters are computationally not expensive.

## 3.1 Specification of the trial wave function

For each flavor, one needs to specify a left and a right trial wave function. In the ALF, they are assumed to be the ground state of single-particle trial Hamiltonians $\hat{H}_{T,L/R}$ and hence correspond to a single Slater determinant each. More specifically, we consider a single-particle Hamiltonian with the same symmetries, color and flavor, as the original Hamiltonian:

$$\hat{H}_{T,L/R} = \sum_{\sigma=1}^{N_{\text{col}}} \sum_{s=1}^{N_{\text{fl}}} \sum_{x,y}^{N_{\text{dim}}} \hat{c}_{x\sigma s}^\dagger h_{xy}^{(s,L/R)} \hat{c}_{y\sigma s}. \tag{93}$$

Ordering the eigenvalues of the Hamiltonian in ascending order yields the ground state

$$|\Psi_{T,L/R}\rangle = \prod_{\sigma=1}^{N_{\text{col}}} \prod_{s=1}^{N_{\text{fl}}} \prod_{n=1}^{N_{\text{part},s}} \left( \sum_{x=1}^{N_{\text{dim}}} \hat{c}_{x\sigma s}^\dagger U_{x,n}^{(s,L/R)} \right) |0\rangle, \tag{94}$$

where

$$U^{\dagger,(s,L/R)} h^{(s,L/R)} U^{(s,L/R)} = \text{Diag}\left( \epsilon_1^{(s,L/R)}, \cdots, \epsilon_{N_{\text{dim}}}^{(s,L/R)} \right). \tag{95}$$

The trial wave function is hence completely defined by the set of orthogonal vectors $U_{x,n}^{(s,L/R)}$ for $n$ ranging from 1 to the number of particles in each flavor sector, $N_{\text{part},s}$. This information is stored in the `WaveFunction` type defined in the module `WaveFunction_mod` (see Sec. 5.5). Note that, owing to the SU($N_{\text{col}}$) symmetry, the color index is not necessary to define the trial wave function. The user will have to specify the trial wave function in the following way:

```
Do s = 1, N_fl
    Do x = 1,Ndim
        Do n = 1, N_part(s)
            WF_L(s)%P(x,n)   =   U_{x,n}^{(s,L)}
            WF_R(s)%P(x,n)   =   U_{x,n}^{(s,R)}
        Enddo
    Enddo
Enddo
```

In the above `WF_L` and `WF_R` are `WaveFunction` arrays of length $N_{\text{fl}}$. ALF comes with a set of predefined trial wave functions, see Sec. 8.5.

Generically, the unitary matrix will be generated by a diagonalization routine such that if the ground state for the given particle number is degenerate, the trial wave function has a degree of ambiguity and does not necessarily share the symmetries of the Hamiltonian $\hat{H}_{T,L/R}$. Since symmetries are the key for guaranteeing the absence of the negative sign problem, violating them in the choice of the trial wave function can very well lead to a sign problem. It is hence recommended to define the trial Hamiltonians $\hat{H}_{T,L/R}$ such that the ground state for the given particle number is non-degenerate. That can be checked using the value of `WL_L/R(s)%Degen`, which stores the energy difference between the last occupied and first unoccupied single particle state. If this value is greater than zero, then the trial wave function is non-degenerate and hence has all the symmetry properties of the trial Hamiltonians, $\hat{H}_{T,L/R}$. When the `projector` variable is set to `.true.`, this quantity is listed in the `info` file.

## 3.2 Some technical aspects of the projective code.

If one is interested solely in zero-temperature properties, the projective code offers many advantages. This comes from the related facts that the Green function matrix is a projector, and that scales can be omitted.

In the projective algorithm, it is known [6] that

$$G(x, \sigma, s, \tau | x', \sigma, s, \tau) = \left[ 1 - U_{(s)}^{>}(\tau) \left( U_{(s)}^{<}(\tau) U_{(s)}^{>}(\tau) \right)^{-1} U_{(s)}^{<}(\tau) \right]_{x,x'} \tag{96}$$

with

$$U_{(s)}^{>}(\tau) = \prod_{\tau'=1}^{\tau} \boldsymbol{B}_{\tau'}^{(s)} P^{(s),R} \quad \text{and} \quad U_{(s)}^{<}(\tau) = P^{(s),L,\dagger} \prod_{\tau'=L_{\text{Trotter}}}^{\tau+1} \boldsymbol{B}_{\tau'}^{(s)}, \tag{97}$$

where $\boldsymbol{B}_{\tau}^{(s)}$ is given by Eq. (22) and $P^{(s),L/R}$ correspond to the $N_{\text{dim}} \times N_{\text{part},s}$ submatrices of $U^{(s),L/R}$. To see that scales can be omitted, we carry out a singular value decomposition:

$$U_{(s)}^{>}(\tau) = \tilde{U}_{(s)}^{>}(\tau) d^{>} v^{>} \quad \text{and} \quad U_{(s)}^{<}(\tau) = v^{<} d^{<} \tilde{U}_{(s)}^{<}(\tau) \tag{98}$$

such that $\tilde{U}_{(s)}^{>}(\tau)$ corresponds to a set of column-wise orthogonal vectors. It can be readily seen that scales can be omitted, since

$$G(x, \sigma, s, \tau | x', \sigma, s, \tau) = \left[ 1 - \tilde{U}_{(s)}^{>}(\tau) \left( \tilde{U}_{(s)}^{<}(\tau) \tilde{U}_{(s)}^{>}(\tau) \right)^{-1} \tilde{U}_{(s)}^{<}(\tau) \right]_{x,x'}. \tag{99}$$

Hence, stabilization is never an issue for the projective code, and arbitrarily large projection parameters can be reached.

The form of the Green function matrix implies that it is a projector: $G^2 = G$. This property has been used in Ref. [129] to very efficiently compute imaginary-time-displaced correlation functions.

## 3.3 Comparison of finite and projective codes.

The finite temperature code operates in the grand canonical ensemble, whereas in the projective approach the particle number is fixed. On finite lattices, the comparison between both approaches can only be made at a temperature scale below which a finite-sized charge gap emerges. In Fig. 3 we consider a semi-metallic phase as realized by the Hubbard model on the Honeycomb lattice at $U/t = 2$. It is evident that, at a scale below which charge fluctuations are suppressed, both algorithms yield identical results.

# 4 Monte Carlo sampling

Error estimates in Monte Carlo simulations are based on the central limit theorem [131] and can be a delicate matter, especially as it requires independent measurements and a finite variance. In this section we give examples of the care that must be taken to satisfy these requirements when using a Monte Carlo code. This is part of the common lore of the field and we cover them briefly in this text. For a deeper understanding of the inherent issues of Markov-chain Monte Carlo methods we refer the reader to the pedagogical introduction in

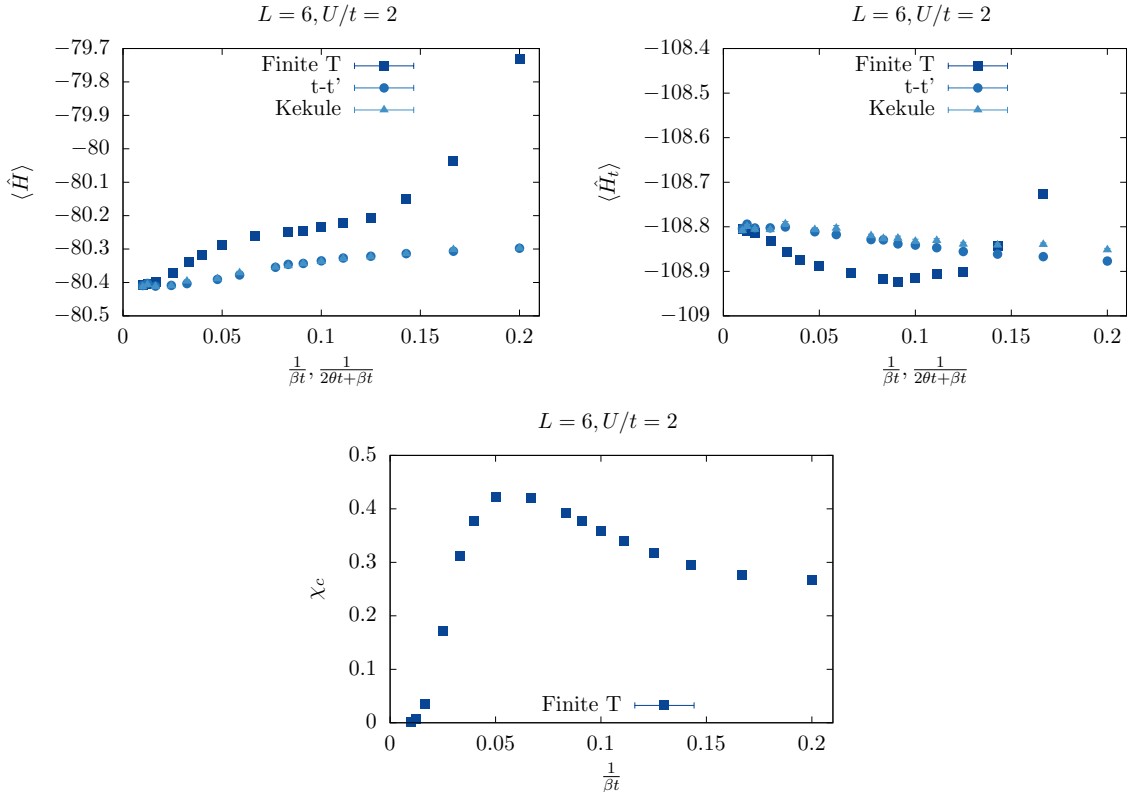

Figure 3: Comparison between the finite-temperature and projective codes for the Hubbard model on a $6 \times 6$ Honeycomb lattice at $U/t = 2$ and with periodic boundary conditions. For the projective code (blue and black symbols) $\beta t = 1$ is fixed, while $\theta$ is varied. In all cases we have $\Delta\tau t = 0.1$, no checkerboard decomposition, and a symmetric Trotter decomposition. For this lattice size and choice of boundary conditions, the non-interacting ground state is degenerate, since the Dirac points belong to the discrete set of crystal momenta. In order to generate the trial wave function we have lifted this degeneracy by either including a Kékulé mass term [46] that breaks translation symmetry (blue symbols), or by adding a next-next nearest neighbor hopping (black symbols) that breaks the symmetry nematically and shifts the Dirac points away from the zone boundary [130]. As apparent, both choices of trial wave functions yield the same answer, which compares very well with the finite temperature code at temperature scales below the finite-size charge gap.

chapter 1.3.5 of Krauth [132], the overview article of Sokal [82], the more specialized literature by Geyer [133] and chapter 6.3 of Neal [134].

In general, one distinguishes local from global updates. As the name suggest, the local update corresponds to a small change of the configuration, e.g., a single spin flip of one of the $L_{\text{Trotter}}(M_I + M_V)$ field entries (see Sec. 2.2), whereas a global update changes a significant part of the configuration. The default update scheme of the ALF implementation are local updates, such that there is a minimum number of moves required for generating an independent configuration. The associated time scale is called the autocorrelation time, $T_{\text{auto}}$, and is generically dependent upon the choice of the observables.

We call a *sweep* a sequential propagation from $\tau = 0$ to $\tau = L_{\text{Trotter}}$ and back, such that each field is visited twice in each sweep. A single sweep will generically not suffice to produce an independent configuration. In fact, the autocorrelation time $T_{\text{auto}}$ characterizes the required time scale to generate independent values of $\langle\langle \hat{O} \rangle\rangle_C$ for the observable $O$. This has several consequences for the Monte Carlo simulation:

- First of all, we start from a randomly chosen field configuration, such that one has to invest a time of *at least* one $T_{\text{auto}}$, but typically many more, in order to generate relevant, equilibrated configurations before reliable measurements are possible. This phase of the simulation is known as the warm-up or burn-in phase. In order to keep the code as flexible as possible (as different simulations might have different autocorrelation times), measurements are taken from the very beginning and, in the analysis phase, the parameter `n_skip` controls the number of initial bins that are ignored.

- Second, our implementation averages over bins with `NSWEEPS` measurements before storing the results on disk. The error analysis requires statistically independent bins in order to generate reliable confidence estimates. If the bins are too small (averaged over a period shorter then $T_{\text{auto}}$), then the error bars are typically underestimated. Most of the time, however, the autocorrelation time is unknown before the simulation is started and, sometimes, single runs long enough to generate appropriately sized bins are not feasible. For this reason, we provide a rebinning facility controlled by the parameter `N_rebin` that specifies the number of bins recombined into each new bin during the error analysis. One can test the suitability of a given bin size by verifying whether an increase in size changes the error estimate (For an explicit example, see Sec. 4.2 and the appendix of Ref. [93]).

- The `N_rebin` variable can be used to control a further issue. The distribution of the Monte Carlo estimates $\langle\langle \hat{O} \rangle\rangle_C$ is unknown, while a result in the form (mean $\pm$ error) assumes a Gaussian distribution. Every distribution with a finite variance turns into a Gaussian one once it is folded often enough (central limit theorem). Due to the internal averaging (folding) within one bin, many observables are already quite Gaussian. Otherwise one can increase `N_rebin` further, even if the bins are already independent [135].

- The last issue we mention concerns time-displaced correlation functions. Even if the configurations are independent, the fields within the configuration are still correlated. Hence, the data for $S_{\alpha,\beta}(\boldsymbol{k}, \tau)$ [see Sec. 5.4; Eq. (123)] and $S_{\alpha,\beta}(\boldsymbol{k}, \tau + \Delta\tau)$ are also correlated. Setting the switch `N_Cov=1` triggers the calculation of the covariance matrix

in addition to the usual error analysis. The covariance is defined by

$$COV_{\tau\tau'} = \frac{1}{N_{\text{Bin}}} \left\langle \left( S_{\alpha,\beta}(\boldsymbol{k}, \tau) - \left\langle S_{\alpha,\beta}(\boldsymbol{k}, \tau) \right\rangle \right) \left( S_{\alpha,\beta}(\boldsymbol{k}, \tau') - \left\langle S_{\alpha,\beta}(\boldsymbol{k}, \tau') \right\rangle \right) \right\rangle . \quad (100)$$

An example where this information is necessary is the calculation of mass gaps extracted by fitting the tail of the time-displaced correlation function. Omitting the covariance matrix will underestimate the error.

## 4.1 The Jackknife resampling method

For each observable $\hat{A}, \hat{B}, \hat{C} \cdots$ the Monte Carlo program computes a data set of $N_{\text{Bin}}$ (ideally) independent values where for each observable the measurements belong to the same statistical distribution. In the general case, we would like to evaluate a function of expectation values, $f(\langle \hat{A} \rangle, \langle \hat{B} \rangle, \langle \hat{C} \rangle \cdots)$ – see for example the expression (27) for the observable including reweighting – and are interested in the statistical estimates of its mean value and the standard error of the mean. A numerical method for the statistical analysis of a given function $f$ which properly handles error propagation and correlations among the observables is the Jackknife method, which is, like the related Bootstrap method, a resampling scheme [136]. Here we briefly review the *delete-1 Jackknife* scheme, which consists in generating $N_{\text{bin}}$ new data sets of size $N_{\text{bin}} - 1$ by consecutively removing one data value from the original set. By $A_{(i)}$ we denote the arithmetic mean for the observable $\hat{A}$, without the $i$-th data value $A_i$, namely

$$A_{(i)} \equiv \frac{1}{N_{\text{Bin}} - 1} \sum_{k=1, \, k \neq i}^{N_{\text{Bin}}} A_k . \quad (101)$$

As the corresponding quantity for the function $f(\langle \hat{A} \rangle, \langle \hat{B} \rangle, \langle \hat{C} \rangle \cdots)$, we define

$$f_{(i)}(\langle \hat{A} \rangle, \langle \hat{B} \rangle, \langle \hat{C} \rangle \cdots) \equiv f(A_{(i)}, B_{(i)}, C_{(i)} \cdots) . \quad (102)$$

Following the convention in the literature, we will denote the final Jackknife estimate of the mean by $f_{(\cdot)}$ and its standard error by $\Delta f$. The Jackknife mean is given by

$$f_{(\cdot)}(\langle \hat{A} \rangle, \langle \hat{B} \rangle, \langle \hat{C} \rangle \cdots) = \frac{1}{N_{\text{Bin}}} \sum_{i=1}^{N_{\text{Bin}}} f_{(i)}(\langle \hat{A} \rangle, \langle \hat{B} \rangle, \langle \hat{C} \rangle \cdots) , \quad (103)$$

and the standard error, including bias correction, is given by

$$(\Delta f)^2 = \frac{N_{\text{Bin}} - 1}{N_{\text{Bin}}} \sum_{i=1}^{N_{\text{Bin}}} \left[ f_{(i)}(\langle \hat{A} \rangle, \langle \hat{B} \rangle, \langle \hat{C} \rangle \cdots) - f_{(\cdot)}(\langle \hat{A} \rangle, \langle \hat{B} \rangle, \langle \hat{C} \rangle \cdots) \right]^2 . \quad (104)$$

For $f = \langle \hat{A} \rangle$, the equations (103) and (104) reduce to the plain sample average and the standard, bias-corrected, estimate of the error.

## 4.2 An explicit example of error estimation

In the following we use one of our examples, the Hubbard model on a square lattice in the $M_z$ HS decoupling (see Sec. 9.1), to show explicitly how to estimate errors. We show as

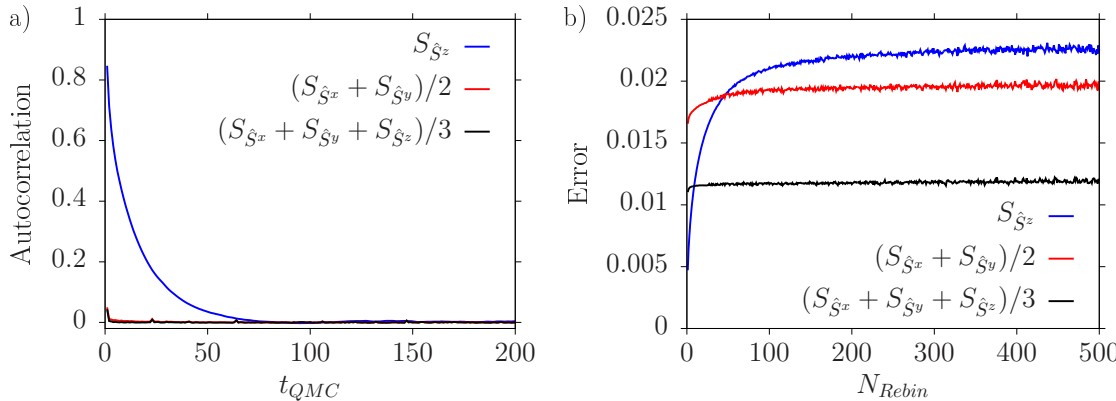

Figure 4: The autocorrelation function $S_{\hat{O}}(t_{\text{Auto}})$ (a) and the scaling of the error with effective bin size (b) of three equal-time, spin-spin correlation functions $\hat{O}$ of the Hubbard model in the $M_z$ decoupling (see Sec. 9.1). Simulations were done on a $6 \times 6$ square lattice, with $U/t = 4$ and $\beta t = 6$. We used N_auto = 500 (see Sec. 6) and a total of approximately one million bins. The original bin contained only one sweep and we calculated around one million bins on a single core. The different autocorrelation times for the $xy$-plane compared to the $z$-direction can be detected from the decay rate of the autocorrelation function (a) and from the point where saturation of the error sets in (b), which defines the required effective bin size for independent measurements. The improved estimator $(S_{\hat{S}^x} + S_{\hat{S}^y} + S_{\hat{S}^z})/3$ appears to have the smallest autocorrelation time, as argued in the text.

well that the autocorrelation time is dependent on the choice of observable. In fact, different observables within the same run can have different autocorrelation times and, of course, this time scale depends on the parameter choice. Hence, the user has to check autocorrelations of individual observables for each simulation! Typical regions of the phase diagram that require special attention are critical points where length scales diverge.

In order to determine the autocorrelation time, we calculate the correlation function

$$S_{\hat{O}}(t_{\text{Auto}}) = \sum_{i=1}^{N_{\text{Bin}}-t_{\text{Auto}}} \frac{\left(O_i - \left\langle \hat{O} \right\rangle\right)\left(O_{i+t_{\text{Auto}}} - \left\langle \hat{O} \right\rangle\right)}{\left(O_i - \left\langle \hat{O} \right\rangle\right)\left(O_i - \left\langle \hat{O} \right\rangle\right)}, \quad (105)$$

where $O_i$ refers to the Monte Carlo estimate of the observable $\hat{O}$ in the $i^{\text{th}}$ bin. This function typically shows an exponential decay and the decay rate defines the autocorrelation time. Figure 4(a) shows the autocorrelation functions $S_{\hat{O}}(t_{\text{Auto}})$ for three spin-spin-correlation functions [Eq. (123)] at momentum $\boldsymbol{k} = (\pi, \pi)$ and at $\tau = 0$:

$\hat{O} = S_{\hat{S}^z}$ for the $z$ spin direction, $\hat{O} = (S_{\hat{S}^x} + S_{\hat{S}^y})/2$ for the $xy$ plane, and $\hat{O} = (S_{\hat{S}^x} + S_{\hat{S}^y} + S_{\hat{S}^z})/3$ for the total spin. The Hubbard model has an SU(2) spin symmetry. However, we chose a HS field which couples to the $z$-component of the magnetization, $M_z$, such that each individual configuration breaks this symmetry. Of course, after Monte Carlo averaging one expects restoration of the symmetry. The model, on bipartite lattices, shows spontaneous spin-symmetry breaking at $T = 0$ and in the thermodynamic limit. At finite temperatures, and within the so-called renormalized classical regime, quantum antiferromagnets have a length scale that diverges exponentially with decreasing temperatures [137]. The

parameter set chosen for Fig. 4 is non-trivial in the sense that it places the Hubbard model in this renormalized classical regime where the correlation length is substantial. Figure 4 clearly shows a very short autocorrelation time for the $xy$-plane whereas we detect a considerably longer autocorrelation time for the $z$-direction. This is a direct consequence of the *long* magnetic length scale and the chosen decoupling. The physical reason for the long autocorrelation time corresponds to the restoration of the SU(2) spin symmetry. This insight can be used to define an improved, SU(2) symmetric estimator for the spin-spin correlation function, namely $(S_{\hat{S}^x} + S_{\hat{S}^y} + S_{\hat{S}^z})/3$. Thereby, global spin rotations are no longer an issue and this improved estimator shows the shortest autocorrelation time, as can be clearly seen in Fig. 4(b). Other ways to tackle large autocorrelations are global updates and parallel tempering.

A simple method to obtain estimates of the mean and its standard error from the time series of Monte Carlo samples is provided by the aforementioned facility of rebinning. Also known in the literature as rebatching, it consists in aggregating a fixed number `N_rebin` of adjacent original bins into a new effective bin. In addition to measuring the decay rate of the autocorrelation function (Eq. (105)), a measure for the autocorrelation time can be also obtained by the rebinning method. For a comparison to other methods of estimating the autocorrelation time we refer the reader to the literature [133, 134, 138]. A reliable error analysis requires independent bins, otherwise the error is typically underestimated. This behavior is observed in Fig. 4 (b), where the effective bin size is systematically increased by rebinning. If the effective bin size is smaller than the autocorrelation time the error will be underestimated. When the effective bin size becomes larger than the autocorrelation time, converging behavior sets in and the error estimate becomes reliable.

### 4.3 Pseudocode description

The Monte Carlo algorithm as implemented in ALF is summarized in Alg. 1. Key control variables include:

**Projector** Uses (=true) the projective instead of finite-$T$ algorithm (see Sec. 3)
$L_\tau$ Measures (Ltau=1) time-displaced observables (see Sec. 2.1.2)
**Tempering** Runs (=true) in parallel tempering mode (see Table 1)
**Global_moves** Carries out (=true) global moves in a single time slice (see Table 1)
**Sequential** Carries out (=true) sequential, single spin-flip updates (see Table 1)
**Langevin** Uses (=true) Langevin dynamics instead of sequential (see Table 1)

Per default, the finite-temperature algorithm is used, `Ltau=0`, and the updating used is Sequential (i.e., `Global_moves`, `Tempering` and `Langevin` default values are all `.false.`).

---
**Algorithm 1** Basic structure of the QMC implementation in `Prog/main.f90`
---
▷ INITIALIZATION
1: **call** Ham_Set ▷ *Set the Hamiltonian and the lattice*
2: **call** Fields_Init ▷ *Set the auxiliary fields*
3: **call** Nsigma%in ▷ *Read in an auxiliary-field configuration or generate it randomly*
4: **for** $n = L_{\text{Trotter}}$ to 1 **do** ▷ *Fill the storage needed for the first actual MC sweep*
5:    **call** Wrapul ▷ *Compute propagation matrices and store them at stabilization points*
6: **end for**

▷ MONTE CARLO RUN

7: **for** $n_{\text{bc}} = 1$ to $N_{\text{Bin}}$ **do**     ▷ *Loop over bins. The bin defines the unit of Monte Carlo time*

8:     **for** $n_{\text{sw}} = 1$ to $N_{\text{Sweep}}$ **do**     ▷ *Loop over sweeps. Each sweep updates twice (upward and downward in imaginary time) the space-time lattice of auxiliary fields*

9:        **if** Tempering **then**

10:          **call** Exchange_Step     ▷ *Perform exchange step in a parallel tempering run*

11:        **end if**

12:        **if** Global_moves **then**

13:          **call** Global_Updates     ▷ *Perform chosen global updates*

14:        **end if**

15:        **if** Langevin **then**

16:          **call** Langevin_update     ▷ UPDATE AND MEASURE *equal-time observables*

17:          **if** $L_\tau == 1$ **then**

18:            **if** Projector **then**

19:              **call** Tau_p     ▷ MEASURE *time-displaced observables (projective code)*

20:            **else**

21:              **call** Tau_m     ▷ MEASURE *time-displaced observables (finite temperature)*

22:            **end if**

23:          **end if**

24:        **end if** (Langevin)

25:        **if** Sequential **then**

         ▷ UPWARD SWEEP

26:          **for** $n_\tau = 1$ to $L_{\text{Trotter}}$ **do**

27:            **call** Wrapgrup     ▷ PROPAGATE *Green function from* $n_\tau - 1$ *to* $n_\tau$*, and compute its new estimate at* $n_\tau$*, using sequential updates*

28:            **if** $n_\tau ==$ stabilization point in imaginary time **then**     ▷ STABILIZE

29:              **call** Wrapur     ▷ *Propagate from previous stabilization point to* $n_\tau$

             ▷ *Storage management:*

               – *Read from storage: propagation from* $L_{\text{Trotter}}$ *to* $n_\tau$

               – *Write to storage: the just computed propagation*

30:              **call** CGR     ▷ *Recalculate the Green function at time* $n_\tau$ *in a stable way*

31:              **call** Control_PrecisionG     ▷ *Compare propagated and recalculated Greens*

32:            **end if**

33:            **if** $n_\tau \in [\text{Lobs\_st}, \text{Lobs\_en}]$ **then**

34:              **call** Obser     ▷ MEASURE *the equal-time observables*

35:            **end if**

36:          **end for**

         ▷ DOWNWARD SWEEP

37:          **for** $n_\tau = L_{\text{Trotter}}$ to $1$ **do**

           ▷ *Same steps as for the upward sweep (propagation and estimate update, stabilization, equal-time measurements) now downwards in imaginary time*

38:            **if** Projector **and** $L_\tau == 1$ **and**

39:              $n_\tau =$ stabilization point in imaginary time **and**

40:              the projection time $\theta$ is within the measurement interval **then**

41:              **call** Tau_p     ▷ MEASURE *time-displaced observables (projective code)*

```
42:              end if
43:            end for

         ▷ MEASURE time-displaced observables (finite temperature)
44:            if L_τ == 1 and not Projector then
45:              call Tau_m
46:            end if

47:         end if (Sequential)

48:      end for (Sweeps)

49:      call Pr_obs        ▷ Calculate and write to disk measurement averages for the current bin
50:      call Nsigma%out                    ▷ Write auxiliary field configuration to disk
51: end for (Bins)
```

## 5 Data Structures and Input/Output

To manipulate the relevant physical quantities in a general model, we define a set of corresponding data types. The `Operator` type (Sec. 5.1) is used to specify the interaction as well as the hopping. The handling of the fields is taken care of by the `Fields` type (Sec. 5.2). To define a Bravais lattice as well as a unit cell we introduce the `Lattice` and `Unit_cell` types (Sec. 5.3). General scalar, equal-time, and time-displaced correlation functions are handled by the `Observable` type (Sec. 5.4). For the projective code, we provide a `WaveFunction` type (Sec. 5.5) to specify the left and right trial wave functions. The Hamiltonian is then specified in the `Hamiltonian` module (Sec. 5.6), making use of the aforementioned types.

### 5.1 The `Operator` type

The fundamental data structure in the code is the `Operator`. It is implemented as a Fortran derived data type designed to efficiently define the Hamiltonian (2).

Let the matrix $\boldsymbol{X}$ of dimension $N_{\text{dim}} \times N_{\text{dim}}$ stand for any of the typically sparse, Hermitian matrices $\boldsymbol{T}^{(ks)}$, $\boldsymbol{V}^{(ks)}$ and $\boldsymbol{I}^{(ks)}$ that define the Hamiltonian. Furthermore, let $\{z_1, \cdots, z_N\}$ denote the subset of $N$ indices such that

$$X_{x,y} \begin{cases} \neq 0 & \text{if } x, y \in \{z_1, \cdots z_N\} \\ = 0 & \text{otherwise.} \end{cases} \tag{106}$$

Usually, we have $N \ll N_{\text{dim}}$. We define the $N \times N_{\text{dim}}$ matrices $\boldsymbol{P}$ as

$$P_{i,x} = \delta_{z_i,x} , \tag{107}$$

where $i \in [1, \cdots, N]$ and $x \in [1, \cdots, N_{\text{dim}}]$. The matrix $\boldsymbol{P}$ selects the non-vanishing entries of $\boldsymbol{X}$, which are contained in the rank-$N$ matrix $\boldsymbol{O}$ defined by:

$$\boldsymbol{X} = \boldsymbol{P}^T \boldsymbol{O} \boldsymbol{P} , \tag{108}$$

and

$$X_{x,y} = \sum_{i,j}^{N} P_{i,x} O_{i,j} P_{j,y} = \sum_{i,j}^{N} \delta_{z_i,x} O_{ij} \delta_{z_j,y} . \tag{109}$$

Since the $\boldsymbol{P}$ matrices have only one non-vanishing entry per column, they can conveniently be stored as a vector $\boldsymbol{P}$, with entries

$$P_i = z_i. \tag{110}$$

There are many useful identities which emerge from this structure. For example:

$$e^{\boldsymbol{X}} = e^{\boldsymbol{P}^T \boldsymbol{O} \boldsymbol{P}} = \sum_{n=0}^{\infty} \frac{\left(\boldsymbol{P}^T \boldsymbol{O} \boldsymbol{P}\right)^n}{n!} = \mathbb{1} + \boldsymbol{P}^T \left(e^{\boldsymbol{O}} - \mathbb{1}\right) \boldsymbol{P} , \tag{111}$$

since

$$\boldsymbol{P} \boldsymbol{P}^T = \mathbb{1}_{N \times N}. \tag{112}$$

In the code, we define a structure called `Operator` that makes use of the properties described above. This type `Operator` bundles the several components, listed in Table 2 and described in the remaining of this section, that are needed to define and use an operator matrix in the program.

| Variable | Type | Description |
|---|---|---|
| Op_X%N | int | Effective dimension $N$ |
| Op_X%O | cmplx | Matrix $\boldsymbol{O}$ of dimension $N \times N$ |
| Op_X%P | int | Matrix $\boldsymbol{P}$ encoded as a vector of dimension $N$ |
| Op_X%g | cmplx | Coupling strength $g$ |
| Op_X%alpha | cmplx | Constant $\alpha$ |
| Op_X%type | int | Sets the type of HS transformation (1: Ising; 2: discrete HS for perfect-square term; 3: continuous real field) |
| Op_X%diag | logical | True if $\boldsymbol{O}$ is diagonal |
| Op_X%U | cmplx | Matrix containing the eigenvectors of $\boldsymbol{O}$ |
| Op_X%E | dble | Eigenvalues of $\boldsymbol{O}$ |
| Op_X%N_non_zero | int | Number of non-vanishing eigenvalues of $\boldsymbol{O}$ |
| Op_X%M_exp | cmplx | Stores $\texttt{M\_exp}(:,:,s) = e^{g\phi(s,\texttt{type})\boldsymbol{O}(:,:)}$ |
| Op_X%E_exp | cmplx | Stores $\texttt{E\_exp}(:,s) = e^{g\phi(s,\texttt{type})\boldsymbol{E}(:)}$ |

Table 2: Member variables of the `Operator` type. In the left column, the letter X is a placeholder for the letters T and V, indicating hopping and interaction operators, respectively. The highlighted variables must be specified by the user. M_exp and E_exp are allocated only if `type` $= 1, 2$.

## 5.2  Handling of the fields: the `Fields` type

The partition function (see Sec. 2.1) consists of terms which, in general, can be written as $\gamma e^{g\phi\boldsymbol{X}}$, where $\boldsymbol{X}$ denotes an arbitrary operator, $g$ is a constant, and $\gamma$ and $\phi$ are fields. The ALF includes three different types of fields:

t=1  This type is for an Ising field, therefore $\gamma = 1$ and $\phi = \pm 1$,

t=2  This type is for the generic HS transformation of Eq. (11) where $\gamma \equiv \gamma(l)$ and $\phi = \eta(l)$ with $l = \pm 1, \pm 2$ [see Eq. (12)],

**t=3** This type is for continuous fields, i.e., $\gamma = 1$ and $\phi \in \mathbb{R}$.

For such auxiliary fields a dedicated type `Fields` is defined, whose components, listed in Table 5.2, include the variables `Field%f` and `Field%t`, which store the field values and types, respectively, and functions such as `Field%flip`, which flips the field values randomly. Before using this variable type, the routine `Fields_init(Amplitude)` should be called (its argument is optional and the default value is of unity (see Sec. 2.2.1), in order for internal variables such as $\eta(l)$ and $\gamma(l)$ [see Eq. (12)] to be initialized.

## 5.3 The `Lattice` and `Unit_cell` types

ALF's lattice module can generate one- and two-dimensional Bravais lattices. Both the lattice and the unit cell are defined in the module `lattices_v3_mod.F90` and their components are detailed in Tables 4 and 5. As its name suggest the module `Predefined_Latt_mod.F90` also provides predefined lattices as described in Sec. 8.1. The user who wishes to define his/her own lattice has to specify: 1) unit vectors $\boldsymbol{a}_1$ and $\boldsymbol{a}_2$, 2) the size and shape of the lattice, characterized by the vectors $\boldsymbol{L}_1$ and $\boldsymbol{L}_2$ and 3) the unit cell characterized be the number of orbitals and their positions. The coordination number of the lattice is specified in the `Unit_cell` data type. The lattice is placed on a torus (periodic boundary conditions):

$$\hat{c}_{\boldsymbol{i}+\boldsymbol{L}_1} = \hat{c}_{\boldsymbol{i}+\boldsymbol{L}_2} = \hat{c}_{\boldsymbol{i}} . \tag{113}$$

The function call

```
Call Make_Lattice( L1, L2, a1, a2, Latt )
```

generates the lattice `Latt` of type `Lattice`. The reciprocal lattice vectors $\boldsymbol{g}_j$ are defined by:

$$\boldsymbol{a}_i \cdot \boldsymbol{g}_j = 2\pi\delta_{i,j}, \tag{114}$$

and the Brillouin zone $BZ$ corresponds to the Wigner-Seitz cell of the lattice. With $\boldsymbol{k} = \sum_i \alpha_i \boldsymbol{g}_i$, the k-space quantization follows from:

$$\begin{bmatrix} \boldsymbol{L}_1 \cdot \boldsymbol{g}_1 & \boldsymbol{L}_1 \cdot \boldsymbol{g}_2 \\ \boldsymbol{L}_2 \cdot \boldsymbol{g_1} & \boldsymbol{L}_2 \cdot \boldsymbol{g}_2 \end{bmatrix} \begin{bmatrix} \alpha_1 \\ \alpha_2 \end{bmatrix} = 2\pi \begin{bmatrix} n \\ m \end{bmatrix} \tag{115}$$

such that

$$\boldsymbol{k} = n\boldsymbol{b}_1 + m\boldsymbol{b}_2, \text{ with} \tag{116}$$

$$\boldsymbol{b}_1 = \frac{2\pi}{(\boldsymbol{L}_1 \cdot \boldsymbol{g}_1)(\boldsymbol{L}_2 \cdot \boldsymbol{g}_2) - (\boldsymbol{L}_1 \cdot \boldsymbol{g}_2)(\boldsymbol{L}_2 \cdot \boldsymbol{g_1})} \left[ (\boldsymbol{L}_2 \cdot \boldsymbol{g}_2)\boldsymbol{g}_1 - (\boldsymbol{L}_2 \cdot \boldsymbol{g_1})\boldsymbol{g}_2 \right],$$

$$\boldsymbol{b}_2 = \frac{2\pi}{(\boldsymbol{L}_1 \cdot \boldsymbol{g}_1)(\boldsymbol{L}_2 \cdot \boldsymbol{g}_2) - (\boldsymbol{L}_1 \cdot \boldsymbol{g}_2)(\boldsymbol{L}_2 \cdot \boldsymbol{g_1})} \left[ (\boldsymbol{L}_1 \cdot \boldsymbol{g}_1)\boldsymbol{g}_2 - (\boldsymbol{L}_1 \cdot \boldsymbol{g_2})\boldsymbol{g}_1 \right]. \tag{117}$$

The `Lattice` module also handles the Fourier transformation. For example, the subroutine `Fourier_R_to_K` carries out the transformation:

$$S(\boldsymbol{k}, :, :, :) = \frac{1}{N_{\text{unit-cell}}} \sum_{\boldsymbol{i},\boldsymbol{j}} e^{-i\boldsymbol{k}\cdot(\boldsymbol{i}-\boldsymbol{j})} S(\boldsymbol{i} - \boldsymbol{j}, :, :, :) \tag{118}$$

| Component | | Description |
|---|---|---|
| **Variable** | **Type** | |
| `Field%t(1:n_op)` | int | Sets the HS transformation type (1: Ising; 2: discrete HS for perfect-square term; 3: continuous real field). The index runs through the operator sequence |
| `Field%f(1:n_op, 1:Ltrot)` | dble | Defines the auxiliary fields. The first index runs through the operator sequence and the second through the time slices. For `t=1`, $f = \pm 1$; for `t=2`, $f = \pm 1, \pm 2$; and for `t=3`, $f \in \mathbb{R}$ |
| `del` | dble | Width $\Delta x$ of box distribution for initial `t=3` fields, with a default value of `1` |
| `amplitude` | dble | Random flip width for fields of type `t=3`, defaults to `1` |
| **Method(arguments)** | | |
| `Field%make(n_op,Ltrot)` | | Reserves memory for the field |
| `Field%clear()` | | Clears field from memory |
| `Field%set()` | | Sets a random configuration |
| `Field%flip(n,nt)` | | Flips the field values randomly for field `n` on time slice `nt`. For `t=1` it flips the sign of the Ising spin. For `t=2` it randomly choose one of the three other values of $l$. For `t=3`, `f = f + amplitude*(ranf() -1/2)` |
| `Field%phi(n,nt)` | | Returns $\phi$ for the `n`-th operator at the time slice `nt` |
| `Field%gamma(n,nt)` | | Returns $\gamma$ for the `n`-th operator at the time slice `nt` |
| `Field%i(n,nt)` | | Returns `Field%f` rounded to nearest integer (if `t=1` or 2) |
| `Field%in(Group_Comm, In_field)` | | If the file `confin_np` exists it reads the field configuration from this file. Otherwise if `In_field` is present it sets the fields to `In_field`. If both `confin_np` and `In_field` are not provided it sets a random field by calling `Field%set()`. Here `np` is the rank number of the process |
| `Field%out(Group_Comm)` | | Writes out the field configuration |

Table 3: Components of a variable of type `Fields` named `Field`. The routine `Fields_-init(del)` should be called before the use of this variable type, since it initializes necessary internal variables such as $\eta(l)$, $\gamma(l)$ [see Eq. (12)]. Note that `del` and `amplitude` are private variables of the fields module. The integers `n_op` and `Ltrot` are the number of interacting operators per time slice and time slices, respectively, `Group_Comm` (integer) is an MPI communicator defined by the main program, and the optional `In_field` stores the initial field configuration.

and `Fourier_K_to_R` the inverse Fourier transform

$$S(\boldsymbol{r},:,:,:) = \frac{1}{N_{\text{unit-cell}}} \sum_{\boldsymbol{k} \in BZ} e^{i\boldsymbol{k}\cdot\boldsymbol{r}} S(\boldsymbol{k},:,:,:). \tag{119}$$

In the above, the unspecified dimensions of the structure factor can refer to imaginary-time and orbital indices.

| Variable | Type | Description |
|---|---|---|
| `Latt%a1_p`, `Latt%a2_p` | dble | Unit vectors $\boldsymbol{a}_1$, $\boldsymbol{a}_2$ |
| `Latt%L1_p`, `Latt%L2_p` | dble | Vectors $\boldsymbol{L}_1$, $\boldsymbol{L}_2$ that define the topology of the lattice. Tilted lattices are thereby possible to implement |
| `Latt%N` | int | Number of lattice points, $N_{\text{unit-cell}}$ |
| `Latt%list` | int | Maps each lattice point $i = 1, \cdots, N_{\text{unit-cell}}$ to a real space vector denoting the position of the unit cell: $\boldsymbol{R}_i = \texttt{list(i,1)}\boldsymbol{a}_1 + \texttt{list(i,2)}\boldsymbol{a}_2 \equiv i_1\boldsymbol{a}_1 + i_2\boldsymbol{a}_2$ |
| `Latt%invlist` | int | Return lattice point from position: $\texttt{Invlist}(i_1, i_2) = i$ |
| `Latt%nnlist` | int | Nearest neighbor indices: $j = \texttt{nnlist}(i, n_1, n_2)$, $n_1, n_2 \in [-1, 1]$, $\boldsymbol{R}_j = \boldsymbol{R}_i + n_1\boldsymbol{a}_1 + n_2\boldsymbol{a}_2$ |
| `Latt%imj` | int | $\boldsymbol{R}_{\text{imj}(i,j)} = \boldsymbol{R}_i - \boldsymbol{R}_j$, with imj, $i, j \in 1, \cdots, N_{\text{unit-cell}}$ |
| `Latt%BZ1_p`, `Latt%BZ2_p` | dble | Reciprocal space vectors $\boldsymbol{g}_i$ [See Eq. (114)] |
| `Latt%b1_p`, `Latt%b2_p` | dble | $k$-quantization [See Eq. (117)] |
| `Latt%listk` | int | Maps each reciprocal lattice point $k = 1, \cdots, N_{\text{unit-cell}}$ to a reciprocal space vector $\boldsymbol{k}_k = \texttt{listk(k,1)}\boldsymbol{b}_1 + \texttt{listk(k,2)}\boldsymbol{b}_2 \equiv k_1\boldsymbol{b}_1 + k_2\boldsymbol{b}_2$ |
| `Latt%invlistk` | int | $\texttt{Invlistk}(k_1, k_2) = k$ |
| `Latt%b1_perp_p`, `Latt%b2_perp_p` | dble | Orthonormal vectors to $\boldsymbol{b}_i$ (for internal use) |

Table 4: Components of the `Lattice` type for two-dimensional lattices using as example the default lattice name `Latt`. The highlighted variables must be specified by the user. Other components of `Lattice` are generated upon calling: `Call Make_Lattice(L1, L2, a1, a2, Latt)`.

The position of an orbital $i$ is given by $\boldsymbol{R}_i + \boldsymbol{\delta}_i$. $\boldsymbol{R}_i$ is a point of the Bravais lattice that defines a unit cell, and $\boldsymbol{\delta}_i$ labels the orbital in the unit cell. This information is stored in the array `Unit_cell%Orb_pos` detailed in Table 5.

| Variable | Type | Description |
|---|---|---|
| `Norb` | int | Number of orbitals |
| `N_coord` | int | Coordination number |
| `Orb_pos(1..Norb,2[3])` | dble | Orbitals' positions, measured from the lattice site |

Table 5: Components of an instance `Latt_unit` of the `Unit_cell` type. The highlighted variables have to be specified by the user. Note that for bilayer lattices the second index of the `Orb_pos` array ranges from 1 to 3.

The total number of orbitals is then given by `Ndim=Lattice%N*Unit_cell%Norb`. To keep track of the orbital and unit cell structure, it is useful to define arrays `List(Ndim,2)` and `Inv_list(Latt%N, Unit_cell%Norb)`. For a superindex $x = (i, n)$ labeling the unit cell, i, and the orbital, n, of a site on the lattice, we have `List(x,1)=i`, `List(x,2)=n` and `Inv_list(i,n)=x`.

## 5.4 The observable types Obser_Vec and Obser_Latt

Our definition of the model includes observables [Eq. (27)]. We define two observable types: Obser_vec for an array of *scalar* observables such as the energy, and Obser_Latt for correlation functions that have the lattice symmetry. In the latter case, translation symmetry can be used to provide improved estimators and to reduce the size of the output. We also obtain improved estimators by taking measurements in the imaginary-time interval [LOBS_ST, LOBS_EN] (see the parameter file in Sec. 5.7.1) thereby exploiting the invariance under translation in imaginary-time. Note that the translation symmetries in space and in time are *broken* for a given configuration $C$ but restored by the Monte Carlo sampling. In general, the user defines size and number of bins in the parameter file, each bin containing a given amount of sweeps. Within a sweep we run sequentially through the HS and bosonic fields, from time slice 1 to time slice $L_{\text{Trotter}}$ and back. The results of each bin are written to a file and analyzed at the end of the run.

To accomplish the reweighting of observables (see Sec. 2.1.3), for each configuration the measured value of an observable is multiplied by the factors ZS and ZP:

$$\texttt{ZS} = \text{sgn}(C) \ , \tag{120}$$

$$\texttt{ZP} = \frac{e^{-S(C)}}{\text{Re}\left[e^{-S(C)}\right]} \ . \tag{121}$$

They are computed from the Monte Carlo phase of a configuration,

$$\texttt{phase} = \frac{e^{-S(C)}}{\left|e^{-S(C)}\right|} \ , \tag{122}$$

which is provided by the main program. Note that each observable structure also includes the average sign [Eq. (28)].

### 5.4.1 Scalar observables

Scalar observables are stored in the data type Obser_vec, described in Table 6. Consider a variable Obs of type Obser_vec. At the beginning of each bin, a call to Obser_Vec_Init in the module observables_mod.F90 will set Obs%N=0, Obs%Ave_sign=0 and Obs%Obs_vec(:)=0. Each time the main program calls the routine Obser in the Hamiltonian module, the counter Obs%N is incremented by one, the sign [see Eq. (26)] is accumulated in the variable Obs%Ave_sign, and the desired observables (multiplied by the sign and $\frac{e^{-S(C)}}{\text{Re}\left[e^{-S(C)}\right]}$, see Sec. 2.1.2) are accumulated in the vector Obs%Obs_vec. At the end of the bin, a call to Print_bin_Vec in module observables_mod.F90 will append the result of the bin in the file File_Vec_scal. Note that this subroutine will automatically append the suffix _scal to the the filename File_Vec. This suffix is important to facilitate automatic analyses of the data at the end of the run. Furthermore, the file File_Vec_scal_info is created (if it does not exist yet), which contains a string that specifies how to analyze the observable and an optional description.

| Variable | Type | Description | Contribution |
|---|---|---|---|
| N | int | Number of measurements | $+1$ |
| Ave_sign | dble | Cumulated average sign [Eq. (28)] | $\mathrm{sgn}(C)$ |
| Obs_vec(:) | cmplx | Cumulated vector of observables [Eq. (27)] | $\langle\langle\hat{O}(:)\rangle\rangle_C \frac{e^{-S(C)}}{\mathrm{Re}[e^{-S(C)}]}\,\mathrm{sgn}(C)$ |
| File_Vec | char | Name of output file | |
| analysis_mode | char | How to analyze the observable Default value: "identity" | |
| description(:) | char | Optional description. Arbitrary number of 64-character lines | |

Table 6: Components of a variable of type `Obser_vec`. The contribution listed is that of each configuration $C$.

### 5.4.2    Equal-time and time-displaced correlation functions

The data type `Obser_latt` (see Table 7) is useful for dealing with both equal-time and imaginary-time-displaced correlation functions of the form:

$$S_{\alpha,\beta}(\boldsymbol{k},\tau) = \frac{1}{N_{\text{unit-cell}}} \sum_{\boldsymbol{i},\boldsymbol{j}} e^{-i\boldsymbol{k}\cdot(\boldsymbol{i}-\boldsymbol{j})} \left( \langle \hat{O}_{\boldsymbol{i},\alpha}(\tau)\hat{O}_{\boldsymbol{j},\beta}\rangle - \langle\hat{O}_{\boldsymbol{i},\alpha}\rangle\langle\hat{O}_{\boldsymbol{j},\beta}\rangle \right), \qquad (123)$$

where $\alpha$ and $\beta$ are orbital indices and $\boldsymbol{i}$ and $\boldsymbol{j}$ lattice positions. Here, translation symmetry

| Variable | Type | Description | Contribution |
|---|---|---|---|
| Obs%N | int | Number of measurements | $+1$ |
| Obs%Ave_sign | dble | Cumulated sign [Eq. (28)] | $\mathrm{sgn}(C)$ |
| Obs%Obs_latt($\boldsymbol{i}$-$\boldsymbol{j}$, $\tau,\alpha,\beta$ | cmplx | Cumulated correlation function [Eq. (27)] | $\langle\langle\hat{O}_{\boldsymbol{i},\alpha}(\tau)\hat{O}_{\boldsymbol{j},\beta}\rangle\rangle_C \times$ $\frac{e^{-S(C)}}{\mathrm{Re}[e^{-S(C)}]}\,\mathrm{sgn}(C)$ |
| Obs%Obs_latt0($\alpha$) | cmplx | Cumulated expected value [Eq. (27)] | $\langle\langle\hat{O}_{\boldsymbol{i},\alpha}\rangle\rangle_C \times$ $\frac{e^{-S(C)}}{\mathrm{Re}[e^{-S(C)}]}\,\mathrm{sgn}(C)$ |
| Obs%File_Latt | char | Name of output file | |
| Obs%Latt | Lattice* | Bravais lattice [Tab. 4] | |
| Obs%Latt_unit | Unit_cell* | Unit cell [Tab. 5] | |
| Obs%dtau | dble | Imaginary time step | |
| Obs%Channel | char | Channel for Maximum Entropy | |

Table 7: Components of a variable of type `Obser_latt` named `Obs`. Be aware: The types marked with asterisks, $*$, are actually pointers, i.e., when the subroutine `Obser_Latt_make` creates an observable `Obs`, the variables `Latt` and `Latt_unit` do not get copied but linked, meaning modifying them after the creation of `Obs` still affects the observable.

of the Bravais lattice is explicitly taken into account. The correlation function splits in a

correlated part $S_{\alpha,\beta}^{(\mathrm{corr})}(\boldsymbol{k}, \tau)$ and a background part $S_{\alpha,\beta}^{(\mathrm{back})}(\boldsymbol{k})$:

$$S_{\alpha,\beta}^{(\mathrm{corr})}(\boldsymbol{k}, \tau) = \frac{1}{N_{\mathrm{unit\text{-}cell}}} \sum_{\boldsymbol{i},\boldsymbol{j}} e^{-i\boldsymbol{k}\cdot(\boldsymbol{i}-\boldsymbol{j})} \langle \hat{O}_{\boldsymbol{i},\alpha}(\tau) \hat{O}_{\boldsymbol{j},\beta} \rangle \,, \tag{124}$$

$$
\begin{aligned}
S_{\alpha,\beta}^{(\mathrm{back})}(\boldsymbol{k}) &= \frac{1}{N_{\mathrm{unit\text{-}cell}}} \sum_{\boldsymbol{i},\boldsymbol{j}} e^{-i\boldsymbol{k}\cdot(\boldsymbol{i}-\boldsymbol{j})} \langle \hat{O}_{\boldsymbol{i},\alpha} \rangle \langle \hat{O}_{\boldsymbol{j},\beta} \rangle \\
&= N_{\mathrm{unit\text{-}cell}} \langle \hat{O}_\alpha \rangle \langle \hat{O}_\beta \rangle \, \delta(\boldsymbol{k}) \,,
\end{aligned} \tag{125}
$$

where translation invariance in space and time has been exploited to obtain the last line. The background part depends only on the expectation value $\langle \hat{O}_\alpha \rangle$, for which we use the following estimator

$$\langle \hat{O}_\alpha \rangle \equiv \frac{1}{N_{\mathrm{unit\text{-}cell}}} \sum_{\boldsymbol{i}} \langle \hat{O}_{\boldsymbol{i},\alpha} \rangle \,. \tag{126}$$

Consider a variable `Obs` of type `Obser_latt`. At the beginning of each bin a call to `Obser_Latt_Init` in the module `observables_mod.F90` will initialize the elements of `Obs` to zero. Each time the main program calls the `Obser` or `ObserT` routines one accumulates $\langle\langle \hat{O}_{\boldsymbol{i},\alpha}(\tau) \hat{O}_{\boldsymbol{j},\beta} \rangle\rangle_C \frac{e^{-S(C)}}{\mathrm{Re}[e^{-S(C)}]} \,\mathrm{sgn}(C)$ in `Obs%Obs_latt(`$\boldsymbol{i} - \boldsymbol{j}, \tau, \alpha, \beta$`)` and $\langle\langle \hat{O}_{\boldsymbol{i},\alpha} \rangle\rangle_C \frac{e^{-S(C)}}{\mathrm{Re}[e^{-S(C)}]} \cdot$ $\mathrm{sgn}(C)$ in `Obs%Obs_latt0(`$\alpha$`)`. At the end of each bin, a call to `Print_bin_Latt` in the module `observables_mod.F90` will append the result of the bin in the specified file `Obs%File_Latt`. Note that the routine `Print_bin_Latt` carries out the Fourier transformation and prints the results in $k$-space. We have adopted the following naming conventions. For equal-time observables, defined by having the second dimension of the array `Obs%Obs_latt(`$\boldsymbol{i} - \boldsymbol{j}, \tau, \alpha, \beta$`)` set to unity, the routine `Print_bin_Latt` attaches the suffix _eq_ to `Obs%File_Latt`. For time-displaced correlation functions we use the suffix _tau_. Furthermore, `Print_bin_Latt` will create a corresponding info file with suffix _eq_info_ or _tau_info_, if not already present. The info file contains the channel, number of imaginary time steps, length of one imaginary time step, unit cell and the vectors defining the Bravais lattice.

## 5.5 The `WaveFunction` type

The projective algorithm (Sec. 3) requires a pair of trial wave functions, $|\Psi_{T,L/R}\rangle$, for which there is the dedicated `WaveFunction` type, defined in the module `WaveFunction_mod` as described in Table 8.

| Variable | Type | Description |
|---|---|---|
| `WF%P(:,:)` | `cmplx` | P is an `Ndim` $\times$ `N_part` matrix, where `N_part` is the number of particles |
| `WF%Degen` | `dble` | It stores the energy difference between the last occupied and first unoccupied single particle state and can be used to check for degeneracy |

Table 8: Components of a variable of type `WaveFunction` named `WF`.

The module `WaveFunction_mod` also includes the routine `WF_overlap(WF_L, WF_R, Z_norm)` for normalizing the right trial wave function `WF_R` by the factor `Z_norm`, such that $\langle \Psi_{T,L} | \Psi_{T,R} \rangle = 1$.

## 5.6 Specification of the Hamiltonian: the `Hamiltonian` module

The module `Hamiltonian_main` in `Prog/Hamiltonian_main_mod.F90` defines the interface for all model-specific variables and subroutines needed by the Monte Carlo algorithm, like the hopping, the interaction, the observables, the trial wave function, and optionally updating schemes (see Sec. 2.2). All Hamiltonians (which is the term we are using for an encapsulated model definition) are derived from this main Hamiltonian. In order to implement a new user-defined Hamiltonian, one only has to set up a single submodule of the module `Hamiltonian_main`. Accordingly, this documentation focuses almost entirely on this module and how to derive a new model from it. The remaining parts of the code may hence be treated as a black box.

Table 9 shows all variables declared in `Hamiltonian_main`, they fully define the model. Note that the procedures listed in Table 10 are part of the variable `ham`.

To define a new Hamiltonian called *New_model*, one has to do two things:

1. Add a new line *New_model* to the file `Prog/Hamiltonians.list`

2. Write the new submodule in `Prog/Hamiltonians/Hamiltonian_New_model_smod.F90`

In this new submodule the user can redefine the procedures listed in Table 10, those have to be bound to a new type, which is derived from the Hamiltonian object `ham_base`. The submodule has access to all variables defined in `Hamiltonian_main`, while all variables defined in the submodule are encapsulated. To expose the new Hamiltonian, the user has to define

```
module Subroutine Ham_Alloc_New_model
  allocate(ham_New_model::ham)
end Subroutine Ham_Alloc_New_model
```

where `ham_New_model` is the name of the new type derived from `ham_base`. The rest of the linking is done automatically through the entry in `Prog/Hamiltonians.list`.

To simplify the implementation of a new Hamiltonian, ALF comes with a set of predefined structures (Sec. 8) which the user can combine together or use as templates.

In order to specify a Hamiltonian, we have to set the matrix representation of the imaginary-time propagators, $e^{-\Delta\tau \boldsymbol{T}^{(ks)}}$, $e^{\sqrt{-\Delta\tau U_k}\eta_{k\tau}\boldsymbol{V}^{(ks)}}$ and $e^{-\Delta\tau s_{k\tau}\boldsymbol{I}^{(ks)}}$, that appear in the partition function (17). For each pair of indices $(k,s)$, these terms have the general form

$$\text{Matrix Exponential} = e^{g\,\phi(\texttt{type})\,\boldsymbol{X}} \ . \tag{127}$$

In case of the perfect-square term, we additionally have to set the constant $\alpha$, see the definition of the operators $\hat{V}^{(k)}$ in Eq. (4). The data structures which hold all the above information are variables of the type `Operator` (see Table 2). For each pair of indices $(k,s)$, we store the following parameters in an `Operator` variable:

- $\boldsymbol{P}$ and $\boldsymbol{O}$ defining the matrix $\boldsymbol{X}$ [see Eq. (108)],

- the constants $g$, $\alpha$,

- optionally: the type `type` of the discrete fields $\phi$.

The latter parameter can take one of three values: Ising (1), discrete HS (2), and real (3), as detailed in Sec. 5.2. Note that we have dropped the color index $\sigma$, since the implementation uses the SU($N_{\text{col}}$) invariance of the Hamiltonian.

Accordingly, the following data structures fully describe the Hamiltonian (2):

| Public Variable | Type | Description |
| --- | --- | --- |
| ham | class(ham_base) | Hamiltonian object. All model dependent procedures are attached to this variable (see Table 10). |
| Op_V | Operator | Interaction |
| Op_T | Operator | Hopping |
| WF_L | WaveFunction | Left trial wave function |
| WF_R | WaveFunction | Right trial wave function |
| nsigma | Fields | Fields |
| Ndim | int | Number of sites |
| N_Fl | int | Number of flavors |
| N_SUN | int | Number of colors |
| Ltrot | int | Total number of trotter silces |
| Thtrot | int | Number of trotter slices reserved for projection |
| Projector | logical | Enable projector code |
| Group_Comm | int | Group communicator for MPI |
| Symm | logical | Symmetric trotter |

| Private Variable | Type | Description |
| --- | --- | --- |
| Obs_scal | Obser_Vec | Storage for measured scalar observables |
| Obs_eq | Obser_Latt | Storage for measured equal time correlations |
| Obs_tau | Obser_Latt | Storage for measured time displaced correlations |

Table 9: List of the public and private variables declared in the module `Hamiltonian`. The highlighted variables have to be set in the subroutine `ham_set`.

- For the hopping Hamiltonian (3), we have to set the exponentiated hopping matrices $e^{-\Delta\tau \boldsymbol{T}^{(ks)}}$:
  In this case $\boldsymbol{X}^{(ks)} = \boldsymbol{T}^{(ks)}$, and a single variable `Op_T` describes the operator matrix

$$\left( \sum_{x,y}^{N_{\text{dim}}} \hat{c}^\dagger_{xs} T^{(ks)}_{xy} \hat{c}_{ys} \right) , \tag{128}$$

  where $k = [1, M_T]$ and $s = [1, N_{\text{fl}}]$. In the notation of the general expression (127), we set $g = -\Delta\tau$ (and $\alpha = 0$). In case of the hopping matrix, the type variable takes its default value `Op_T%type` $= 0$. All in all, the corresponding array of structure variables is `Op_T(M_T,N_fl)`.

- For the interaction Hamiltonian (4), which is of perfect-square type, we have to set the exponentiated matrices $e^{\sqrt{-\Delta\tau U_k}\eta_{k\tau}\boldsymbol{V}^{(ks)}}$:
  In this case, $\boldsymbol{X} = \boldsymbol{V}^{(ks)}$ and a single variable `Op_V` describes the operator matrix:

$$\left[ \left( \sum_{x,y}^{N_{\text{dim}}} \hat{c}^\dagger_{xs} V^{(ks)}_{x,y} \hat{c}_{ys} \right) + \alpha_{ks} \right] , \tag{129}$$

  where $k = [1, M_V]$ and $s = [1, N_{\text{fl}}]$, $g = \sqrt{-\Delta\tau U_k}$ and $\alpha = \alpha_{ks}$. The discrete HS decomposition which is used for the perfect-square interaction, is selected by setting the

| Procedure | Description | Section |
|---|---|---|
| Ham_Set | Reads in model and lattice parameters from the file parameters. Sets the Hamiltonian, which is commonly split up into subroutines Ham_Latt, Ham_Hop, Ham_V and Ham_Trial | 5.6, 9 |
| | Ham_Latt:   Sets the Lattice and the Unit_cell as well as the the arrays List and Inv_list required for multiorbital problems | 5.3, 7.2 8.1 |
| | Ham_hop:   Sets the hopping term $\hat{\mathcal{H}}_T$ (i.e., operator Op_T) by calling Op_make and Op_set | 5.1, 7.3, 8.2 |
| | Ham_V:   Sets the interaction term $\hat{\mathcal{H}}_V$ (i.e., operator Op_V) by calling Op_make and Op_set | 5.1, 7.4, 8.3 |
| | Ham_Trial:   Sets the trial wave function for the projective code $|\Psi_{T,L/R}\rangle$ specified by the Wavefunction type | 5.5, 7.5, 8.5 |
| Alloc_obs | Assigns memory storage to the observable | 5.4 , 7.6.1 |
| Obser | Computes the scalar and equal-time observables | 5.4, 7.6.2, 8.4 |
| ObserT | Computes time-displaced correlation functions | 5.4, 7.6.3, 8.4 |
| S0 | Returns the ratio $e^{S_0(C')}/e^{-S_0(C)}$ for a single spin flip | 2.2.2 |
| Global_move_tau | Generates a global move on a given time slice $\tau$. This routine is only called if Global_tau_moves=True and N_Global_tau>0 | 2.2.3 |
| Overide_global_tau_sampling_parameters | Allows setting global_tau parameters at run time | 2.2.3 |
| Hamiltonian_set_nsigma | Sets the initial field configuration. This routine is to be modified if one wants to specify the initial configuration. By default the initial configuration is assumed to be random | |
| Global_move | Handles global moves in time and space | 2.2.4 |
| Delta_S0_global | Computes $e^{S_0(C')}/e^{-S_0(C)}$ for a global move | 2.2.4 |
| Init_obs | Initializes the observables to zero. Usually, this doesn't have to be modified. | |
| Pr_obs | Writes the observables to disk by calling Print_bin of the Observables module. Usually, this doesn't have to be modified. | |

Table 10: Typebound procedures bound to type ham_base. To define a new model, at least Ham_Set has to be overloaded in the Hamiltonian submodule. For measurements Alloc_obs, Obser (and ObserT for time displaced observables) are necessary. The other procedures are needed for optional features.

type variable to Op_V%type = 2. All in all, the required structure variables Op_V are defined using the array Op_V($M_V$,$N_{fl}$).

- For the bosonic interaction Hamiltonian (5), we have to set the exponentiated matrices $e^{-\Delta\tau s_{k\tau}\boldsymbol{I}^{(ks)}}$:

  In this case, $\boldsymbol{X} = \boldsymbol{I}^{(k,s)}$ and a single variable `Op_V` then describes the operator matrix:

$$\left(\sum_{x,y}^{N_{\mathrm{dim}}} \hat{c}_{xs}^{\dagger} I_{xy}^{(ks)} \hat{c}_{ys}\right) , \tag{130}$$

  where $k = [1, M_I]$ and $s = [1, N_{\mathrm{fl}}]$ and $g = -\Delta\tau$ (and $\alpha = 0$). It this operator couples to an Ising field, we specify the type variable `Op_V%type=1`. On the other hand, if it couples to a scalar field (i.e. real number) then we specify `Op_V%type=3`. All in all, the required structure variables are contained in the array `Op_V(M_I,N_fl)`.

- In case of a full interaction [perfect-square term (4) and bosonic term (5)], we define the corresponding doubled array `Op_V(M_V+M_I,N_fl)` and set the variables separately for both ranges of the array according to the above.

## 5.7 File structure

| Directory | Description |
|---|---|
| `Prog/` | Main program and subroutines |
| `Libraries/` | Collection of mathematical routines |
| `Analysis/` | Routines for error analysis |
| `Scripts_and_Parameters_files/` | Helper scripts and the `Start/` directory, which contains the files required to start a run |
| `Documentation/` | This documentation |
| `Mathematica/` | Mathematica notebooks to evaluate higher order correlation functions with Wicks theorem |
| `testsuite/` | An automatic test suite for various parts of the code |

Table 11: Overview of the directories included in the ALF package.

The code package, summarized in Table 11, consists of the program directories `Prog/`, `Libraries/`, `Analysis/`, and the directory `Scripts_and_Parameters_files/`, which contains supporting scripts and, in its subdirectory `Start`, the input files necessary for a run, described in the Sec. 5.7.1 as well as `Mathematica/` that contains Mathematica notebooks to evaluate higher order correlation functions with Wicks theorem as described in Appendix A. The routines available in the directory `Analysis/` are described in Sec. 6.3, and the testsuite in Sec. 6.2.

Below we describe the structure of ALF's input and output files. Notice that the input/output files for the Analysis routines are described in Sec. 6.3.

### 5.7.1 Input files

The package's two input files are described in Table 12. The parameter file `Start/parameters` has the following form – using as an example the Hubbard model on a square lattice (see

| File | Description |
|------|-------------|
| `parameters` | Defines which Hamiltonian to use and sets the parameters for: lattice, model, QMC process, and error analysis |
| `seeds` | List of integer numbers to initialize the random number generator and to start a simulation from scratch |

Table 12: Overview of the input files required for a simulation, which can be found in the subdirectory `Scripts_and_Parameters_files/Start/`.

Sec. 9.1 for the general SU(N) Hubbard and Sec. 7 for a detailed walk-through on its plain vanilla version):

```
!==============================================================================
!  Input variables for a general ALF run
!------------------------------------------------------------------------------
&VAR_ham_name                    !! Use Hamiltonian defined in
ham_name = "Hubbard"              ! Prog/Hamiltonians/Hamiltonian_{ham_name}_smod.F90
/

&VAR_lattice                     !! Parameters defining the specific lattice and base model
L1            = 6                 ! Length in direction a_1
L2            = 6                 ! Length in direction a_2
Lattice_type = "Square"          ! Sets a_1 = (1,0), a_2=(0,1), Norb=1, N_coord=2
Model        = "Hubbard"         ! Sets the Hubbard model, to be specified in &VAR_Hubbard
/

&VAR_Model_Generic               !! Common model parameters
Checkerboard = .T.               ! Whether checkerboard decomposition is used
Symm         = .T.               ! Whether symmetrization takes place
N_SUN        = 2                 ! Number of colors
N_FL         = 1                 ! Number of flavors
Phi_X        = 0.d0              ! Twist along the L_1 direction, in units of the flux quanta
Phi_Y        = 0.d0              ! Twist along the L_2 direction, in units of the flux quanta
Bulk         = .T.               ! Twist as a vector potential (.T.); at the boundary (.F.)
N_Phi        = 0                 ! Total number of flux quanta traversing the lattice
Dtau         = 0.1d0             ! Thereby Ltrot=Beta/dtau
Beta         = 5.d0              ! Inverse temperature
Projector    = .F.               ! Whether the projective algorithm is used
Theta        = 10.d0             ! Projection parameter
/

&VAR_QMC                         !! Variables for the QMC run
Nwrap              = 10   ! Stabilization. Green functions will be computed from
                         ! scratch after each time interval Nwrap*Dtau
NSweep             = 20   ! Number of sweeps
NBin               = 5    ! Number of bins
Ltau               = 1    ! 1 to calculate time-displaced Green functions; 0 otherwise
LOBS_ST            = 0    ! Start measurements at time slice LOBS_ST
LOBS_EN            = 0    ! End measurements at time slice LOBS_EN
CPU_MAX            = 0.0  ! Code stops after CPU_MAX hours, if 0 or not
                         ! specified, the code stops after Nbin bins
Propose_S0         = .F.  ! Proposes single spin flip moves with probability exp(-S0)
Global_moves       = .F.  ! Allows for global moves in space and time
```

```
N_Global            = 1     ! Number of global moves per sweep
Global_tau_moves    = .F.   ! Allows for global moves on a single time slice.
N_Global_tau        = 1     ! Number of global moves that will be carried out on a
                            ! single time slice
Nt_sequential_start = 0     ! One can combine sequential & global moves on a time slice
Nt_sequential_end   = -1    ! The program then carries out sequential local moves in the
                            ! range [Nt_sequential_start, Nt_sequential_end] followed by
                            ! N_Global_tau global moves
Langevin            = .F.   ! Langevin update
Delta_t_Langevin_HMC = 0.01 ! Default time step for Langevin and HMC updates
Max_Force           = 1.5   ! Max Force for  Langevin
/

&VAR_errors               !! Variables for analysis programs
n_skip  = 1               ! Number of bins that to be skipped
N_rebin = 1               ! Rebinning
N_Cov   = 0               ! If set to 1 covariance computed for non-equal-time
                          ! correlation functions
N_auto   = 0              ! If > 0  triggers  calculation of autocorrelation
N_Back   = 1              ! If set to 1, substract background in correlation functions
/

&VAR_TEMP                 !! Variables for parallel tempering
N_exchange_steps     = 6   ! Number of exchange moves [see Eq. (39)]
N_Tempering_frequency = 10  ! The frequency in units of sweeps at which the
                           ! exchange moves are carried out
mpi_per_parameter_set = 2   ! Number of mpi-processes per parameter set
Tempering_calc_det    = .T. ! Specifies whether the fermion weight has to be taken
                           ! into account while tempering. The default is .true.,
                           ! and it can be set to .F. if the parameters that
                           ! get varied only enter the free bosonic action S_0
/

&VAR_Max_Stoch            !! Variables for Stochastic Maximum entropy
Ngamma    = 400           ! Number of Dirac delta-functions for parametrization
Om_st     = -10.d0        ! Frequency range lower bound
Om_en     = 10.d0         ! Frequency range upper bound
NDis      = 2000          ! Number of boxes for histogram
Nbins     = 250           ! Number of bins for Monte Carlo
Nsweeps   = 70            ! Number of sweeps per bin
NWarm     = 20            ! The Nwarm first bins will be ommitted
N_alpha   = 14            ! Number of temperatures
alpha_st  = 1.d0          ! Smallest inverse temperature increment for inverse
R         = 1.2d0         ! temperature (see above)
Checkpoint = .F.          ! Whether to produce dump files, allowing the simulation
                          ! to be resumed later on
Tolerance = 0.1d0         ! Data points for which the relative error exceeds the
                          ! tolerance threshold will be omitted.
/

&VAR_Hubbard              !! Variables for the specific model
Mz        = .T.           ! When true, sets the M_z-Hubbard model: Nf=2, demands that
                          ! N_sun is even, HS field couples to the z-component of
                          ! magnetization; otherwise, HS field couples to the density
Continuous = .F.          ! Uses (T: continuous; F: discrete) HS transformation
ham_T     = 1.d0          ! Hopping parameter
ham_chem  = 0.d0          ! Chemical potential
```

```
ham_U      = 4.d0            ! Hubbard interaction
ham_T2     = 1.d0            ! For bilayer systems
ham_U2     = 4.d0            ! For bilayer systems
ham_Tperp  = 1.d0            ! For bilayer systems
/
```

The program allows for a number of different updating schemes. If no other variables are specified in the `VAR_QMC` name space, then the program will run in its default mode, namely the sequential single spin-flip mode. In particular, note that if `Nt_sequential_start` and `Nt_sequential_end` are not specified and that the variable `Global_tau_moves` is set to true, then the program will carry out only global moves, by setting `Nt_sequential_start=1` and `Nt_sequential_end=0`.

### 5.7.2 Output files – observables

The standard output files are listed in Table 13. Notice that, besides these files, which contain direct QMC outputs, ALF can also produce a number of analysis output files, discussed in Sec. 6.3.

The output of the measured data is organized in bins. One bin corresponds to the arithmetic average over a fixed number of individual measurements which depends on the chosen measurement interval `[LOBS_ST,LOBS_EN]` on the imaginary-time axis and on the number `NSweep` of Monte Carlo sweeps. If the user runs an MPI parallelized version of the code, the average also extends over the number of MPI threads.

| File | Description |
|---|---|
| `info` | After completion of the simulation, this file documents the parameters of the model, as well as the QMC run and simulation metrics (precision, acceptance rate, wallclock time) |
| `X_scal` | Results of equal-time measurements of scalar observables. The placeholder `X` stands for the observables `Kin`, `Pot`, `Part`, and `Ener` |
| `X_scal_info` | Contains info on how to analyze the observable and optionally a description. |
| `Y_eq,Y_tau` | Results of equal-time and time-displaced measurements of correlation functions. The placeholder `Y` stands for `Green`, `SpinZ`, `SpinXY`, `Den`, etc. |
| `Y_eq_info,Y_tau_info` | Additional info, like Bravais lattice and unit cell, for equal-time and time-displaced observables |
| `confout_<threadnumber>` | Output files (one per MPI instance) for the HS and bosonic configuration |

Table 13: Overview of the standard output files. See Sec. 5.4 for the definitions of observables and correlation functions.

The formatting of a single bin's output depends on the observable type, `Obs_vec` or `Obs_Latt`:

- Observables of type `Obs_vec`: For each additional bin, a single new line is added to the output file. In case of an observable with `N_size` components, the formatting is

```
N_size+1  <measured value, 1> ... <measured value, N_size>  <measured sign>
```

  The counter variable `N_size+1` refers to the number of measurements per line, including the phase measurement. This format is required by the error analysis routine (see Sec. 6.3). Scalar observables like kinetic energy, potential energy, total energy and particle number are treated as a vector of size `N_size=1`.

- Observables of type `Obs_Latt`: For each additional bin, a new data block is added to the output file. The block consists of the expectation values [Eq. (126)] contributing to the background part [Eq. (125)] of the correlation function, and the correlated part [Eq. (124)] of the correlation function. For imaginary-time displaced correlation functions, the formatting of the block is given by:

```
<measured sign> <N_orbital> <N_unit_cell> <N_time_slices> <dtau> <Channel>
do alpha = 1, N_orbital
    ⟨Ô_α⟩
enddo
do i = 1, N_unit_cell
    <reciprocal lattice vector k(i)>
    do tau = 1, N_time_slices
        do alpha = 1, N_orbital
            do beta = 1, N_orbital
                ⟨S^(corr)_{α,β}(k(i),τ)⟩
            enddo
        enddo
    enddo
enddo
```

  The same block structure is used for equal-time correlation functions, except for the entries `<N_time_slices>`, `<dtau>` and `<Channel>`, which are then omitted. Using this structure for the bins as input, the full correlation function $S_{\alpha,\beta}(\boldsymbol{k},\tau)$ [Eq. (123)] is then calculated by calling the error analysis routine (see Sec. 6.3).

# 6 Using the Code

In this section we describe the steps for compiling and running the code from the shell, and describe how to search for optimal parameter values as well as how to perform the error analysis of the data.

The source code of ALF 2.1 is available at https://git.physik.uni-wuerzburg.de/ALF/ALF/-/tree/ALF-2.1 and can be cloned with git or downloaded from the repository (make sure to chose the appropriate release, 2.1).

A Python interface, **pyALF**, is also available and can be found, together with a number of Jupyter notebooks exploring the interface's capabilities, at https://git.physik.uni-wuerzburg.de/ALF/pyALF/-/tree/ALF-2.0/. This interface facilitates setting up simple runs and is ideal for setting benchmarks and getting acquainted with ALF. Some of pyALF's notebooks form the core of the introductory part of the ALF Tutorial, where pyALF's usage is described in more detail.

We start out by providing step-by-step instructions that allow a first-time user to go from zero to performing a simulation and reading out their first measurement using ALF.

## 6.1    Zeroth step

The aim of this section is to provide a fruitful and stress-free first contact with the package. Ideally, it should be possible to copy and paste the instructions below to a Debian/Ubuntu-based Linux shell without further thought[4]. Explanations and further options and details are found in the remaining sections and in the Tutorial.

**Prerequisites**: You should have access to a shell and the permissions to install – or have already installed – the numerical packages Lapack and Blas, a Fortran compiler and the tools `make` and `git`.

The following commands can be executed in a Debian-based shell[5] in order to install ALF 2.1 and its dependencies, run a demonstration simulation and output one of the measurements performed:

- `sudo apt-get install gfortran liblapack-dev make git`
- `git clone -b ALF-2.1 https://git.physik.uni-wuerzburg.de/ALF/ALF.git`
- `cd ALF`
- `source configure.sh GNU noMPI`
- `make Hubbard_Plain_Vanilla`
- `cp -r ./Scripts_and_Parameters_files/Start ./Run && cd ./Run/`
- `$ALF_DIR/Prog/Hubbard_Plain_Vanilla.out`
- `$ALF_DIR/Analysis/ana.out Ener_scal`
- `cat Ener_scalJ`

The last command will output a few lines, including one similar to:

```
 OBS :    1      -30.009191       0.110961
```

which is listing the internal energy of the system and its error.

## 6.2    Compiling and running

The necessary environment variables and the directives for compiling the code are set by the script `configure.sh`:

```
source configure.sh [MACHINE] [MODE] [STAB]
```

If run with no arguments, it lists the available options and sets a generic, serial GNU compiler with minimal flags `-cpp -O3 -ffree-line-length-none -ffast-math`. The predefined machine configurations and parallelization modes available, as well as the options for stabilization schemes for the matrix multiplications (see Sec. 2.4) are shown in Table 14. The stabilization scheme choice, in particular, is critical for performance and is discussed further in Sec. 6.4.

In order to compile the libraries, the analysis routines and the QMC program at once, just execute the single command:

---

[4]For other systems and distributions see the package's README.

[5]Avoid folder names containing spaces, which are not supported.

```
make
```

Related auxiliary directories, object files and executables can be removed by executing the command `make clean`. The accompanying `Makefile` also provides rules for compiling and cleaning up the library, the analysis routines and the QMC program separately.

| Argument | Selected feature |
|----------|------------------|
| `MACHINE` | |
| `GNU` | GNU compiler (`gfortran` or `mpifort`) for a generic machine (*default*) |
| `Intel` | Intel compiler (`ifort` or `mpiifort`) for a generic machine[6] |
| `PGI` | PGI compiler (`pgfortran` or `mpifort`) for a generic machine |
| `SuperMUC-NG` | Intel compiler (`mpiifort`) and loads modules for SuperMUC-NG[7] |
| `JUWELS` | Intel compiler (`mpiifort`) and loads modules for JUWELS[8] |
| `Development` | GNU compiler (`gfortran` or `mpifort`) with debugging flags |
| `MODE` | |
| `noMPI\|Serial` | No parallelization |
| `MPI` | MPI parallelization (*default* – if a machine is selected) |
| `Tempering` | Parallel tempering (Sec. 2.2.5) and the required MPI as well |
| `STAB` | |
| `STAB1` | Simplest stabilization, with UDV (QR-, not SVD-based) decompositions |
| `STAB2` | QR-based UDV decompositions with additional normalizations |
| `STAB3` | Newest scheme, additionally separates large and small scales (*default*) |
| `LOG` | Log storage for internal scales, increases accessible ranges |

Table 14: Available arguments for the script `configure.sh`, called before compilation of the package: predefined machines, parallelization modes, and stabilization schemes (see also Sec. 6.4).

A suite of tests for individual parts of the code (subroutines, functions, operations, etc.) is available at the directory `testsuite`. The tests can be run by executing the following sequence of commands (the script `configure.sh` sets environment variables as described above):

```
source configure.sh Devel serial
gfortran -v
make lib
make ana
make program
```

---

[6] A known issue with the alternative Intel Fortran compiler `ifort` is the handling of automatic, temporary arrays which `ifort` allocates on the stack. For large system sizes and/or low temperatures this may lead to a runtime error. One solution is to demand allocation of arrays above a certain size on the heap instead of the stack. This is accomplished by the `ifort` compiler flag `-heap-arrays [n]` where `[n]` is the minimal size (in kilobytes, for example `n=1024`) of arrays that are allocated on the heap.

[7] Supercomputer at the Leibniz Supercomputing Centre.

[8] Supercomputer at the Jülich Supercomputing Centre.

```
cd testsuite
cmake -E make_directory tests
cd tests
cmake -G "Unix Makefiles" -DCMAKE_Fortran_FLAGS_RELEASE=${F90OPTFLAGS} \
-DCMAKE_BUILD_TYPE=RELEASE ..
cmake --build . --target all --config Release
ctest -VV -O log.txt
```

which will output test results and total success rate.

### Starting a simulation

In order to start a simulation from scratch, the following files have to be present: `parameters` and `seeds` (see Sec. 5.7.1). To run serial simulation, issue the command

`$ALF_DIR/Prog/ALF.out`

In order to run with MPI parallelization, the appropriate MPI execution command should be called. For instance, a program compiled with OpenMPI can be run in parallel by issuing

`orterun -np <number of processes> $ALF_DIR/Prog/ALF.out`

The environment variable `ALF_SHM_CHUNK_SIZE_GB` can be used to reduce the program's memory footprint by sharing memory between MPI processes on the same node. The variable, a positive real number, defines the chunk size of the shared memory objects in units of GB. Typical values are 1.0 or 2.0 GB, but larger values can be used, if otherwise the total number of MPI communicators so large as to trigger MPI error messages. If `ALF_SHM_CHUNK_SIZE_GB` is not defined or set to values smaller that one, then the memory is not shared between MPI processes, which is the default behavior.

To restart the code using the configuration from a previous simulation as a starting point, first run the script `out_to_in.sh`, which copies outputted field configurations into input files, before calling the ALF executable. This file is located in the directory `$ALF_DIR/Scripts_and_Parameters_files/Start/`

### 6.3 Error analysis

The ALF package includes the analysis program `ana.out` for performing simple error analysis and correlation function calculations on the three observable types. To perform an error analysis based on the Jackknife resampling method [136] (Sec. 4.1) of the Monte Carlo bins for a list of observables run

`$ALF_DIR/Analysis/ana.out <list of files>`

or run

`$ALF_DIR/Analysis/ana.out *`

for all observables.

The program `ana.out` is based on the included module `ana_mod`, which provides subroutines for reading and analyzing ALF Monte Carlo bins, that can be used to implement more specialized analysis. The three high-level analysis routines employed by `ana_mod` are listed in Table 15. The files taken as input, as well as the output files are listed in Table 16.

| Program | Description |
|---------|-------------|
| cov_vec(name) | The bin file **name**, which should have suffix _scal, is read in, and the corresponding file with suffix _scalJ is produced. It contains the result of the Jackknife rebinning analysis (see Sec. 4) |
| cov_eq(name) | The bin file **name**, which should have suffix _eq, is read in, and the corresponding files with suffix _eqJR and _eqJK are produced. They correspond to correlation functions in real and Fourier space, respectively |
| cov_tau(name) | The bin file **name**, which should have suffix _tau, is read in, and the directories X_kx_ky are produced for all kx and ky greater or equal to zero. Here X is a place holder from Green, SpinXY, etc., as specified in Alloc_obs(Ltau) (See section 7.6.1). Each directory contains a file g_dat containing the time-displaced correlation function traced over the orbitals. It also contains the covariance matrix if N_cov is set to unity in the parameter file (see Sec. 5.7.1). Besides, a directory X_R0 for the local time displaced correlation function is generated. For particle-hole, imaginary-time correlation functions (Channel = "PH") such as spin and charge, we use the fact that these correlation functions are symmetric around $\tau = \beta/2$ so that we can define an improved estimator by averaging over $\tau$ and $\beta - \tau$ |

Table 15: Overview of analysis subroutines called within the program `ana.out`.

The error analysis is based on the central limit theorem, which requires bins to be statistically independent, and also the existence of a well-defined variance for the observable under consideration (see Sec. 4). The former will be the case if bins are longer than the autocorrelation time – autocorrelation functions are computed by setting the parameter N_auto to a nonzero value – which has to be checked by the user. In the parameter file described in Sec. 5.7.1, the user can specify how many initial bins should be omitted (variable n_skip). This number should be comparable to the autocorrelation time. The rebinning variable N_rebin will merge N_rebin bins into a single new bin. If the autocorrelation time is smaller than the effective bin size, the error should become independent of the bin size and thereby of the variable N_rebin. The analysis output files listed in Table 16 and are formatted in the following way:

- For the scalar quantities X, the output files X_scalJ have the following formatting:

```
Effective number of bins, and bins:  <N_bin - N_skip>/<N_rebin>  <N_bin>
OBS :  1    <mean(X)>       <error(X)>
OBS :  2    <mean(sign)>    <error(sign)>
```

- For the equal-time correlation functions Y, the formatting of the output files Y_eqJR and Y_eqJK follows the structure:

```
do i = 1, N_unit_cell
   <k_x(i)>   <k_y(i)>
   do alpha = 1, N_orbital
      do beta  = 1, N_orbital
          alpha  beta  Re<mean(Y)>  Re<error(Y)>  Im<mean(Y)>  Im<error(Y)>
```

| File | Description |
|---|---|
| **Input** | |
| `parameters` | Includes error analysis variables `N_skip`, `N_rebin`, and `N_Cov` (see Sec. 5.7.1) |
| `X_scal`, `Y_eq`, `Y_tau` | Monte Carlo bins (see Table 13) |
| **Output** | |
| `X_scalJ` | Jackknife mean and error of X, where X stands for `Kin`, `Pot`, `Part`, or `Ener` |
| `Y_eqJR` and `Y_eqJK` | Jackknife mean and error of Y, which stands for `Green`, `SpinZ`, `SpinXY`, or `Den`. The suffixes R and K refer to real and reciprocal space, respectively |
| `Y_R0/g_R0` | Time-resolved and spatially local Jackknife mean and error of Y, where Y stands for `Green`, `SpinZ`, `SpinXY`, and `Den` |
| `Y_kx_ky/g_kx_ky` | Time resolved and $k$-dependent Jackknife mean and error of Y, where Y stands for `Green`, `SpinZ`, `SpinXY`, and `Den` |
| `Part_scal_Auto` | Autocorrelation functions $S_{\hat{O}}(t_{\text{Auto}})$ in the range $t_{\text{Auto}} = [0, \texttt{N\_auto}]$ for the observable $\hat{O}$ |

Table 16: Standard input and output files of the error analysis program `ana.out`.

```
        enddo
     enddo
  enddo
```

where `Re` and `Im` refer to the real and imaginary part, respectively.

- The imaginary-time displaced correlation functions Y are written to the output files `g_R0` inside folders `Y_R0`, when measured locally in space; and to the output files `g_kx_ky` inside folders `Y_kx_ky` when they are measured $k$-resolved (where $k = (\texttt{kx}, \texttt{ky})$). The first line of the file contains the number of imaginary times, the effective number of bins, $\beta$, the number of orbitals and the channel. Both output files have the following formatting:

```
do i = 0, Ltau
    tau(i)   <mean( Tr[Y] )>   <error( Tr[Y])>
enddo
```

where `Tr` corresponds to the trace over the orbital degrees of freedom. For particle-hole quantities at finite temperature, $\tau$ runs from 0 to $\beta/2$. In all other cases it runs from 0 to $\beta$.

- The file `Y_tauJK` contains the susceptibilities defined as:

$$\chi(\boldsymbol{q}) = \sum_{n,n'=1}^{\text{Norb}} \int_0^\beta d\tau \left( \langle Y_n(\boldsymbol{q}, \tau) Y_{n'}(-\boldsymbol{q}, 0) \rangle - \langle Y_n(\boldsymbol{q}) \rangle \langle Y_{n'}(-\boldsymbol{q}) \rangle \delta_{\boldsymbol{q},\boldsymbol{0}} \right) \qquad (131)$$

The output file has the following formatting:

```
do i = 0, Ltau
   q_x, q_y, <mean(Real(chi(q)) )>,  <error(Real(chi(q)))>, &
         & <mean(Im  (chi(q)) )>,  <error(lmi (chi(q)))>
enddo
```

- Setting the parameter `N_auto` to a finite value triggers the computation of autocorrelation functions $S_{\hat{O}}(t_{\text{Auto}})$ in the range $t_{\text{Auto}} = [0, \text{N\_auto}]$. The output is written to the file `Part_scal_Auto`, where the data in organized in three columns:

  $t_{\text{Auto}}$      $S_{\hat{O}}(t_{\text{Auto}})$      `error`

  Since these computations are quite time consuming and require many Monte Carlo bins, our default is `N_auto=0`.

## 6.4   Parameter optimization

The finite-temperature, auxiliary-field QMC algorithm is known to be numerically unstable, as discussed in Sec. 2.4. The numerical instabilities arise from the imaginary-time propagation, which invariably leads to exponentially small and exponentially large scales. As shown in Ref. [6], scales can be omitted in the ground state algorithm – thus rendering it very stable – but have to be taken into account in the finite-temperature code.

Numerical stabilization of the code is a delicate procedure that has been pioneered in Ref. [2] for the finite-temperature algorithm and in Refs. [3,4] for the zero-temperature, projective algorithm. It is important to be aware of the fragility of the numerical stabilization and that there is no guarantee that it will work for a given model. It is therefore crucial to always check the file `info`, which, apart from runtime data, contains important information concerning the stability of the code, in particular `Precision Green`. If the numerical stabilization fails, one possible measure is to reduce the value of the parameter `Nwrap` in the parameter file, which will however also impact performance – see Table. 17 for further optimization tips for the Monte Carlo algorithm (Sec. 4). Typical values for the numerical precision ALF can achieve can be found in Sec. 9.1.

In particular, for the stabilization of the involved matrix multiplications we rely on routines from LAPACK. Notice that results are very likely to change depending on the specific implementation of the library used[9]. In order to deal with this possibility, we offer a simple baseline which can be used as a quick check as tho whether results depend on the library used for linear algebra routines. Namely, we have included QR-decomposition related routines of the LAPACK-3.7.0 reference implementation from http://www.netlib.org/lapack/, which you can use by running the script `configure.sh`, (described in Sec. 6), with the flag `STAB1` and recompiling ALF[10]. The stabilization flags available are described in Tables 14 and 17. The performance of the package is further discussed in Sec. B.

# 7   The plain vanilla Hubbard model on the square lattice

All the data structures necessary to implement a given model have been introduced in the previous sections. Here we show how to implement the Hubbard model by specifying the

---

[9]The linked library should implement at least the LAPACK-3.4.0 interface.
[10]This flag may trigger compiling issues, in particular, the Intel ifort compiler version 10.1 fails for all optimization levels.

| Element | Suggestion |
|---|---|
| `Precision Green,` `Precision Phase` | Should be found to be *small*, of order $< 10^{-8}$ (see Sec. 2.4) |
| `theta` | Should be *large* enough to guarantee convergence to ground state |
| `dtau` | Should be set *small* enough to limit Trotter errors |
| `Nwrap` | Should be set *small* enough to keep `Precision`s small |
| `Nsweep` | Should be set *large* enough for bins to be of the order of the autocorrelation time |
| `Nbin` | Should be set *large* enough to provide desired statistics |
| `nskip` | Should be set *large* enough to allow for equilibration ($\sim$ autocorrelation time) |
| `Nrebin` | Can be set to 1 when `Nsweep` is large enough; otherwise, and for testing, larger values can be used |
| Stabilization scheme | Use the default `STAB3` – newest and fastest, if it works for your model; alternatives are: `STAB1` – simplest, for reference only; `STAB2` – with additional normalizations; and `LOG` – for dealing with more extreme scales (see also Tab. 14) |
| Parallelism | For some models and systems, restricting parallelism in your BLAS library can improve performance: for OpenBLAS try setting `OPENBLAS_NUM_THREADS=1` in the shell |
| `ALF_SHM_CHUNK_-` `SIZE_GB` | An environment variable that sets the chunk size in GBs for the memory shared between different MPI processes on the same computing node. By default it is zero (i.e., no sharing), but can be set to, e.g., 1.0 or 2.0 GB or larger if, for instance, the total number of MPI communicators is so large as to trigger MPI error messages. |

Table 17: Rules of thumb for obtaining best results and performance from ALF. It is important to fine tune the parameters to the specific model under consideration and perform sanity checks throughout. Most suggestions can severely impact performance and numerical stability if overdone.

lattice, the hopping, the interaction, the trial wave function (if required), and the observables. Consider the *plain vanilla* Hubbard model written as:

$$\mathcal{H} = -t \sum_{\langle \boldsymbol{i}, \boldsymbol{j} \rangle, \sigma = \uparrow, \downarrow} \left( \hat{c}_{\boldsymbol{i}, \sigma}^{\dagger} \hat{c}_{\boldsymbol{j}, \sigma} + \text{H.c.} \right) - \frac{U}{2} \sum_{\boldsymbol{i}} \left[ \hat{c}_{\boldsymbol{i}, \uparrow}^{\dagger} \hat{c}_{\boldsymbol{i}, \uparrow} - \hat{c}_{\boldsymbol{i}, \downarrow}^{\dagger} \hat{c}_{\boldsymbol{i}, \downarrow} \right]^2 - \mu \sum_{\boldsymbol{i}, \sigma} \hat{c}_{\boldsymbol{i}, \sigma}^{\dagger} \hat{c}_{\boldsymbol{i}, \sigma}. \qquad (132)$$

Here $\langle \boldsymbol{i}, \boldsymbol{j} \rangle$ denotes nearest neighbors. We can make contact with the general form of the Hamiltonian [see Eq. (2)] by setting: $N_{\text{fl}} = 2$, $N_{\text{col}} \equiv \texttt{N\_SUN} = 1$, $M_T = 1$,

$$T_{xy}^{(ks)} = \begin{cases} -t & \text{if } x, y \text{ are nearest neighbors} \\ -\mu & \text{if } x = y \\ 0 & \text{otherwise,} \end{cases} \qquad (133)$$

$M_V = N_{\text{unit-cell}}$, $U_k = \frac{U}{2}$, $V_{xy}^{(k,s=1)} = \delta_{x,y}\delta_{x,k}$, $V_{xy}^{(k,s=2)} = -\delta_{x,y}\delta_{x,k}$, $\alpha_{ks} = 0$ and $M_I = 0$. The coupling of the HS fields to the $z$-component of the magnetization breaks the SU(2) spin

symmetry. Nevertheless, the $z$-component of the spin remains a good quantum number such that the imaginary-time propagator – for a given HS field – is block diagonal in this quantum number. This corresponds to the flavor index running from 1 to 2, labeling spin up and spin down degrees of freedom. We note that in this formulation the hopping matrix can be flavor dependent such that a Zeeman magnetic field can be introduced. If the chemical potential is set to zero, this will not generate a negative sign problem [74, 139, 140]. The code that we describe below can be found in the submodule `Prog/Hamiltonians/Hamiltonian_plain_vanilla_hubbard_smod.F90`. This file may be a good starting point for implementing a new model Hamiltonian.

## 7.1   Setting the Hamiltonian: `Ham_set`

The main program will call the subroutine `Ham_set` in the submodule `Hamiltonian_plain_vanilla_hubbard_smod.F90`. The latter subroutine defines the public variables

```
Type(Operator),     dimension(:,:), allocatable :: Op_V ! Interaction
Type(Operator),     dimension(:,:), allocatable :: Op_T ! Hopping
Type(WaveFunction), dimension(:),   allocatable :: WF_L ! Left trial wave function
Type(WaveFunction), dimension(:),   allocatable :: WF_R ! Right trial wave function
Type(Fields)        :: nsigma                           ! Fields
Integer             :: Ndim                             ! Number of sites
Integer             :: N_FL                             ! number of flavors
Integer             :: N_SUN                            ! Number of colors
Integer             :: Ltrot                            ! Total number of trotter silces
Integer             :: Thtrot                           ! Number of trotter slices
                                                        ! reserved for projection
Logical             :: Projector                        ! Projector code
Integer             :: Group_Comm                       ! Group communicator for MPI
Logical             :: Symm                             ! Symmetric trotter
```

which specify the model. The routine `Ham_set` will first read the parameter file `parameters` (see Sec. 5.7.1); then set the lattice: `Call Ham_latt`; set the hopping: `Call Ham_hop`; set the interaction: `call Ham_V`; and if required, set the trial wave function: `call Ham_trial`.

## 7.2   The lattice: `Ham_latt`

The routine, which sets the square lattice, reads:

```
a1_p(1) = 1.0  ; a1_p(2) = 0.d0
a2_p(1) = 0.0  ; a2_p(2) = 1.d0
L1_p    = dble(L1)*a1_p
L2_p    = dble(L2)*a2_p
Call Make_Lattice(L1_p, L2_p, a1_p, a2_p, Latt)
Latt_unit%Norb = 1
Latt_unit%N_coord = 2
allocate(Latt_unit%Orb_pos_p(Latt_unit%Norb,2))
Latt_unit%Orb_pos_p(1, :) = [0.d0, 0.d0]
Ndim = Latt%N*Latt_unit\%Norb
```

In its last line, the routine sets the total number of single particle states per flavor and color: `Ndim = Latt%N*Latt_unit%Norb`.

### 7.3 The hopping: Ham_hop

The hopping matrix is implemented as follows. We allocate an array of dimension $1 \times N_{\mathrm{fl}}$ of type operator called Op_T and set the dimension for the hopping matrix to $N = N_{\mathrm{dim}}$. The operator allocation and initialization is performed by the subroutine Op_make:

```
do nf = 1,N_FL
   call Op_make(Op_T(1,nf),Ndim)
enddo
```

Since the hopping does not break down into small blocks, we have $\boldsymbol{P} = \mathbb{1}$ and

```
Do nf = 1, N_FL
  Do i = 1,Latt%N
     Op_T(1,nf)%P(i) = i
  Enddo
Enddo
```

We set the hopping matrix with

```
Do nf = 1, N_FL
   Do I = 1, Latt%N
      Ix = Latt%nnlist(I,1,0)
      Iy = Latt%nnlist(I,0,1)
      Op_T(1,nf)%O(I,  Ix) = cmplx(-Ham_T,    0.d0, kind(0.D0))
      Op_T(1,nf)%O(Ix, I ) = cmplx(-Ham_T,    0.d0, kind(0.D0))
      Op_T(1,nf)%O(I,  Iy) = cmplx(-Ham_T,    0.d0, kind(0.D0))
      Op_T(1,nf)%O(Iy, I ) = cmplx(-Ham_T,    0.d0, kind(0.D0))
      Op_T(1,nf)%O(I,  I ) = cmplx(-Ham_chem, 0.d0, kind(0.D0))
   Enddo
   Op_T(1,nf)%g     = -Dtau
   Op_T(1,nf)%alpha = cmplx(0.d0,0.d0, kind(0.D0))
   Call Op_set(Op_T(1,nf))
Enddo
```

Here, the integer function Latt%nnlist(I,n,m) is defined in the lattice module and returns the index of the lattice site $\boldsymbol{I} + n\boldsymbol{a}_1 + m\boldsymbol{a}_2$. Note that periodic boundary conditions are already taken into account. The hopping parameter Ham_T, as well as the chemical potential Ham_chem are read from the parameter file. To completely define the hopping we further set: Op_T(1,nf)%g = -Dtau , Op_T(1,nf)%alpha = cmplx(0.d0,0.d0, kind(0.D0)) and call the routine Op_set(Op_T(1,nf)) so as to generate the unitary transformation and eigenvalues as specified in Table 2. Recall that for the hopping, the variable Op_set(Op_T(1,nf))%type takes its default value of 0. Finally, note that, although a checkerboard decomposition is not used here, it can be implemented by considering a larger number of sparse hopping matrices.

### 7.4 The interaction: Ham_V

To implement the interaction, we allocate an array of Operator type. The array is called Op_V and has dimensions $N_{\mathrm{dim}} \times N_{\mathrm{fl}} = N_{\mathrm{dim}} \times 2$. We set the dimension for the interaction term to $N = 1$, and allocate and initialize this array of type Operator by repeatedly calling the subroutine Op_make:

```
Allocate(Op_V(Ndim,N_FL))
do nf = 1,N_FL
   do i  = 1, Ndim
      Call Op_make(Op_V(i,nf), 1)
```

```
      enddo
   enddo
Do nf = 1,N_FL
   X = 1.d0
   if (nf == 2)  X = -1.d0
   Do i = 1,Ndim
      nc = nc + 1
      Op_V(i,nf)%P(1)   = I
      Op_V(i,nf)%O(1,1) = cmplx(1.d0, 0.d0, kind(0.D0))
      Op_V(i,nf)%g      = X*SQRT(CMPLX(DTAU*ham_U/2.d0, 0.D0, kind(0.D0)))
      Op_V(i,nf)%alpha  = cmplx(0.d0, 0.d0, kind(0.D0))
      Op_V(i,nf)%type   = 2
      Call Op_set( Op_V(i,nf) )
   Enddo
Enddo
```

The code above makes it explicit that there is a sign difference between the coupling of the HS field in the two flavor sectors.

## 7.5 The trial wave function: `Ham_Trial`

As argued in Sec. 3.1, it is useful to generate the trial wave function from a non-interacting trial Hamiltonian. Here we will use the same left and right flavor-independent trial wave functions that correspond to the ground state of:

$$\hat{H}_T = -t \sum_{\boldsymbol{i}} \left[ \left(1 + (-1)^{i_x+i_y}\delta\right) \hat{c}_{\boldsymbol{i}}^{\dagger}\hat{c}_{\boldsymbol{i}+\boldsymbol{a}_x} + (1-\delta)\,\hat{c}_{\boldsymbol{i}}^{\dagger}\hat{c}_{\boldsymbol{i}+\boldsymbol{a}_y} + \text{H.c.}\right] \equiv \sum_{\boldsymbol{i},\boldsymbol{j}} \hat{c}_{\boldsymbol{i}}^{\dagger}h_{\boldsymbol{i},\boldsymbol{j}}\hat{c}_{\boldsymbol{i}}. \tag{134}$$

For the half-filled case, the dimerization $\delta = 0^+$ opens up a gap at half-filling, thus generating the desired non-degenerate trial wave function that has the same symmetries (particle-hole for instance) as the trial Hamiltonian.

Diagonalization of $h_{\boldsymbol{i},\boldsymbol{j}}$, $U^{\dagger}hU = \text{Diag}\,(\epsilon_1, \cdots, \epsilon_{N_{\text{dim}}})$ with $\epsilon_i < \epsilon_j$ for $i < j$, allows us to define the trial wave function. In particular, for the half-filled case, we set

```
Do s = 1, N_fl
   Do x = 1,Ndim
      Do n = 1, N_part
         WF_L(s)%P(x,n)  =  U_{x,n}
         WF_R(s)%P(x,n)  =  U_{x,n}
      Enddo
   Enddo
Enddo
```

with `N_part = Ndim/2`. The variable `Degen` belonging to the `WaveFunction` type is given by `Degen`$= \epsilon_{N_{\text{Part}}+1} - \epsilon_{N_{\text{Part}}}$. This quantity should be greater than zero for non-degenerate trial wave functions.

## 7.6 Observables

At this point, all the information for starting the simulation has been provided. The code will sequentially go through the operator list `Op_V` and update the fields. Between time slices `LOBS_ST` and `LOBS_EN` the main program will call the routine `Obser(GR,Phase,Ntau)`, which handles equal-time correlation functions, and, if `Ltau=1`, the routine `ObserT(NT, GT0,GOT,GOO,GTT, PHASE)` which handles imaginary-time displaced correlation functions.

Both `Obser` and `ObserT` should be provided by the user, who can either implement themselves the observables they want to compute or use the predefined structures of Chap. 8. Here we describe how to proceed in order to define an observable.

### 7.6.1 Allocating space for the observables: `Alloc_obs(Ltau)`

For four scalar or vector observables, the user will have to declare the following:

```
Allocate ( Obs_scal(4) )
Do I = 1,Size(Obs_scal,1)
   select case (I)
   case (1)
      N = 2;  Filename ="Kin"
   case (2)
      N = 1;  Filename ="Pot"
   case (3)
      N = 1;  Filename ="Part"
   case (4)
      N = 1,  Filename ="Ener"
   case default
      Write(6,*) ' Error in Alloc_obs '
   end select
   Call Obser_Vec_make(Obs_scal(I), N, Filename)
enddo
```

Here, `Obs_scal(1)` contains a vector of two observables so as to account for the $x$- and $y$-components of the kinetic energy, for example.

For equal-time correlation functions we allocate `Obs_eq` of type `Obser_Latt`. Here we include the calculation of spin-spin and density-density correlation functions alongside equal-time Green functions.

```
Allocate ( Obs_eq(5) )
Do I = 1,Size(Obs_eq,1)
   select case (I)
   case (1)
      Filename = "Green"
   case (2)
      Filename = "SpinZ"
   case (3)
      Filename = "SpinXY"
   case (4)
      Filename = "SpinT"
   case (5)
      Filename = "Den"
   case default
      Write(6,*) "Error in Alloc_obs"
   end select
   Nt = 1
   Channel = "--"
   Call Obser_Latt_make(Obs_eq(I), Nt, Filename, Latt, Latt_unit, Channel, dtau)
Enddo
```

Be aware that `Obser_Latt_make` does not copy the Bravais lattice `Latt` and unit cell `Latt_unit`, but links them through pointers to be more memory efficient. One can have different lattices attached to different observables by declaring additional instances of `Type(Lattice)`

and `Type(Unit_cell)`. For equal-time correlation functions, we set `Nt = 1` and `Channel` specification is not necessary.

If `Ltau = 1`, then the code allocates space for time displaced quantities. The same structure as for equal-time correlation functions is used, albeit with `Nt = Ltrot + 1` and the channel should be set. Whith `Channel="PH"`, for instance, the analysis algorithm assumes the observable to be particle-hole symmetric. For more details on this parameter, see Sec. 10.

At the beginning of each bin, the main program will set the bin observables to zero by calling the routine `Init_obs(Ltau)`. The user does not have to edit this routine.

### 7.6.2  Measuring equal-time observables: `Obser(GR,Phase,Ntau)`

Having allocated the necessary memory, we proceed to define the observables. The equal-time Green function,

$$\texttt{GR(x,y},\sigma) = \langle \hat{c}_{x,\sigma} \hat{c}^{\dagger}_{y,\sigma} \rangle, \tag{135}$$

the phase factor `phase` [Eq. (122)], and time slice `Ntau` are provided by the main program.

Here, $x$ and $y$ label both unit cell as well as the orbital within the unit cell. For the Hubbard model described here, $x$ corresponds to the unit cell. The Green function does not depend on the color index, and is diagonal in flavor. For the SU(2) symmetric implementation there is only one flavor, $\sigma = 1$ and the Green function is independent on the spin index. This renders the calculation of the observables particularly easy.

An explicit calculation of the potential energy $\langle U \sum_{i} \hat{n}_{i,\uparrow} \hat{n}_{i,\downarrow} \rangle$ reads

```
Obs_scal(2)%N        = Obs_scal(2)%N + 1
Obs_scal(2)%Ave_sign = Obs_scal(2)%Ave_sign + Real(ZS,kind(0.d0))
Do i = 1,Ndim
  Obs_scal(2)%Obs_vec(1)= Obs_scal(2)%Obs_vec(1) +(1-GR(i,i,1))*(1-GR(i,i,2))*Ham_U*ZS*ZP
Enddo
```

Here $\texttt{ZS} = \text{sgn}(C)$ [see Eq. (26)], $\texttt{ZP} = \frac{e^{-S(C)}}{\text{Re}[e^{-S(C)}]}$ [see Eq. (122)] and `Ham_U` corresponds to the Hubbard $U$ term.

Equal-time correlations are also computed in this routine. As an explicit example, we consider the equal-time density-density correlation:

$$\langle \hat{n}_{\boldsymbol{i}} \hat{n}_{\boldsymbol{j}} \rangle - \langle \hat{n}_{\boldsymbol{i}} \rangle \langle \hat{n}_{\boldsymbol{j}} \rangle, \tag{136}$$

with

$$\hat{n}_{\boldsymbol{i}} = \sum_{\sigma} \hat{c}^{\dagger}_{\boldsymbol{i},\sigma} \hat{c}_{\boldsymbol{i},\sigma}. \tag{137}$$

For the calculation of such quantities, it is convenient to define:

$$\texttt{GRC(x,y,s)} = \delta_{x,y} - \texttt{GR(y,x,s)} \tag{138}$$

such that `GRC(x,y,s)` corresponds to $\langle\langle \hat{c}^{\dagger}_{x,s} \hat{c}_{y,s} \rangle\rangle$. In the program code, the calculation of the equal-time density-density correlation function looks as follows:

```
Obs_eq(4)%N = Obs_eq(4)%N + 1                ! Even if it is redundant, each observable
                                             ! carries its own counter and sign.
Obs_eq(4)%Ave_sign = Obs_eq(4)%Ave_sign + Real(ZS,kind(0.d0))
Do I = 1,Ndim
   Do J = 1,Ndim
```

```
     imj = latt%imj(I,J)
     Obs_eq(4)%Obs_Latt(imj,1,1,1) =  Obs_eq(4)%Obs_Latt(imj,1,1,1) + &
                   &    ( (GRC(I,I,1)+GRC(I,I,2)) * (GRC(J,J,1)+GRC(J,J,2))      + &
                   &       GRC(I,J,1)*GR(I,J,1)   +  GRC(I,J,2)*GR(I,J,2)  ) * ZP * ZS
   Enddo
   Obs_eq(4)%Obs_Latt0(1) = Obs_eq(4)%Obs_Latt0(1) + (GRC(I,I,1)+GRC(I,I,2))*ZP*ZS
Enddo
```

At the end of each bin the main program calls the routine `Pr_obs(LTAU)`. This routine appends the result for the current bins to the corresponding file, with the appropriate suffix.

### 7.6.3 Measuring time-displaced observables: `ObserT(NT, GT0, G0T, G00, GTT, PHASE)`

This subroutine is called by the main program at the beginning of each sweep, provided that `LTAU` is set to 1. The variable `NT` runs from `0` to `Ltrot` and denotes the imaginary time difference. For a given time displacement, the main program provides:

$$
\begin{aligned}
\text{GT0(x,y,s)} &= \langle\langle \hat{c}_{x,s}(Nt\Delta\tau)\hat{c}_{y,s}^{\dagger}(0)\rangle\rangle = \langle\langle \mathcal{T}\hat{c}_{x,s}(Nt\Delta\tau)\hat{c}_{y,s}^{\dagger}(0)\rangle\rangle \\
\text{G0T(x,y,s)} &= -\langle\langle \hat{c}_{y,s}^{\dagger}(Nt\Delta\tau)\hat{c}_{x,s}(0)\rangle\rangle = \langle\langle \mathcal{T}\hat{c}_{x,s}(0)\hat{c}_{y,s}^{\dagger}(Nt\Delta\tau)\rangle\rangle \\
\text{G00(x,y,s)} &= \langle\langle \hat{c}_{x,s}(0)\hat{c}_{y,s}^{\dagger}(0)\rangle\rangle \\
\text{GTT(x,y,s)} &= \langle\langle \hat{c}_{x,s}(Nt\Delta\tau)\hat{c}_{y,s}^{\dagger}(Nt\Delta\tau)\rangle\rangle .
\end{aligned}
\tag{139}
$$

In the above we have omitted the color index since the Green functions are color independent. The time-displaced spin-spin correlations $4\langle\langle \hat{S}_{\boldsymbol{i}}^{z}(\tau)\hat{S}_{\boldsymbol{j}}^{z}(0)\rangle\rangle$ are then given by:

$$
\begin{aligned}
4\langle\langle \hat{S}_{\boldsymbol{i}}^{z}(\tau)\hat{S}_{\boldsymbol{j}}^{z}(0)\rangle\rangle = (\text{GTT(I,I,1)} - \text{GTT(I,I,2)}) * (\text{G00(J,J,1)} - \text{G00(J,J,2)}) \\
- \text{G0T(J,I,1)} * \text{GT0(I,J,1)} - \text{G0T(J,I,2)} * \text{GT0(I,J,2)}
\end{aligned}
\tag{140}
$$

The handling of time-displaced correlation functions is identical to that of equal-time correlations.

## 7.7 Numerical precision

Information on the numerical stability is included in the following lines of the corresponding file `info`. For a *short* simulation on a $4 \times 4$ lattice at $U/t = 4$ and $\beta t = 10$ we obtain

```
Precision Green  Mean, Max :    5.0823874429126405E-011  5.8621144596315844E-006
Precision Phase  Max       :    0.0000000000000000
Precision tau    Mean, Max :    1.5929357848647394E-011  1.0985132530727526E-005
```

showing the mean and maximum difference between the *wrapped* and from scratched computed equal and time-displaced Green functions [6]. A stable code should produce results where the mean difference is smaller than the stochastic error. The above example shows a very stable simulation since the Green function is of order one.

## 7.8 Running the code and testing

To test the code, one can carry out high precision simulations. After compilation, the executable `ALF.out` is found in the directory `$ALF_DIR/Prog/` and can be run from any directory containing the files `parameters` and `seeds` (See Sec. 5.7).

Alternatively, as we do bellow, it may be convenient to use `pyALF` to compile and run the code, especially when using one of the scripts or notebooks available.

**One-dimensional case**

The `pyALF` python script `Hubbard_Plain_Vanilla.py` runs the projective version of the code for the four-site Hubbard model. At $\theta t = 10$, $\Delta \tau t = 0.05$ with the symmetric Trotter decomposition, we obtain after 40 bins of 2000 sweeps each the total energy:

$$\langle \hat{H} \rangle = -2.103750 \pm 0.004825,$$

and the exact result is

$$\langle \hat{H} \rangle_{\texttt{Exact}} = -2.100396.$$

**Two-dimensional case**

For the two-dimensional case, with similar parameters, we obtain the results listed in Table 18. The exact results stem from Ref. [141] and the slight discrepancies from the exact results can

|  | QMC | Exact |
|---|---|---|
| Total energy | -13.618 $\pm$ 0.002 | -13.6224 |
| $Q = (\pi, \pi)$ spin correlations | 3.630 $\pm$ 0.006 | 3.64 |

Table 18: Test results for the `Hubbard_Plain_Vanilla` code on a two-dimensional lattice with default parameters.

be assigned to the finite value of $\Delta \tau$. Note that all the simulations were carried out with the default value of the Hubbard interaction, $U/t = 4$.

# 8 Predefined Structures

The ALF package includes predefined structures, which the user can combine together or use as templates for defining new ones. Using the data types defined in the Sec. 5 the following modules are available:

- lattices and unit cells – `Predefined_Latt_mod.F90`

- hopping Hamiltonians – `Predefined_Hop_mod.F90`

- interaction Hamiltonians – `Predefined_Int_mod.F90`

- observables – `Predefined_Obs_mod.F90`

- trial wave functions – `Predefined_Trial_mod.F90`

which we describe in the remaining of this section.

## 8.1 Predefined lattices

The types `Lattice` and `Unit_cell`, described in Section 5.3, allow us to define arbitrary one- and two-dimensional Bravais lattices. The subroutine `Predefined_Latt` provides some of the most common lattices, as described bellow.

The subroutine is called as:

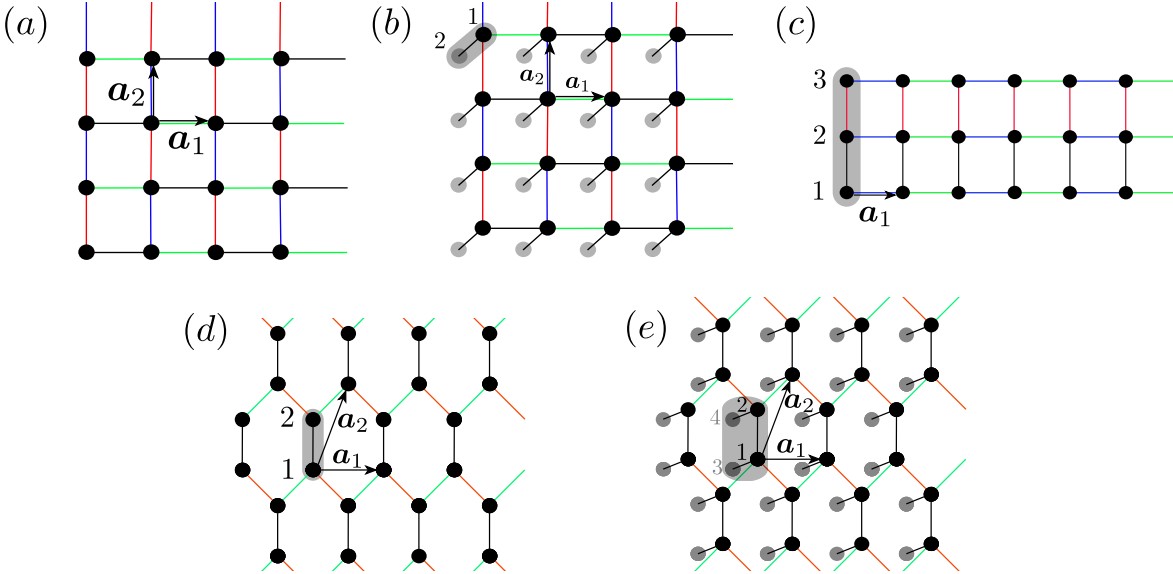

Figure 5: Predefined lattices in ALF: (a) square, (b) bilayer square, (c) 3-leg ladder, (d) honeycomb, and (e) bilayer honeycomb. Nontrivial unit cells are shown as gray regions, while gray sites belong to the second layer in bilayer systems. The links between the orbitals denote the hopping matrix elements and we have assumed, for the purpose of the plot, the absence of hopping in the second layer for bilayer systems. The color coding of the links denotes the checkerboard decomposition.

```
Predefined_Latt(Lattice_type, L1, L2, Ndim, List, Invlist, Latt, Latt_Unit)
```

which returns a lattice of size L1×L2 of the given `Lattice_type`, as detailed in Table 19. Notice that the orbital position `Latt_Unit%Orb_pos_p(1,:)` is set to zero unless otherwise specified.

| Argument | Type | Role | Description |
|---|---|---|---|
| Lattice_type | char | Input | Lattice configuration, which can take the values: `Square` <br> - `Honeycomb` <br> - `Pi_Flux` (deprecated) <br> - `N_leg_ladder` <br> - `Bilayer_square` <br> - `Bilayer_honeycomb` |
| L1, L2 | int | Input | Lattice sizes (set L2=1 for 1D lattices) |
| Ndim | int | Output | Total number of orbitals |
| List | int | Output | For every site index I $\in [1, \texttt{Ndim}]$, stores the corresponding lattice position, `List(I,1)`, and the (local) orbital index, `List(I,2)` |
| Invlist | int | Output | For every `lattice_position` $\in [1, \texttt{Latt\%N}]$ and `orbital` $\in [1, \texttt{Norb}]$ stores the corresponding site index `I(lattice_position,orbital)` |
| Latt | Lattice | Output | Sets the lattice |
| Latt_Unit | Unit_cell | Output | Sets the unit cell |

Table 19: Arguments of the subroutine `Predefined_Latt`. Note that the `Pi_Flux` lattice is deprecated, since it can be emulated with the Square lattice with half a flux quanta piercing each plaquette.

In order to easily keep track of the orbital and unit cell, `List` and `Invlist` make use of a super-index, defined as shown below:

```
nc = 0                                  ! Super-index labeling unit cell and orbital
Do I = 1,Latt%N                         ! Unit-cell index
   Do no = 1,Norb                       ! Orbital index
      nc = nc + 1
      List(nc,1) = I                    ! Unit-cell of super index nc
      List(nc,2) = no                   ! Orbital of super index nc
      Invlist(I,no) = nc                ! Super-index for given unit cell and orbital
   Enddo
Enddo
```

With the above-defined lists one can run through all the orbitals while keeping track of the unit-cell and orbital index. We note that when translation symmetry is completely absent one can work with a single unit cell, and the number of orbitals will then correspond to the number of lattice sites.

### 8.1.1 Square lattice, Fig. 5(a)

The choice `Lattice_type = "Square"` sets $\boldsymbol{a}_1 = (1, 0)$ and $\boldsymbol{a}_2 = (0, 1)$ and for an $L_1 \times L_2$ lattice $\boldsymbol{L}_1 = L_1 \boldsymbol{a}_1$ and $\boldsymbol{L}_2 = L_2 \boldsymbol{a}_2$:

```
Latt_Unit%N_coord   = 2
Latt_Unit%Norb      = 1
Latt_Unit%Orb_pos_p(1,:) = 0.d0
a1_p(1) =  1.0  ; a1_p(2) =  0.d0
a2_p(1) =  0.0  ; a2_p(2) =  1.d0
L1_p    = dble(L1)*a1_p
L2_p    = dble(L2)*a2_p
Call Make_Lattice( L1_p, L2_p, a1_p, a2_p, Latt )
```

Also, the number of orbitals per unit cell is given by `NORB=1` such that $N_{\text{dim}} \equiv N_{\text{unit-cell}} \cdot \texttt{NORB} = \texttt{Latt\%N} \cdot \texttt{NORB}$, since $N_{\text{unit-cell}} = \texttt{Latt\%N}$.

### 8.1.2 Bilayer Square lattice, Fig. 5(b)

The `"Bilayer_square"` configuration sets:

```
Latt_Unit%Norb      = 2
Latt_Unit%N_coord   = 2
do no = 1,2
   Latt_Unit%Orb_pos_p(no,1) = 0.d0
   Latt_Unit%Orb_pos_p(no,2) = 0.d0
   Latt_Unit%Orb_pos_p(no,3) = real(1-no,kind(0.d0))
enddo
Call Make_Lattice( L1_p, L2_p, a1_p, a2_p, Latt )
Latt%a1_p(1) =  1.0  ; Latt%a1_p(2) =  0.d0
Latt%a2_p(1) =  0.0  ; Latt%a2_p(2) =  1.d0
Latt%L1_p    = dble(L1)*a1_p
Latt%L2_p    = dble(L2)*a2_p
```

### 8.1.3 N-leg Ladder lattice, Fig. 5(c)

The `"N_leg_ladder"` configuration sets:

```
Latt_Unit%Norb      = L2
Latt_Unit%N_coord   = 1
do no = 1,L2
   Latt_Unit%Orb_pos_p(no,1) = 0.d0
   Latt_Unit%Orb_pos_p(no,2) = real(no-1,kind(0.d0))
enddo
a1_p(1) =  1.0   ; a1_p(2) =  0.d0
a2_p(1) =  0.0   ; a2_p(2) =  1.d0
L1_p    = dble(L1)*a1_p
L2_p    =           a2_p
Call Make_Lattice( L1_p, L2_p, a1_p, a2_p, Latt )
```

### 8.1.4 Honeycomb lattice, Fig. 5(d)

In order to carry out simulations on the Honeycomb lattice, which is a triangular Bravais lattice with two orbitals per unit cell, choose `Lattice_type="Honeycomb"`, which sets

```
a1_p(1) =  1.D0    ; a1_p(2) =  0.d0
a2_p(1) =  0.5D0   ; a2_p(2) =  sqrt(3.D0)/2.D0
L1_p    =  Dble(L1) * a1_p
L2_p    =  dble(L2) * a2_p
Call Make_Lattice( L1_p, L2_p, a1_p, a2_p, Latt )
Latt_Unit%Norb    = 2
Latt_Unit%N_coord = 3
Latt_Unit%Orb_pos_p(1,:) = 0.d0
Latt_Unit%Orb_pos_p(2,:) = (a2_p(:) - 0.5D0*a1_p(:)) * 2.D0/3.D0
```

The coordination number of this lattice is `N_coord=3` and the number of orbitals per unit cell, `NORB=2`. The total number of orbitals is therefore $N_{\mathrm{dim}}$=`Latt%N*NORB`.

### 8.1.5   Bilayer Honeycomb lattice, Fig. 5(e)

The `"Bilayer_honeycomb"` configuration sets:

```
Latt_Unit%Norb     = 4
Latt_Unit%N_coord  = 3
Latt_unit%Orb_pos_p = 0.d0
do n = 1,2
   Latt_Unit%Orb_pos_p(1,n) = 0.d0
   Latt_Unit%Orb_pos_p(2,n) = (a2_p(n) - 0.5D0*a1_p(n)) * 2.D0/3.D0
   Latt_Unit%Orb_pos_p(3,n) = 0.d0
   Latt_Unit%Orb_pos_p(4,n) = (a2_p(n) - 0.5D0*a1_p(n)) * 2.D0/3.D0
enddo
Latt_Unit%Orb_pos_p(3,3) = -1.d0
Latt_Unit%Orb_pos_p(4,3) = -1.d0
a1_p(1) =  1.D0    ; a1_p(2) =  0.d0
a2_p(1) =  0.5D0   ; a2_p(2) =  sqrt(3.D0)/2.D0
L1_p    =  dble(L1)*a1_p
L2_p    =  dble(L2)*a2_p
Call Make_Lattice( L1_p, L2_p, a1_p, a2_p, Latt )
```

### 8.1.6   $\pi$-Flux lattice (deprecated)

The `"Pi_Flux"` lattice has been deprecated, since it can be emulated with the Square lattice with half a flux quanta piercing each plaquette. Nonetheless, the configuration is still available, and sets:

```
Latt_Unit%Norb    = 2
Latt_Unit%N_coord = 4
a1_p(1) =  1.D0    ; a1_p(2) =   1.d0
a2_p(1) =  1.D0    ; a2_p(2) =  -1.d0
Latt_Unit%Orb_pos_p(1,:) = 0.d0
Latt_Unit%Orb_pos_p(2,:) = (a1_p(:) - a2_p(:))/2.d0
L1_p    =  dble(L1) * (a1_p - a2_p)/2.d0
L2_p    =  dble(L2) * (a1_p + a2_p)/2.d0
Call Make_Lattice( L1_p, L2_p, a1_p, a2_p, Latt )
```

## 8.2   Generic hopping matrices on Bravais lattices

The module `Predefined_Hopping` provides a generic way to specify a hopping matrix on a multi-orbital Bravais lattice. The only assumption that we make is translation symmetry. We

allow for twisted boundary conditions in the $\boldsymbol{L}_1$ and $\boldsymbol{L}_2$ lattice directions. The twist is given by `Phi_X` and `Phi_Y` respectively. If the flag `bulk=.true.`, then the twist is implemented with a vector potential. Otherwise, if `bulk=.false.`, the twist is imposed at the boundary. The routine also accounts for the inclusion of a total number of `N_Phi` flux quanta traversing the lattice. All phase factors mentioned above can be flavor dependent. Finally, the checkerboard decomposition can also be specified in this module.

### 8.2.1 Setting up the hopping matrix: the `Hopping_Matrix_type`

All information for setting up a generic hopping matrix on a lattice, including the checkerboard decomposition, is specified in the `Hopping_Matrix_type` type, which we describe in the remaining of this section. The information stored in this type (see Table 20) fully defines the array of operator type `OP_T` that accounts for the single particle propagation in one time step, from which the kinetic energy can be derived as well.

**Generic hopping matrices**

The generic Hopping Hamiltonian reads:

$$\hat{H}_T = \sum_{(i,\delta),(j,\delta'),s,\sigma} T^{(s)}_{(i,\delta),(j,\delta')} \hat{c}^\dagger_{(i,\delta),s,\sigma} e^{\frac{2\pi i}{\Phi_0} \int_{i+\delta}^{j+\delta'} \boldsymbol{A}^{(s)}(\boldsymbol{l})d\boldsymbol{l}} \hat{c}_{(j,\delta'),s,\sigma} \tag{141}$$

with boundary conditions

$$\hat{c}^\dagger_{(i+L_i,\delta),s,\sigma} = e^{-2\pi i \frac{\Phi^{(s)}_i}{\Phi_0}} e^{\frac{2\pi i}{\Phi_0} \chi^{(s)}_{L_i}(i+\delta)} \hat{c}^\dagger_{(i,\delta),s,\sigma}. \tag{142}$$

Here $i$ labels the unit cell and $\delta$ the orbital. Both the twist and vector potential can have a flavor dependency. These and the other components of the generic Hopping Hamiltonian are described bellow. For now onwards we will mostly omit the flavor index $s$.

**Phase factors**. The vector potential accounts for an orbital magnetic field in the $z$ direction that is implemented in the Landau gauge: $\boldsymbol{A}(\boldsymbol{x}) = -B(y,0,0)$ with $\boldsymbol{x} = (x,y,z)$. $\Phi_0$ corresponds to the flux quanta and the scalar function $\chi$ is defined through:

$$\boldsymbol{A}(\boldsymbol{x} + \boldsymbol{L}_i) = \boldsymbol{A}(\boldsymbol{x}) + \boldsymbol{\nabla}\chi_{L_i}(\boldsymbol{x}). \tag{143}$$

Provided that the bare hopping Hamiltonian, $T$ (i.e., without phases, see Eq. (149)), is invariant under lattice translations, $\hat{H}_T$ commutes with magnetic translations that satisfy the algebra:

$$\hat{T}_{\boldsymbol{a}}\hat{T}_{\boldsymbol{b}} = e^{\frac{2\pi i}{\Phi_0} \boldsymbol{B}\cdot(\boldsymbol{a}\times\boldsymbol{b})} \hat{T}_{\boldsymbol{b}}\hat{T}_{\boldsymbol{a}}. \tag{144}$$

On the torus, the uniqueness of the wave functions requires that $\hat{T}_{\boldsymbol{L}_1}\hat{T}_{\boldsymbol{L}_2} = \hat{T}_{\boldsymbol{L}_2}\hat{T}_{\boldsymbol{L}_1}$ such that

$$\frac{\boldsymbol{B}\cdot(\boldsymbol{a}\times\boldsymbol{b})}{\Phi_0} = N_\Phi \tag{145}$$

with $N_\Phi$ an integer. The variable `N_Phi`, specified in the parameter file, denotes the number of flux quanta piercing the lattice. The variables `Phi_X` and `Phi_Y` also in the parameter file denote the twists – in units of the flux quanta – along the $\boldsymbol{L}_1$ and $\boldsymbol{L}_2$ directions. There

are gauge equivalent ways to insert the twist in the boundary conditions. In the above we have inserted the twist as a boundary condition such that for example setting `Phi_1=0.5` corresponds to anti-periodic boundary conditions along the $L_1$ axis. Alternatively we can consider the Hamiltonian:

$$\hat{H}_T = \sum_{(i,\delta),(j,\delta'),s,\sigma} T^{(s)}_{(i,\delta),(j,\delta')} \tilde{c}^{\dagger}_{(i,\delta),s,\sigma} e^{\frac{2\pi i}{\Phi_0} \int_{i+\delta}^{j+\delta'} \left( \boldsymbol{A}(\boldsymbol{l}) + \boldsymbol{A}_{\phi} \right) d\boldsymbol{l}} \tilde{c}_{(j,\delta'),s,\sigma} \tag{146}$$

with boundary conditions

$$\tilde{c}^{\dagger}_{(i+L_i,\delta),s,\sigma} = e^{\frac{2\pi i}{\Phi_0} \chi_{L_i}(i+\delta)} \tilde{c}^{\dagger}_{(i,\delta),s,\sigma}. \tag{147}$$

Here

$$\boldsymbol{A}_{\phi} = \frac{\phi_1 |\boldsymbol{a}_1|}{2\pi |\boldsymbol{L}_1|} \boldsymbol{b}_1 + \frac{\phi_2 |\boldsymbol{a}_2|}{2\pi |\boldsymbol{L}_2|} \boldsymbol{b}_2 \tag{148}$$

and $\boldsymbol{b}_i$ corresponds to the reciprocal lattice vectors satisfying $\boldsymbol{a}_i \cdot \boldsymbol{b}_j = 2\pi \delta_{i,j}$. The logical variable `bulk` chooses between these two gauge equivalent ways of inserting the twist angle. If `bulk=.true.` then we use periodic boundary conditions – in the absence of an orbital field – otherwise twisted boundaries are used. The above phase factors are computed in the module function:

```
complex function Generic_hopping(i, no_i, n_1, n_2, no_j, N_Phi, Phi_1, Phi_2, Bulk,
                    Latt, Latt_Unit)
```

which returns the phase factor involved in the hopping of a hole from lattice site $\boldsymbol{i} + \boldsymbol{\delta}_{\mathrm{no}_i}$ to $\boldsymbol{i} + n_1 \boldsymbol{a}_1 + n_2 \boldsymbol{a}_2 + \boldsymbol{\delta}_{\mathrm{no}_j}$. Here $\boldsymbol{\delta}_{\mathrm{no}_i}$ is the position of the $\mathrm{no}_i$ orbital in the unit cell $\boldsymbol{i}$. The information for the phases is encoded in the type `Hopping_matrix_type`.

**The Hopping matrix elements**. The hopping matrix is specified assuming only translation invariance. (The point group symmetry of the lattice can be broken.) That is, we assume that for each flavor index:

$$T^{(s)}_{(\boldsymbol{i},\boldsymbol{\delta}),(\boldsymbol{i}+n_1\boldsymbol{a}_1+n_2\boldsymbol{a}_2,\boldsymbol{\delta}')} = T^{(s)}_{(\boldsymbol{0},\boldsymbol{\delta}),(n_1\boldsymbol{a}_1+n_2\boldsymbol{a}_2,\boldsymbol{\delta}')}. \tag{149}$$

The right hand side of the above equation is given the type `Hopping_matrix_type`.

**The checkerboard decomposition.** Aside from the hopping phases and hopping matrix elements, the `Hopping_matrix_type` type contains information concerning the checkerboard decomposition. In Eq. (72) we wrote the hopping Hamiltonian as:

$$\hat{\mathcal{H}}_T = \sum_{i=1}^{N_T} \sum_{k \in \mathcal{S}_i^T} \hat{T}^{(k)}, \tag{150}$$

with the rule that if $k$ and $k'$ belong to the same set $\mathcal{S}_i^T$ then $\left[ \hat{T}^{(k)}, \hat{T}^{(k')} \right] = 0$. In the checkerboard decomposition, $\hat{T}^{(k)}$ corresponds to hopping on a bond. The checkerboard decomposition depends on the lattice type, as well as on the hopping matrix elements. The required information is stored in `Hopping_matrix_type`. In this data type, `N_FAM` corresponds to the number of sets (or families) ($N_T$ in the above equation). `L_FAM(1:N_FAM)` corresponds

to the number of bonds in the set, and finally, `LIST_FAM(1:N_FAM, 1:max(L_FAM(:)), 2)` contains information concerning the two legs of the bonds. In the checkerboard decomposition, care has to be taken for local terms: each site occurs multiple times in the list of bonds. Since we have postulated translation symmetry, a one-dimensional array, `Multiplicity`, of length given by the number of orbitals per unit cell suffices to encode the required information. Finally, to be able to generate the imaginary time step of length $\Delta\tau$ we have to know by which fraction of $\Delta\tau$ we have to propagate each set. This information is given in the array `Prop_Fam`.

As an example we can consider the three-leg ladder lattice of Figure 5(c). Here the number of sets (or families) `N_FAM` is equal to four, corresponding to the red, green, black and blue bonds. It is clear from the figure that bonds in a given set do not have common legs, so that hopping instances on the bonds of a given set commute. For this three-leg ladder, we see that the middle orbital in a unit cell appears in each set or family. It hence has a multiplicity of four. On the other hand, the top and bottom orbitals have a multiplicity of 3 since they appear in only three of the four sets.

### Usage: the `Hopping_Matrix_type`

There are `N_bonds` hopping matrix elements emanating from a given unit cell, defined so that looping over all of the elements does not overcount the bonds. For each bond, the array `List` contains the full information to define the RHS of Eq. (149). The hopping amplitudes are stored in the array `T` and the local potentials in the array `T_loc` (See Table 20). The `Hopping_Matrix_type` type also contains the information for the checkerboard decomposition.

The data in the `Hopping_matrix_type` type suffices to uniquely define the unit step propagation for the kinetic energy, and for any combinations of the `Checkerboard` and `Symm` options (see Sec. 2.3). The propagation is set through the call:

```
Call Predefined_Hoppings_set_OPT(Hopping_Matrix, List, Invlist, Latt, Latt_unit, Dtau,
                                 Checkerboard, Symm, OP_T)
```

in which the operator array `OP_T(*,N_FL)` is allocated and defined. In the simplest case, where no checkerboard is used, the array's first dimension is unity.

The data in the `Hopping_matrix_type` type equally suffices to compute the kinetic energy. This is carried out in the routine `Predefined_Hoppings_Compute_Kin`.

### 8.2.2   An example: nearest neighbor hopping on the honeycomb lattice

For the honeycomb lattice of Fig. 5(d) the number of bond within and emanating from a unit cell is `N_bonds = 3`. The list array of the `Hopping_matrix_type` reads:

```
list(1,1) = 1;  list(1,2) = 2;  list(1,3) = 0;   list(1,4) =  0 ! Intra unit-cell hopping
list(2,1) = 2;  list(2,2) = 1;  list(2,3) = 0:   list(2,4) =  1 ! Inter unit-cell hopping
list(3,1) = 1;  list(3,2) = 2;  list(3,3) = 1:   list(3,4) = -1 ! Inter unit-cell hopping
T(1) = -1.0;  T(2) = -1.0;  T(3) = -1.0                         ! Hopping
T_loc(1) = 0.0;  T_loc(2) = 0.0                                 ! Chemical potential
```

In the last two lines, we have set the hopping matrix element for each bond to $-1$ and the chemical potential to zero. The fields, can then be specified with the variables `N_phi`, `Phi_x`, `Phi_y`. Setting the twists, `Phi_x`, `Phi_y` to zero and looping over `N_phi` from $1\cdots L^2$ produces the single particle spectrum of Fig. 6(a).

| Variable | Type | Description |
| --- | --- | --- |
| N_bonds | int | Number of hopping matrix elements within and emanating from a unit cell |
| List(N_bonds,4) | int | List($\bullet$,1) $= \delta$ <br> List($\bullet$,2) $= \delta'$ <br> List($\bullet$,3) $= n_1$ <br> List($\bullet$,4) $= n_2$ |
| T(N_bonds) | cmplx | Hopping amplitude |
| T_loc(Norb) | cmplx | On site potentials (e.g., chemical potential, Zeeman field) |
| N_Phi | int | Number of flux quanta piercing the lattice |
| Phi_X | dble | Twist in $\boldsymbol{a}_1$ direction |
| Phi_Y | dble | Twist in $\boldsymbol{a}_2$ direction |
| Bulk | logical | Twist as vector potential (T) or boundary condition (F) |
| N_Fam | int | Number of sets, $N_T$ in Eq. (72) |
| L_Fam(N_FAM) | int | Number of bonds per set $\mathcal{S}^T$ |
| List_Fam(N_FAM,max(L_FAM(:)),2) | int | List_Fam($\bullet,\bullet$,1) = Unit cell <br> List_Fam($\bullet,\bullet$,2) = Bond number |
| Multiplicity(Norb) | int | Number of times a given orbital occurs in the list of bonds |
| Prop_Fam(N_FAM) | dble | The fraction of $\Delta\tau$ with which the set will be propagated |

Table 20: Member variables of the `Hopping_Matrix_type` type.

For the honeycomb lattice the checkerboard decomposition for the nearest neighbor hopping consists of three sets: `N_Fam = 3` each of length corresponding to the number of unit cells. In Fig. 5(d) these sets are denoted by different colors. In the code, the elements of the sets are specified as:

```
do I = 1,Latt%N
   do nf = 1,N_FAM
      List_Fam(nf,I,1) = I  ! Unit cell
      List_Fam(nf,I,2) = nf ! The bond
   enddo
enddo
Multiplicity  = 3
```

Since each site of the honeycomb lattice occurs in the three sets, their multiplicity is equal to 3.

### 8.2.3 Predefined hoppings

The module provides hopping and checkerboard decompositions, defining a `Hopping_Matrix` (an array of length `N_FL` of type `Hopping_Matrix_type`, see Sec. 8.2.1) for each of the following predefined lattices.

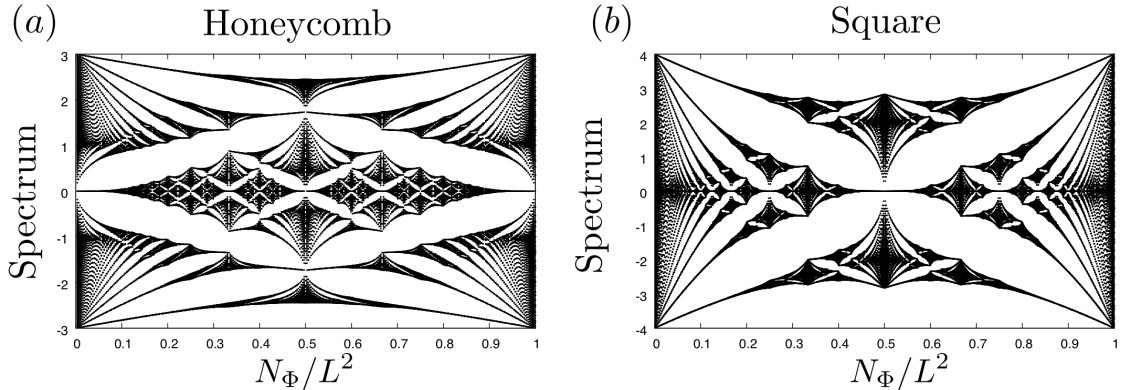

Figure 6: The single particle spectrum of the tight binding model on the honeycomb (a) and square (b) lattices as a function of the flux $N_\Phi$. This corresponds to the well known Hofstadter butterflies.

### Square

The call:

```
Call Set_Default_hopping_parameters_square(Hopping_Matrix, T_vec, Chem_vec, Phi_X_vec,
        Phi_Y_vec, Bulk, N_Phi_vec, N_FL, List, Invlist, Latt, Latt_unit)
```

defines the `Hopping_Matrix` for the square lattice:

$$\hat{H}_T = \sum_{\boldsymbol{i},\sigma,s} \left( \left[ \sum_{\boldsymbol{\delta}=\{\boldsymbol{a}_1,\boldsymbol{a}_2\}} -t^{(s)} \hat{c}^\dagger_{\boldsymbol{i},s,\sigma} e^{\frac{2\pi i}{\Phi_0} \int_{\boldsymbol{i}}^{\boldsymbol{i}+\boldsymbol{\delta}} \boldsymbol{A}^{(s)}(\boldsymbol{l})d\boldsymbol{l}} \hat{c}_{\boldsymbol{i}+\boldsymbol{\delta},s,\sigma} + \text{H.c.} \right] - \mu^{(s)} \hat{c}^\dagger_{\boldsymbol{i},s,\sigma} \hat{c}_{\boldsymbol{i},s,\sigma} \right). \quad (151)$$

The vectors `T_vec` and `Chem_vec` have length `N_FL` and specify the hopping and the chemical potentials, while the vectors `Phi_X_vec`, `Phi_Y_vec` and `N_Phi_vec`, also of length `N_FL`, define the vector potential.

### Honeycomb

The call:

```
Call Set_Default_hopping_parameters_honeycomb(Hopping_Matrix,T_vec, Chem_vec, Phi_X_vec,
        Phi_Y_vec, Bulk, N_Phi_vec, N_FL, List, Invlist, Latt, Latt_unit)
```

defines the `Hopping_Matrix` for the honeycomb lattice:

$$\hat{H}_T = \sum_{\boldsymbol{i},\sigma,s} \left( \sum_{\boldsymbol{\delta}=\{\boldsymbol{\delta}_1,\boldsymbol{\delta}_2,\boldsymbol{\delta}_3\}} -t^{(s)} \hat{c}^\dagger_{\boldsymbol{i},s,\sigma} e^{\frac{2\pi i}{\Phi_0} \int_{\boldsymbol{i}}^{\boldsymbol{i}+\boldsymbol{\delta}} \boldsymbol{A}^{(s)}(\boldsymbol{l})d\boldsymbol{l}} \hat{c}_{\boldsymbol{i}+\boldsymbol{\delta},s,\sigma} + \text{H.c.} \right)$$
$$+ \sum_{\boldsymbol{i},\sigma,s} -\mu^{(s)} \left( \hat{c}^\dagger_{\boldsymbol{i},s,\sigma} \hat{c}_{\boldsymbol{i},s,\sigma} + \hat{c}^\dagger_{\boldsymbol{i}+\boldsymbol{\delta}_1,s,\sigma} \hat{c}_{\boldsymbol{i}+\boldsymbol{\delta}_1,s,\sigma} \right), \quad (152)$$

where the `T_vec` and `Chem_vec` have length `N_FL` and specify the hopping and the chemical potentials, while the vectors `Phi_X_vec`, `Phi_Y_vec` and `N_Phi_vec`, also of length `N_FL`, define

the vector potential. Here $\boldsymbol{i}$ runs over sublattice A, and $\boldsymbol{i} + \boldsymbol{\delta}$ over the three nearest neighbors of site $\boldsymbol{i}$.

### Square bilayer

The call:

```
Call Set_Default_hopping_parameters_Bilayer_square(Hopping_Matrix, T1_vec, T2_vec,
        Tperp_vec, Chem_vec, Phi_X_vec, Phi_Y_vec, Bulk, N_Phi_vec, N_FL, List, Invlist,
        Latt, Latt_unit)
```

defines the `Hopping_Matrix` for the bilayer square lattice:

$$
\hat{H}_T = \sum_{\boldsymbol{i},\sigma,s,n} \left( \left[ \sum_{\boldsymbol{\delta}=\{\boldsymbol{a}_1,\boldsymbol{a}_2\}} -t_n^{(s)} \hat{c}_{\boldsymbol{i},s,\sigma,n}^\dagger e^{\frac{2\pi i}{\Phi_0} \int_{\boldsymbol{i}}^{\boldsymbol{i}+\boldsymbol{\delta}} \boldsymbol{A}^{(s)}(\boldsymbol{l})d\boldsymbol{l}} \hat{c}_{\boldsymbol{i}+\boldsymbol{\delta},s,\sigma,n} + \text{H.c.} \right] - \mu^{(s)} \hat{c}_{\boldsymbol{i},s,\sigma,n}^\dagger \hat{c}_{\boldsymbol{i},s,\sigma,n} \right)
$$
$$
+ \sum_{\boldsymbol{i},\sigma,s} -t_\perp^{(s)} \left( \hat{c}_{\boldsymbol{i},s,\sigma,1}^\dagger \hat{c}_{\boldsymbol{i},s,\sigma,2} + \text{H.c.} \right), \quad (153)
$$

where the additional index $n$ labels the layers.

### Honeycomb bilayer

The call:

```
Call Set_Default_hopping_parameters_Bilayer_honeycomb(Hopping_Matrix, T1_vec, T2_vec,
        Tperp_vec, Chem_vec, Phi_X_vec, Phi_Y_vec, Bulk, N_Phi_vec, N_FL, List, Invlist,
        Latt, Latt_unit)
```

defines the `Hopping_Matrix` for the bilayer honeycomb lattice:

$$
\hat{H}_T = \sum_{\boldsymbol{i},\sigma,s,n} \left( \sum_{\boldsymbol{\delta}=\{\boldsymbol{\delta}_1,\boldsymbol{\delta}_2,\boldsymbol{\delta}_3\}} -t_n^{(s)} \hat{c}_{\boldsymbol{i},s,\sigma,n}^\dagger e^{\frac{2\pi i}{\Phi_0} \int_{\boldsymbol{i}}^{\boldsymbol{i}+\boldsymbol{\delta}} \boldsymbol{A}^{(s)}(\boldsymbol{l})d\boldsymbol{l}} \hat{c}_{\boldsymbol{i}+\boldsymbol{\delta},s,\sigma,n} + \text{H.c.} \right)
$$
$$
+ \sum_{\boldsymbol{i},\sigma,s} -t_\perp^{(s)} \left( \hat{c}_{\boldsymbol{i},s,\sigma,1}^\dagger \hat{c}_{\boldsymbol{i},s,\sigma,2} + \hat{c}_{\boldsymbol{i}+\boldsymbol{\delta}_1,s,\sigma,1}^\dagger \hat{c}_{\boldsymbol{i}+\boldsymbol{\delta}_1,s,\sigma,2} + \text{H.c.} \right)
$$
$$
+ \sum_{\boldsymbol{i},\sigma,s,n} -\mu^{(s)} \left( \hat{c}_{\boldsymbol{i},s,\sigma,n}^\dagger \hat{c}_{\boldsymbol{i},s,\sigma,n} + \hat{c}_{\boldsymbol{i}+\boldsymbol{\delta}_1,s,\sigma,n}^\dagger \hat{c}_{\boldsymbol{i}+\boldsymbol{\delta}_1,s,\sigma,n} \right) \quad (154)
$$

Here, the additional index $n$ labels the layer. $\boldsymbol{i}$ runs over the unit cells and $\boldsymbol{\delta} = \{\boldsymbol{\delta}_1,\boldsymbol{\delta}_2,\boldsymbol{\delta}_3\}$ over the three nearest neighbors.

### N-leg ladder

The call:

```
Call Set_Default_hopping_parameters_n_lag_ladder(Hopping_Matrix, T_vec, Tperp_vec,
    Chem_vec, Phi_X_vec, Phi_Y_vec, Bulk, N_Phi_vec, N_FL, List, Invlist, Latt, Latt_unit)
```

defines the `Hopping_Matrix` for the the N-leg ladder lattice:

$$
\hat{H}_T = \sum_{\boldsymbol{i},\sigma,s} \sum_{n=1}^{\texttt{Norb}} \left( -t^{(s)} \hat{c}^\dagger_{\boldsymbol{i},s,\sigma,n} e^{\frac{2\pi i}{\Phi_0} \int_{\boldsymbol{i}}^{\boldsymbol{i}+\boldsymbol{a}_1} \boldsymbol{A}^{(s)}(\boldsymbol{l})d\boldsymbol{l}} \hat{c}_{\boldsymbol{i}+\boldsymbol{a}_1,s,\sigma,n} + \text{H.c.} - \mu^{(s)} \hat{c}^\dagger_{\boldsymbol{i},s,\sigma,n} \hat{c}_{\boldsymbol{i},s,\sigma,n} \right)
$$

$$
+ \sum_{\boldsymbol{i},\sigma,s} \sum_{n=1}^{\texttt{Norb}-1} -t^{(s)}_\perp \left( \hat{c}^\dagger_{\boldsymbol{i}+\boldsymbol{\delta}_1,s,\sigma,n} e^{\frac{2\pi i}{\Phi_0} \int_{(n-1)\boldsymbol{a}_2}^{(n)\boldsymbol{a}_2} \boldsymbol{A}^{(s)}(\boldsymbol{l})d\boldsymbol{l}} \hat{c}_{\boldsymbol{i}+\boldsymbol{\delta}_1,s,\sigma,n+1} + \text{H.c.} \right). \quad (155)
$$

Here, the additional index $n$ defines the orbital. Note that this lattice has open boundary conditions in the $\boldsymbol{a}_2$ direction.

## 8.3 Predefined interaction vertices

In its most general form, an interaction Hamiltonian, expressed in terms of sums of perfect squares, can be written, as presented in Section 1, as a sum of $M_V$ vertices:

$$
\hat{\mathcal{H}}_V = \sum_{k=1}^{M_V} U_k \left\{ \sum_{\sigma=1}^{N_{\text{col}}} \sum_{s=1}^{N_{\text{fl}}} \left[ \left( \sum_{x,y} \hat{c}^\dagger_{x\sigma s} V^{(ks)}_{xy} \hat{c}_{y\sigma s} \right) + \alpha_{ks} \right] \right\}^2 \equiv \sum_{k=1}^{M_V} U_k \left( \hat{V}^{(k)} \right)^2 \quad (4)
$$

$$
\equiv \sum_{k=1}^{M_V} \hat{\mathcal{H}}^{(k)}_V,
$$

which are encoded in one or more variables of type `Operator`, described in Sec. 5.1. We often use arrays of `Operator` type, which should be initialized by repeatedly calling the subroutine `Op_make`.

The module `Predefined_Int_mod.F90` implements some of the most common of such interaction vertices $\hat{\mathcal{H}}^{(k)}_V$, as detailed in the remainder of this section, where we drop the superscript $(k)$ when unambiguous.

### 8.3.1 SU(N) Hubbard interaction

The SU(N) Hubbard interaction on a given site $i$ is given by

$$
\hat{\mathcal{H}}_{V,i} = +\frac{U}{N_{\text{col}}} \left[ \sum_{\sigma=1}^{N_{\text{col}}} \left( \hat{c}^\dagger_{i\sigma} \hat{c}_{i\sigma} - 1/2 \right) \right]^2. \quad (156)
$$

Assuming that no other term in the Hamiltonian breaks the SU(N) color symmetry, then this interaction term conveniently corresponds to a single operator, obtained by calling, for each of the $N_{\text{dim}}$ sites $i$:

```
Call Predefined_Int_U_SUN(OP, I, N_SUN, DTAU, U)
```

which defines:

```
Op%P(1)   = I
Op%O(1,1) = cmplx(1.d0,  0.d0, kind(0.D0))
Op%alpha  = cmplx(-0.5d0,0.d0, kind(0.D0))
Op%g      = SQRT(CMPLX(-DTAU*U/(DBLE(N_SUN)), 0.D0, kind(0.D0)))
Op%type   = 2
```

To relate to Eq. (4), we have $V^{(is)}_{xy} = \delta_{x,y}\delta_{x,i}$, $\alpha_{is} = -\frac{1}{2}$ and $U_k = \frac{U}{N_{\text{col}}}$. Here the flavor index, $s$, plays no role.

### 8.3.2 $M_z$-Hubbard interaction

```
Call Predefined_Int_U_MZ(OP_up, Op_do, I, DTAU, U)
```

The $M_z$-Hubbard interaction is given by

$$\hat{\mathcal{H}}_V = -\frac{U}{2} \sum_i \left[ \hat{c}_{i\uparrow}^\dagger \hat{c}_{i\uparrow} - \hat{c}_{i\downarrow}^\dagger \hat{c}_{i\downarrow} \right]^2, \tag{157}$$

which corresponds to the general form of Eq. (4) by setting: $N_{\mathrm{fl}} = 2$, $N_{\mathrm{col}} \equiv \texttt{N\_SUN} = 1$, $M_V = N_{\mathrm{unit\text{-}cell}}$, $U_k = \frac{U}{2}$, $V_{xy}^{(i,s=1)} = \delta_{x,y}\delta_{x,i}$, $V_{xy}^{(i,s=2)} = -\delta_{x,y}\delta_{x,i}$, and $\alpha_{is} = 0$; and which is defined in the subroutine $\texttt{Predefined\_Int\_U\_MZ}$ by two operators:

```
Op_up%P(1)   = I
Op_up%O(1,1) = cmplx(1.d0, 0.d0, kind(0.D0))
Op_up%alpha  = cmplx(0.d0, 0.d0, kind(0.D0))
Op_up%g      = SQRT(CMPLX(DTAU*U/2.d0, 0.D0, kind(0.D0)))
Op_up%type   = 2

Op_do%P(1)   = I
Op_do%O(1,1) = cmplx(1.d0, 0.d0, kind(0.D0))
Op_do%alpha  = cmplx(0.d0, 0.d0, kind(0.D0))
Op_do%g      = -SQRT(CMPLX(DTAU*U/2.d0, 0.D0, kind(0.D0)))
Op_do%type   = 2
```

### 8.3.3 SU(N) $V$-interaction

```
Call Predefined_Int_V_SUN(OP, I, J, N_SUN, DTAU, V)
```

The interaction term of the generalized t-V model, given by

$$\hat{\mathcal{H}}_{V,i,j} = -\frac{V}{N_{\mathrm{col}}} \left[ \sum_{\sigma=1}^{N_{\mathrm{col}}} \left( \hat{c}_{i\sigma}^\dagger \hat{c}_{j\sigma} + \hat{c}_{j\sigma}^\dagger \hat{c}_{i\sigma} \right) \right]^2, \tag{158}$$

is coded in the subroutine $\texttt{Predefined\_Int\_V\_SUN}$ by a single symmetric operator:

```
Op%P(1)   = I
Op%P(2)   = J
Op%O(1,2) = cmplx(1.d0 ,0.d0, kind(0.D0))
Op%O(2,1) = cmplx(1.d0 ,0.d0, kind(0.D0))
Op%g      = SQRT(CMPLX(DTAU*V/real(N_SUN,kind(0.d0)), 0.D0, kind(0.D0)))
Op%alpha  = cmplx(0.d0, 0.d0, kind(0.D0))
Op%type   = 2
```

### 8.3.4 Fermion-Ising coupling

```
Call Predefined_Int_Ising_SUN(OP, I, J, DTAU, XI)
```

The interaction between the Ising and a fermion degree of freedom, given by

$$\hat{\mathcal{H}}_{V,i,j} = \hat{Z}_{i,j} \xi \sum_{\sigma=1}^{N_{\mathrm{col}}} \left( \hat{c}_{i\sigma}^\dagger \hat{c}_{j\sigma} + \hat{c}_{j\sigma}^\dagger \hat{c}_{i\sigma} \right), \tag{159}$$

where $\xi$ determines the coupling strength, is implemented in the subroutine `Predefined_Int_Ising_SUN`:

```
Op%P(1)   = I
Op%P(2)   = J
Op%O(1,2) = cmplx(1.d0 ,0.d0, kind(0.D0))
Op%O(2,1) = cmplx(1.d0 ,0.d0, kind(0.D0))
Op%g      = cmplx(-DTAU*XI,0.D0,kind(0.D0))
Op%alpha  = cmplx(0d0,0.d0, kind(0.D0))
Op%type   = 1
```

### 8.3.5 Long-Range Coulomb repulsion

```
Call Predefined_Int_LRC(OP, I, DTAU)
```

The Long-Range Coulomb (LRC) interaction can be written as

$$\hat{\mathcal{H}}_V = \frac{1}{N} \sum_{i,j} \left( \hat{n}_i - \frac{N}{2} \right) V_{i,j} \left( \hat{n}_j - \frac{N}{2} \right), \tag{160}$$

where

$$\hat{n}_i = \sum_{\sigma=1}^{N} \hat{c}_{i,\sigma}^\dagger \hat{c}_{i,\sigma} \tag{161}$$

and $i$ corresponds to a super-index labelling the unit cell and orbital.

The code uses the following HS decomposition:

$$e^{-\Delta\tau \hat{H}_{V,k}} = \int \prod_i d\phi_i e^{-\frac{N\Delta\tau}{4}\phi_i V_{i,j}^{-1}\phi_j - \sum_i i\Delta\tau\phi_i\left(\hat{n}_i - \frac{N}{2}\right)}. \tag{162}$$

The above holds only provided that the matrix $V$ is positive definite and the implementation follows Ref. [51].

The LRC interaction is implemented in the subroutine `Predefined_Int_LRC`:

```
Op%P(1)   = I
Op%O(1,1) = cmplx(1.d0 ,0.d0, kind(0.D0))
Op%alpha  = cmplx(-0.5d0,0.d0, kind(0.D0))
Op%g      = cmplx(0.d0 ,DTAU, kind(0.D0))
Op%type   = 3
```

### 8.3.6 $J_z$-$J_z$ interaction

```
Call Predefined_Int_Jz(OP_up, Op_do, I, J, DTAU, Jz)
```

Another predefined vertex is:

$$\hat{\mathcal{H}}_{V,i,j} = -\frac{|J_z|}{2} \left( S_i^z - \text{sgn}\,|J_z| S_j^z \right)^2 = J_z S_i^z S_j^z - \frac{|J_z|}{2}(S_i^z)^2 - \frac{|J_z|}{2}(S_j^z)^2 \tag{163}$$

which, if particle fluctuations are frozen on the $i$ and $j$ sites, then $(S_i^z)^2 = 1/4$ and the interaction corresponds to a $J_z$-$J_z$ ferromagnetic or antiferromagnetic coupling.

The implementation of the interaction in `Predefined_Int_Jz` defines two operators:

```
Op_up%P(1)   = I
Op_up%P(2)   = J
Op_up%O(1,1) = cmplx(1.d0,              0.d0, kind(0.D0))
Op_up%O(2,2) = cmplx(-Jz/Abs(Jz),       0.d0, kind(0.D0))
Op_up%alpha  = cmplx(0.d0,              0.d0, kind(0.D0))
Op_up%g      = SQRT(CMPLX(DTAU*Jz/8.d0, 0.d0, kind(0.D0)))
Op_up%type   = 2

Op_do%P(1)   = I
Op_do%P(2)   = J
Op_do%O(1,1) = cmplx(1.d0,              0.d0, kind(0.d0))
Op_do%O(2,2) = cmplx(-Jz/Abs(Jz),       0.d0, kind(0.d0))
Op_do%alpha  = cmplx(0.d0,              0.d0, kind(0.d0))
Op_do%g      = -SQRT(CMPLX(DTAU*Jz/8.d0, 0.d0, kind(0.d0)))
Op_do%type   = 2
```

## 8.4 Predefined observables

The types `Obser_Vec` and `Obser_Latt` described in Section 5.4 handle arrays of scalar observables and correlation functions with lattice symmetry respectively. The module `Predefined_Obs` provides a set of standard equal-time and time-displaced observables, as described below. It contains procedures and functions. Procedures provide a complete handling of the observable structure. That is, they take care, for example, of incrementing the counter and of the average sign. On the other hand, functions only provide the Wick decomposition result, and the handling of the observable structure is left to the user.

The predefined measurements methods take as input Green functions `GR`, `GT0`, `G0T`, `G00`, and `GTT`, defined in Sec. 7.6.2 and 7.6.3, as well as `N_SUN`, time slice `Ntau`, lattice information, and so on – see Table 21.

### 8.4.1 Equal-time SU(N) spin-spin correlations

A measurement of SU(N) spin-spin correlations can be obtained through:

```
Call Predefined_Obs_eq_SpinSUN_measure(Latt, Latt_unit, List, GR, GRC, N_SUN, ZS, ZP, Obs)
```

If `N_FL = 1` then this routine returns

$$\mathtt{Obs}(\boldsymbol{i}-\boldsymbol{j},n_{\boldsymbol{i}},n_{\boldsymbol{j}}) = \frac{2N}{N^2-1}\sum_{a=1}^{N^2-1}\langle\langle\hat{\boldsymbol{c}}^{\dagger}_{\boldsymbol{i},n_{\boldsymbol{i}}}T^a\hat{\boldsymbol{c}}_{\boldsymbol{i},n_{\boldsymbol{i}}}\;\hat{\boldsymbol{c}}^{\dagger}_{\boldsymbol{j},n_{\boldsymbol{j}}}T^a\hat{\boldsymbol{c}}_{\boldsymbol{j},n_{\boldsymbol{j}}}\rangle\rangle_C, \tag{164}$$

where $T^a$ are the generators of SU(N) satisfying the normalization conditions $\mathrm{Tr}[T^aT^b] = \delta_{a,b}/2$ , $\mathrm{Tr}[T^a] = 0$, $\hat{\boldsymbol{c}}^{\dagger}_{\boldsymbol{j},n_{\boldsymbol{j}}} = \left(\hat{c}^{\dagger}_{\boldsymbol{j},n_{\boldsymbol{j}},1},\cdots,\hat{c}^{\dagger}_{\boldsymbol{j},n_{\boldsymbol{j}},N}\right)$ is an N-flavored spinor, $\boldsymbol{j}$ corresponds to the unit-cell index and $n_{\boldsymbol{j}}$ labels the orbital.

Using Wick's theorem, valid for a given configuration of fields, we obtain

$$\mathtt{Obs} = \frac{2N}{N^2-1}\sum_{a=1}^{N^2-1}\sum_{\alpha,\beta,\gamma,\delta=1}^{N}T^a_{\alpha,\beta}T^a_{\gamma,\delta}\times$$

$$\left(\langle\langle\hat{c}^{\dagger}_{\boldsymbol{i},n_{\boldsymbol{i}},\alpha}\hat{c}_{\boldsymbol{i},n_{\boldsymbol{i}},\beta}\rangle\rangle_C\langle\langle\hat{c}^{\dagger}_{\boldsymbol{j},n_{\boldsymbol{j}},\gamma}\hat{c}_{\boldsymbol{j},n_{\boldsymbol{j}},\delta}\rangle\rangle_C + \langle\langle\hat{c}^{\dagger}_{\boldsymbol{i},n_{\boldsymbol{i}},\alpha}\hat{c}_{\boldsymbol{j},n_{\boldsymbol{j}},\delta}\rangle\rangle_C\langle\langle\hat{c}_{\boldsymbol{i},n_{\boldsymbol{i}},\beta}\hat{c}^{\dagger}_{\boldsymbol{j},n_{\boldsymbol{j}},\gamma}\rangle\rangle_C\right). \tag{165}$$

| Argument | Type | Description |
|---|---|---|
| `Latt` | `Lattice` | Lattice as a variable of type `Lattice`, see Sec. 5.3 |
| `Latt_Unit` | `Unit_cell` | Unit cell as a variable of type `Unit_cell`, see Sec. 5.3 |
| `List(Ndim,2)` | `int` | For every site index `I`, stores the corresponding lattice position, `List(I,1)`, and the (local) orbital index, `List(I,2)`. |
| `NT` | `int` | Imaginary time $\tau$ |
| `GR(Ndim,Ndim,N_FL)` | `cmplx` | Equal-time Green function $\mathtt{GR(i,j,s)} = \langle c_{i,s} c_{j,s}^\dagger \rangle$ |
| `GRC(Ndim,Ndim,N_FL)` | `cmplx` | $\mathtt{GRC(i,j,s)} = \langle c_{i,s}^\dagger c_{j,s} \rangle = \delta_{i,j} - \mathtt{GR(j,i,s)}$ |
| `GT0(Ndim,Ndim,N_FL)` | `cmplx` | Time-displaced Green function $\langle\langle \mathcal{T} \hat{c}_{i,s}(\tau) \hat{c}_{j,s}^\dagger(0) \rangle\rangle$ |
| `G0T(Ndim,Ndim,N_FL)` | `cmplx` | Time-displaced Green function $\langle\langle \mathcal{T} \hat{c}_{i,s}(0) \hat{c}_{j,s}^\dagger(\tau) \rangle\rangle$ |
| `G00(Ndim,Ndim,N_FL)` | `cmplx` | Time-displaced Green function $\langle\langle \mathcal{T} \hat{c}_{i,s}(0) \hat{c}_{j,s}^\dagger(0) \rangle\rangle$ |
| `GTT(Ndim,Ndim,N_FL)` | `cmplx` | Time-displaced Green function $\langle\langle \mathcal{T} \hat{c}_{i,s}(\tau) \hat{c}_{j,s}^\dagger(\tau) \rangle\rangle$ |
| `N_SUN` | `int` | Number of fermion colors $N_{\mathrm{col}}$ |
| `ZS` | `cmplx` | $\mathtt{ZS} = \mathrm{sgn}(C)$, see Sec. 5.4 |
| `ZP` | `cmplx` | $\mathtt{ZP} = e^{-S(C)} / \mathrm{Re}\left[ e^{-S(C)} \right]$, see Sec. 5.4 |
| `Obs` | `Obser_Latt` | **Output**: one or more measurement result |

Table 21: Arguments taken by the subroutines in the module `Predefined_Obs`. Note that a given method makes use of only a subset of this list, as described in this section. Note also that we use the superindex $i = (\boldsymbol{i}, n_{\boldsymbol{i}})$ where $\boldsymbol{i}$ denotes the unit cell and $n_{\boldsymbol{i}}$ the orbital.

For this SU(N) symmetric code, the Green function is diagonal in the spin index and spin independent:

$$\langle\langle \hat{c}_{\boldsymbol{i},n_{\boldsymbol{i}},\alpha}^\dagger \hat{c}_{\boldsymbol{j},n_{\boldsymbol{j}},\beta} \rangle\rangle_C = \delta_{\alpha,\beta} \langle\langle \hat{c}_{\boldsymbol{i},n_{\boldsymbol{i}}}^\dagger \hat{c}_{\boldsymbol{j},n_{\boldsymbol{j}}} \rangle\rangle_C. \tag{166}$$

Hence,

$$
\begin{aligned}
\mathtt{Obs} &= \frac{2N}{N^2-1} \sum_{a=1}^{N^2-1} \left( [\mathrm{Tr} T^a]^2 \langle\langle \hat{c}_{\boldsymbol{i},n_{\boldsymbol{i}}}^\dagger \hat{c}_{\boldsymbol{i},n_{\boldsymbol{i}}} \rangle\rangle_C \langle\langle \hat{c}_{\boldsymbol{j},n_{\boldsymbol{j}}}^\dagger \hat{c}_{\boldsymbol{j},n_{\boldsymbol{j}}} \rangle\rangle_C \right. \\
&\qquad \left. + \mathrm{Tr}\left[ T^a T^a \right] \langle\langle \hat{c}_{\boldsymbol{i},n_{\boldsymbol{i}}}^\dagger \hat{c}_{\boldsymbol{j},n_{\boldsymbol{j}}} \rangle\rangle_C \langle\langle \hat{c}_{\boldsymbol{i},n_{\boldsymbol{i}}} \hat{c}_{\boldsymbol{j},n_{\boldsymbol{j}}}^\dagger \rangle\rangle_C \right) \\
&= N \langle\langle \hat{c}_{\boldsymbol{i},n_{\boldsymbol{i}}}^\dagger \hat{c}_{\boldsymbol{j},n_{\boldsymbol{j}}} \rangle\rangle_C \langle\langle \hat{c}_{\boldsymbol{i},n_{\boldsymbol{i}}} \hat{c}_{\boldsymbol{j},n_{\boldsymbol{j}}}^\dagger \rangle\rangle_C.
\end{aligned}
\tag{167}
$$

Note that we can also define the generators of SU(N) as

$$\hat{S}^\mu_\nu(x) = \hat{c}_{x,\mu}^\dagger \hat{c}_{x,\nu} - \delta_{\mu,\nu} \frac{1}{N} \sum_{\alpha=1}^{N} \hat{c}_{x,\alpha}^\dagger \hat{c}_{x,\alpha}. \tag{168}$$

With this definition, the spin-spin correlations read:

$$\sum_{\mu,\nu=1}^{N} \langle\langle \hat{S}^\mu_\nu(x) \hat{S}^\nu_\mu(y) \rangle\rangle_C = (N^2-1) \langle\langle \hat{c}_{\boldsymbol{x}}^\dagger \hat{c}_{\boldsymbol{y}} \rangle\rangle_C \langle\langle \hat{c}_{\boldsymbol{x}} \hat{c}_{\boldsymbol{y}}^\dagger \rangle\rangle_C. \tag{169}$$

In the above $x$ denotes a super index defining site and orbital. Aside from the normalization, this formulation gives the same result.

### 8.4.2 Equal-time spin correlations

A measurement of the equal-time spin correlations can be obtained by:

```
Call Predefined_Obs_eq_SpinMz_measure(Latt, Latt_unit, List, GR, GRC, N_SUN, ZS, ZP,
                                       ObsZ, ObsXY, ObsXYZ)
```

If `N_FL=2` and `N_SUN=1`, then the routine returns:

$$
\begin{aligned}
\mathtt{ObsZ}\,(\boldsymbol{i} - \boldsymbol{j}, n_{\boldsymbol{i}}, n_{\boldsymbol{j}}) &= 4\langle\langle \hat{\boldsymbol{c}}^{\dagger}_{\boldsymbol{i},n_{\boldsymbol{i}}} S^z \hat{\boldsymbol{c}}_{\boldsymbol{i},n_{\boldsymbol{i}}}\, \hat{\boldsymbol{c}}^{\dagger}_{\boldsymbol{j},n_{\boldsymbol{j}}} S^z \hat{\boldsymbol{c}}_{\boldsymbol{j},n_{\boldsymbol{j}}}\rangle\rangle_C \\
&\quad - 4\langle\langle \hat{\boldsymbol{c}}^{\dagger}_{\boldsymbol{i},n_{\boldsymbol{i}}} S^z \hat{\boldsymbol{c}}_{\boldsymbol{i},n_{\boldsymbol{i}}}\rangle\rangle_C \langle\langle \hat{\boldsymbol{c}}^{\dagger}_{\boldsymbol{j},n_{\boldsymbol{j}}} S^z \hat{\boldsymbol{c}}_{\boldsymbol{j},n_{\boldsymbol{j}}}\rangle\rangle_C, \\
\mathtt{ObsXY}\,(\boldsymbol{i} - \boldsymbol{j}, n_{\boldsymbol{i}}, n_{\boldsymbol{j}}) &= 2\left(\langle\langle \hat{\boldsymbol{c}}^{\dagger}_{\boldsymbol{i},n_{\boldsymbol{i}}} S^x \hat{\boldsymbol{c}}_{\boldsymbol{i},n_{\boldsymbol{i}}}\, \hat{\boldsymbol{c}}^{\dagger}_{\boldsymbol{j},n_{\boldsymbol{j}}} S^x \hat{\boldsymbol{c}}_{\boldsymbol{j},n_{\boldsymbol{j}}}\rangle\rangle_C + \langle\langle \hat{\boldsymbol{c}}^{\dagger}_{\boldsymbol{i},n_{\boldsymbol{i}}} S^y \hat{\boldsymbol{c}}_{\boldsymbol{i},n_{\boldsymbol{i}}}\, \boldsymbol{c}^{\dagger}_{\boldsymbol{j},n_{\boldsymbol{j}}} S^y \hat{\boldsymbol{c}}_{\boldsymbol{j},n_{\boldsymbol{j}}}\rangle\rangle_C\right), \\
\mathtt{ObsXYZ} &= \frac{2 \cdot \mathtt{ObsXY} + \mathtt{ObsZ}}{3}.
\end{aligned}
\tag{170}
$$

Here $\hat{\boldsymbol{c}}^{\dagger}_{\boldsymbol{i},n_{\boldsymbol{i}}} = \left(\hat{c}^{\dagger}_{\boldsymbol{i},n_{\boldsymbol{i}},\uparrow}, \hat{c}^{\dagger}_{\boldsymbol{i},n_{\boldsymbol{i}},\downarrow}\right)$ is a two component spinor and $\boldsymbol{S} = \frac{1}{2}\boldsymbol{\sigma}$, with

$$
\boldsymbol{\sigma} = \left(\begin{bmatrix} 0 & 1 \\ 1 & 0 \end{bmatrix}, \begin{bmatrix} 0 & -i \\ i & 0 \end{bmatrix}, \begin{bmatrix} 1 & 0 \\ 0 & -1 \end{bmatrix}\right),
\tag{171}
$$

the Pauli spin matrices.

### 8.4.3 Equal-time Green function

A measurement of the equal-time Green function can be obtained by:

```
Call Predefined_Obs_eq_Green_measure(Latt, Latt_unit, List, GR, GRC, N_SUN, ZS, ZP, Obs)
```

Which returns:

$$
\mathtt{Obs}(\boldsymbol{i} - \boldsymbol{j}, n_{\boldsymbol{i}}, n_{\boldsymbol{j}}) = \sum_{\sigma=1}^{N_{\mathrm{col}}} \sum_{s=1}^{N_{\mathrm{fl}}} \langle \hat{c}^{\dagger}_{\boldsymbol{i},n_{\boldsymbol{i}},\sigma,s} \hat{c}_{\boldsymbol{j},n_{\boldsymbol{j}},\sigma,s}\rangle.
\tag{172}
$$

### 8.4.4 Equal-time density-density correlations

A measurement of equal-time density-density correlations can be obtained by:

```
Call Predefined_Obs_eq_Den_measure(Latt, Latt_unit, List, GR, GRC, N_SUN, ZS, ZP, Obs)
```

Which returns:

$$
\mathtt{Obs}(\boldsymbol{i} - \boldsymbol{j}, n_{\boldsymbol{i}}, n_{\boldsymbol{j}}) = \langle\langle \hat{N}_{\boldsymbol{i},n_{\boldsymbol{i}}} \hat{N}_{\boldsymbol{j},n_{\boldsymbol{j}}}\rangle - \langle \hat{N}_{\boldsymbol{i},n_{\boldsymbol{i}}}\rangle \langle \hat{N}_{\boldsymbol{j},n_{\boldsymbol{j}}}\rangle\rangle_C,
\tag{173}
$$

where

$$
\hat{N}_{\boldsymbol{i},n_{\boldsymbol{i}}} = \sum_{\sigma=1}^{N_{\mathrm{col}}} \sum_{s=1}^{N_{\mathrm{fl}}} \hat{c}^{\dagger}_{\boldsymbol{i},n_{\boldsymbol{i}},\sigma,s} \hat{c}_{\boldsymbol{i},n_{\boldsymbol{i}},\sigma,s}.
\tag{174}
$$

### 8.4.5 Time-displaced Green function

A measurement of the time-displaced Green function can be obtained by:

```
Call Predefined_Obs_tau_Green_measure(Latt, Latt_unit, List, NT, GT0, GOT, GOO, GTT,
                                      N_SUN, ZS, ZP, Obs)
```

Which returns:

$$\texttt{Obs}(\boldsymbol{i}-\boldsymbol{j},\tau,n_i,n_j) = \sum_{\sigma=1}^{N_{\text{col}}}\sum_{s=1}^{N_{\text{fl}}}\langle\langle\hat{c}^\dagger_{\boldsymbol{i},n_i,\sigma,s}(\tau)\hat{c}_{\boldsymbol{j},n_j,\sigma,s}\rangle\rangle_C \tag{175}$$

### 8.4.6 Time-displaced SU(N) spin-spin correlations

A measurement of time-displaced spin-spin correlations for SU(N) models ($N_{\text{fl}} = 1$) can be obtained by:

```
Call Predefined_Obs_tau_SpinSUN_measure(Latt, Latt_unit, List, NT, GT0, GOT, GOO, GTT,
                                        N_SUN, ZS, ZP, Obs)
```

$$\texttt{Obs}(\boldsymbol{i}-\boldsymbol{j},\tau,n_i,n_j) = \frac{2N}{N^2-1}\sum_{a=1}^{N^2-1}\langle\hat{\boldsymbol{c}}^\dagger_{\boldsymbol{i},n_i}(\tau)T^a\hat{\boldsymbol{c}}_{\boldsymbol{i},n_i}(\tau)\,\hat{\boldsymbol{c}}^\dagger_{\boldsymbol{j},n_j}T^a\hat{\boldsymbol{c}}_{\boldsymbol{j},n_j}\rangle\rangle_C \tag{176}$$

where $T^a$ are the generators of SU(N) (see Sec. 8.4.1 for more details).

### 8.4.7 Time-displaced spin correlations

A measurement of time-displaced spin-spin correlations for $Mz$ models ($N_{\text{fl}} = 2, N_{\text{col}} = 1$) is returned by:

```
Call Predefined_Obs_tau_SpinMz_measure(Latt, Latt_unit, List, NT, GT0, GOT, GOO, GTT,
                                       N_SUN, ZS, ZP, ObsZ, ObsXY, ObsXYZ)
```

Which calculates the following observables:

$$\begin{aligned}
\texttt{ObsZ}(\boldsymbol{i}-\boldsymbol{j},\tau,n_i,n_j) &= 4\langle\langle\hat{\boldsymbol{c}}^\dagger_{\boldsymbol{i},n_i}(\tau)S^z\hat{\boldsymbol{c}}_{\boldsymbol{i},n_i}(\tau)\,\hat{\boldsymbol{c}}^\dagger_{\boldsymbol{j},n_j}S^z\hat{\boldsymbol{c}}_{\boldsymbol{j},n_j}\rangle\rangle_C \\
&\quad - 4\langle\langle\hat{\boldsymbol{c}}^\dagger_{\boldsymbol{i},n_i}S^z\hat{\boldsymbol{c}}_{\boldsymbol{i},n_i}\rangle\rangle_C\langle\langle\,\hat{\boldsymbol{c}}^\dagger_{\boldsymbol{j},n_j}S^z\hat{\boldsymbol{c}}_{\boldsymbol{j},n_j}\rangle\rangle_C \\
\texttt{ObsXY}(\boldsymbol{i}-\boldsymbol{j},\tau,n_i,n_j) &= 2\left(\langle\langle\hat{\boldsymbol{c}}^\dagger_{\boldsymbol{i},n_i}(\tau)S^x\hat{\boldsymbol{c}}_{\boldsymbol{i},n_i}(\tau)\,\hat{\boldsymbol{c}}^\dagger_{\boldsymbol{j},n_j}S^x\hat{\boldsymbol{c}}_{\boldsymbol{j},n_j}\rangle\rangle_C\right. \\
&\quad \left. + \langle\langle\hat{\boldsymbol{c}}^\dagger_{\boldsymbol{i},n_i}(\tau)S^y\hat{\boldsymbol{c}}_{\boldsymbol{i},n_i}(\tau)\,\boldsymbol{c}^\dagger_{\boldsymbol{j},n_j}S^y\hat{\boldsymbol{c}}_{\boldsymbol{j},n_j}\rangle\rangle_C\right) \\
\texttt{ObsXYZ} &= \frac{2\cdot\texttt{ObsXY} + \texttt{ObsZ}}{3}.
\end{aligned} \tag{177}$$

### 8.4.8 Time-displaced density-density correlations

A measurement of time-displaced density-density correlations for general SU(N) models is given by:

```
Call Predefined_Obs_tau_Den_measure(Latt, Latt_unit, List,  NT, GT0, GOT, GOO, GTT,
                                    N_SUN, ZS, ZP,  Obs)
```

Which returns:

$$\texttt{Obs}(\boldsymbol{i}-\boldsymbol{j},\tau,n_{\boldsymbol{i}},n_{\boldsymbol{j}}) = \langle\langle\hat{N}_{\boldsymbol{i},n_{\boldsymbol{i}}}(\tau)\hat{N}_{\boldsymbol{j},n_{\boldsymbol{j}}}\rangle - \langle\hat{N}_{\boldsymbol{i},n_{\boldsymbol{i}}}\rangle\langle\hat{N}_{\boldsymbol{j},n_{\boldsymbol{j}}}\rangle\rangle_C. \tag{178}$$

The density operator is defined in Eq. (174).

### 8.4.9 Dimer-Dimer correlations

Let

$$\hat{S}^\mu_\nu(x) = \hat{c}^\dagger_{x,\mu}\hat{c}_{x,\nu} - \delta_{\mu,\nu}\frac{1}{N}\sum_{\alpha=1}^N \hat{c}^\dagger_{x,\alpha}\hat{c}_{x,\alpha} \tag{179}$$

be the generators of SU(N). Dimer-Dimer correlations are defined as:

$$\langle\langle\hat{S}^\mu_\nu(x,\tau)\hat{S}^\nu_\mu(y,\tau)\hat{S}^\gamma_\delta(w)\hat{S}^\delta_\gamma(z)\rangle\rangle_C, \tag{180}$$

where the sum over repeated indices from $1\cdots N$ is implied. The calculation is carried out for the self-adjoint antisymmetric representation of SU(N) for which $\sum_{\alpha=1}^N \hat{c}^\dagger_{x,\alpha}\hat{c}_{x,\alpha} = N/2$, such that the generators can be replaced by:

$$\hat{S}^\mu_\nu(x) = \hat{c}^\dagger_{x,\mu}\hat{c}_{x,\nu} - \delta_{\mu,\nu}\frac{1}{2}. \tag{181}$$

The function

```
Complex (Kind=Kind(0.d0)) function Predefined_Obs_dimer_tau(x, y, w, z, GT0, GOT, G00,
                                                           GTT, N_SUN, N_FL)
```

returns the value of the time-displaced dimer-dimer correlation function. The function

```
Complex (Kind=Kind(0.d0)) function Predefined_Obs_dimer_eq(x, y, w, z, GR, GRC, N_SUN,
                                                          N_FL)
```

returns the value of the equal time dimer-dimer correlation function:

$$\langle\langle\hat{S}^\mu_\nu(x,\tau)\hat{S}^\nu_\mu(y,\tau)\hat{S}^\gamma_\delta(w,\tau)\hat{S}^\delta_\gamma(z,\tau)\rangle\rangle_C. \tag{182}$$

Here, both `GR` and `GRC` are on time slice $\tau$.

To compute the background terms, the function

```
Complex (Kind=Kind(0.d0)) function Predefined_Obs_dimer0_eq(x, y, GR, N_SUN, N_FL)
```

returns

$$\langle\langle\hat{S}^\mu_\nu(x,\tau)\hat{S}^\nu_\mu(y,\tau)\rangle\rangle_C. \tag{183}$$

All routines are programmed for `N_SUN = 2,4,6,8` at `N_FL=1`. The routines also handle the case of broken SU(2) spin symmetry corresponding to `N_FL=2` and `N_SUN=1`. To carry out the Wick decomposition and sums over spin indices, we use the Mathematica notebooks `DimerDimer_SU2_NFL_2.nb` and `DimerDimer_SUN_NFL_1.nb`.

### 8.4.10 Cotunneling for Kondo models

The Kondo lattice model (KLM), $\hat{H}_{KLM}$ is obtained by carrying out a canonical Schrieffer-Wolf [142] transformation of the periodic Anderson model (PAM), $\hat{H}_{PAM}$. Hence, $e^{\hat{S}}\,\hat{H}_{PAM}e^{-S} = \hat{H}_{KLM}$ with $\hat{S}^\dagger = -\hat{S}$. Let $\hat{f}_{x,\sigma}$ create an electron on the correlation f-orbital of the PAM. Then,

$$e^{\hat{S}}\hat{f}^\dagger_{x,\sigma'}e^{-\hat{S}} \simeq \frac{2V}{U}\left(\hat{c}^\dagger_{x,-\sigma'}\hat{S}^{\sigma'}_x + \sigma'\hat{c}^\dagger_{x,\sigma'}\hat{S}^z_x\right) \equiv \frac{2V}{U}\tilde{\hat{f}}^\dagger_{x,\sigma'}. \tag{184}$$

In the above, it is understood that $\sigma'$ takes the value 1 $(-1)$ for up (down) spin degrees of freedom, that $\hat{S}^{\sigma'}_x = \hat{f}^\dagger_{x,\sigma'}\hat{f}_{x,-\sigma'}$ and that $\hat{S}^z_x = \frac{1}{2}\sum_{\sigma'}\sigma'\hat{f}^\dagger_{x,\sigma'}\hat{f}_{x,\sigma'}$. Finally, $\hat{c}^\dagger_{x,\sigma'}$ corresponds to the conduction electron that hybridizes with $\hat{f}^\dagger_{x,\sigma'}$. This form matches that derived in Ref. [143] and a calculation of the former equation can be found in Ref. [144]. An identical, but more transparent formulation is given in Ref. [145] and reads:

$$\tilde{\hat{f}}^\dagger_{x,\sigma} = \sum_{\sigma'}\hat{c}^\dagger_{x,\sigma'}\boldsymbol{\sigma}_{\sigma',\sigma}\cdot\hat{\boldsymbol{S}}_x, \tag{185}$$

where $\boldsymbol{\sigma}$ denotes the vector of Pauli spin matrices. With the above, one will readily show that the $\tilde{\hat{f}}^\dagger_{x,\sigma}$ transforms as $\hat{f}^\dagger_{x,\sigma}$ under an SU(2) spin rotation. The function

```
Complex (Kind=Kind(0.d0)) function Predefined_Obs_Cotunneling(x_c, x, y_c, y, GT0, GOT,
                                             GOO, GTT, N_SUN, N_FL)
```

returns the value of the time displaced correlation function:

$$\sum_\sigma\langle\langle\tilde{\hat{f}}^\dagger_{x,\sigma}(\tau)\tilde{\hat{f}}_{y,\sigma}(0)\rangle\rangle_C. \tag{186}$$

Here, $x_c$ and $y_c$ correspond to the conduction orbitals that hybridize with the $x$ and $y$ f-orbitals. The routine works for SU(N) symmetric codes corresponding to N_FL=1 and N_SUN = 2,4,6,8. For the larger N-values, we have replaced the generators of SU(2) with that of SU(N). The routine also handles the case where spin-symmetry is broken by e.g. a Zeeman field. This corresponds to the case N_FL=2 and N_SUN=1. Note that the function only carries out the Wick decomposition and the handling of the observable type corresponding to this quantity has to be done by the user. To carry out the Wick decomposition and sums over spin indices, we use the Mathematica notebooks `Cotunneling_SU2_NFL_2.nb` and `Cotunneling_-SUN_NFL_1.nb`.

### 8.4.11 Rényi Entropy

The module `entanglement_mod.F90` allows one to compute the $2^{\mathrm{nd}}$ Rényi entropy, $S_2$, for a subsystem. Using Eq. (24), $S_2$ can be expressed as a stochastic average of an observable constructed from two independent simulations of the model [60]:

$$e^{-S_2} = \sum_{C_1,C_2}P(C_2)P(C_1)\det\left[G_A(\tau_0;C_1)G_A(\tau_0;C_2) - (\mathbb{1} - G_A(\tau_0;C_1))(\mathbb{1} - G_A(\tau_0;C_2))\right],$$
$$\tag{187}$$

where $G_A(\tau_0;C_i)$, $i = 1,2$ is the Green function matrix restricted to the desired subsystem $A$ at a given time-slice $\tau_0$, and for the configuration $C_i$ of the replica $i$. The degrees of freedom defining the subsystem $A$ are lattice site, flavor index, and color index.

Notice that, due to its formulation, sampling $S_2$ requires an MPI simulation with at least 2 processes. Also, only real-space partitions are currently supported.

A measurement of the $2^{\text{nd}}$ Rényi entropy can be obtained by:

```
Call Predefined_Obs_scal_Renyi_Ent(GRC, List, Nsites, N_SUN, ZS, ZP, Obs)
```

which returns the observable `Obs`, for which $\langle \text{Obs} \rangle = e^{-S_2}$. The subsystem $A$ can be defined in a number of different ways, which are handled by what we call *specializations* of the subroutine, described as follows.

In the most general case, `List(:, N_FL, N_SUN)` is a three-dimensional array that contains the list of lattice sites in $A$ for every flavor and color index; `Nsites(N_FL, N_SUN)` is then a bidimensional array that provides the number of lattice sites in the subsystem for every flavor and color index; and the argument `N_SUN` must be omitted in the call.

For a subsystem whose degrees of freedom, for a given flavor index, have a common value of color indexes, `Predefined_Obs_scal_Renyi_Ent` can be called by providing `List(:, N_FL)` as a bidimensional array that contains the list of lattice sites for every flavor index. In this case, `Nsites(N_FL)` provides the number of sites in the subsystem for any given flavor index, while `N_SUN(N_FL)` contains the number of color indexes for a given flavor index.

Finally, a specialization exists for the simple case of a subsystem whose lattice degrees of freedom are flavor- and color-independent. In this case, `List(:)` is a one-dimensional array containing the lattice sites of the subsystem. `Nsites` is the number of sites, and `N_SUN` is the number of color indexes belonging to the subsystem. Accordingly, for every element `I` of `List`, the subsystem contains all degrees of freedom with site index `I`, any flavor index, and $1 \ldots$ `N_SUN` color index.

### Mutual Information

The mutual information between two subsystems $A$ and $B$ is given by

$$I_2 = -\ln\langle \text{Renyi\_A} \rangle - \ln\langle \text{Renyi\_B} \rangle + \ln\langle \text{Renyi\_AB} \rangle, \tag{188}$$

where `Renyi_A`, `Renyi_B`, and `Renyi_AB` are the second Rényi entropies of $A$, $B$, and $A \cup B$, respectively.

The measurements necessary for computing $I_2$ are obtained by:

```
Call Predefined_Obs_scal_Mutual_Inf(GRC, List_A, Nsites_A, List_B, Nsites_B, N_SUN,
                                    ZS, ZP, Obs)
```

which returns the $2^{\text{nd}}$ Rényi entropies mentioned above, stored in the variable `Obs`. Here, `List_A` and `Nsites_A` are input parameters describing the subsystem $A$ – with the same conventions and specializations described above – and `List_B` and `Nsites_B` are the corresponding input parameters for the subsystem $B$, while `N_SUN` is assumed to be identical for $A$ and $B$.

## 8.5 Predefined trial wave functions

When using the projective algorithm (see Sec. 3), trial wave functions must be specified. These are stored in variables of the `WaveFunction` type (Sec. 5.5). The ALF package provides a set of predefined trial wave functions $|\Psi_{T,L/R}\rangle =$ `WF_L/R`, returned by the call:

```
Call Predefined_TrialWaveFunction(Lattice_type, Ndim, List, Invlist, Latt, Latt_unit,
                                  N_part, N_FL, WF_L, WF_R)
```

Twisted boundary conditions (`Phi_X_vec=0.01`) are implemented for some lattices in order to generate non-degenerate trial wave functions. Here the marker "`_vec`" indicates the variable may assume different values depending on the flavor (e.g., spin up and down). Currently predefined trial wave functions are flavor independent.

The predefined trial wave functions correspond to the solution of the non-interacting tight binding Hamiltonian on each of the predefined lattices. These solutions are the ground states of the predefined hopping matrices (Sec. 8.2) with default parameters, for each lattice, as follows.

### 8.5.1 Square

Parameter values for the predefined trial wave function on the square lattice:

```
Checkerboard  = .false.
Symm          = .false.
Bulk          = .false.
N_Phi_vec     = 0
Phi_X_vec     = 0.01d0
Phi_Y_vec     = 0.d0
Ham_T_vec     = 1.d0
Ham_Chem_vec  = 0.d0
Dtau          = 1.d0
```

### 8.5.2 Honeycomb

The twisted boundary condition for the square lattice lifts the degeneracy present at half-band filling, but breaks time reversal symmetry as well as the $C_4$ lattice symmetry. If time reversal symmetry is required to avoid the negative sign problem (that would be the case for the attractive Hubbard model at finite doping), then this choice of the trial wave function will introduce a negative sign. One should then use the trial wave function presented in Sec. 7.5. For the Honeycomb case, the trial wave function we choose is the ground state of the tight binding model with small next-next-next nearest hopping matrix element $t'$ [130]. This breaks the $C_3$ symmetry and shifts the Dirac cone away from the zone boundary. Time reversal symmetry is however not broken. Alternatively, one could include a small Kekule mass term. As shown in Sec. 3.3 both choices of trial wave functions produce good results.

### 8.5.3 N-leg ladder

Parameter values for the predefined trial wave function on the N-leg ladder lattice:

```
Checkerboard   = .false.
Symm           = .false.
Bulk           = .false.
N_Phi_vec      = 0
Phi_X_vec      = 0.01d0
Phi_Y_vec      = 0.d0
Ham_T_vec      = 1.d0
Ham_Tperp_vec  = 1.d0
Ham_Chem_vec   = 0.d0
Dtau           = 1.d0
```

### 8.5.4 Bilayer square

Parameter values for the predefined trial wave function on the bilayer square lattice:

```
Checkerboard  = .false.
Symm          = .false.
Bulk          = .false.
N_Phi_vec     = 0
Phi_X_vec     = 0.d0
Phi_Y_vec     = 0.d0
Ham_T_vec     = 1.d0
Ham_T2_vec    = 0.d0
Ham_Tperp_vec = 1.d0
Ham_Chem_vec  = 0.d0
Dtau          = 1.d0
```

### 8.5.5 Bilayer honeycomb

Parameter values for the predefined trial wave function on the bilayer honeycomb lattice:

```
Checkerboard  = .false.
Symm          = .false.
Bulk          = .false.
N_Phi_vec     = 0
Phi_X_vec     = 0.d0
Phi_Y_vec     = 0.d0
Ham_T_vec     = 1.d0
Ham_T2_vec    = 0.d0
Ham_Tperp_vec = 1.d0
Ham_Chem_vec  = 0.d0
Dtau          = 1.d0
```

# 9 Model Classes

The ALF library comes with five model classes: (i) SU(N) Hubbard models, (ii) O(2N) t-V models, (iii) Kondo models, (iv) long-range Coulomb models, and (v) generic $\mathbf{Z}_2$ lattice gauge theories coupled to $\mathbf{Z}_2$ matter and fermions. Below we detail the functioning of these classes.

## 9.1 SU(N) Hubbard models `Hamiltonian_Hubbard_smod.F90`

The parameter space for this model class reads:

```
&VAR_Hubbard              !! Variables for the Hubbard class
Mz        = .T.           ! Whether to use the M_z-Hubbard model: Nf=2; N_SUN must be
                          ! even. HS field couples to the z-component of magnetization
ham_T     = 1.d0          ! Hopping parameter
ham_chem  = 0.d0          ! Chemical potential
ham_U     = 4.d0          ! Hubbard interaction
ham_T2    = 1.d0          ! For bilayer systems
ham_U2    = 4.d0          ! For bilayer systems
ham_Tperp = 1.d0          ! For bilayer systems
Continuous  = .F.         ! For continuous HS decomposition
/
```

In the above listing, `ham_T` and `ham_T2` correspond to the hopping in the first and second layers respectively and `ham_Tperp` is to the interlayer hopping. The Hubbard $U$ term has an orbital index, `ham_U` for the first and `ham_U2` for the second layers. Finally, `ham_chem` corresponds to the chemical potential. If the flag `Mz` is set to `.False.`, then the code simulates the following SU(N) symmetric Hubbard model:

$$\hat{H} = \sum_{(\boldsymbol{i},\boldsymbol{\delta}),(\boldsymbol{j},\boldsymbol{\delta}')} \sum_{\sigma=1}^{N} T_{(\boldsymbol{i},\boldsymbol{\delta}),(\boldsymbol{j},\boldsymbol{\delta}')} \hat{c}^{\dagger}_{(\boldsymbol{i},\boldsymbol{\delta}),\sigma} e^{\frac{2\pi i}{\Phi_0} \int_{\boldsymbol{i}+\boldsymbol{\delta}}^{\boldsymbol{j}+\boldsymbol{\delta}'} \boldsymbol{A}(\boldsymbol{l})d\boldsymbol{l}} \hat{c}_{(\boldsymbol{j},\boldsymbol{\delta}'),\sigma}$$
$$+ \sum_{\boldsymbol{i}} \sum_{\delta} \frac{U_{\delta}}{N} \left( \sum_{\sigma=1}^{N} \left[ \hat{c}^{\dagger}_{(\boldsymbol{i},\boldsymbol{\delta}),\sigma} \hat{c}_{(\boldsymbol{i},\boldsymbol{\delta}),\sigma} - 1/2 \right] \right)^2 - \mu \sum_{(\boldsymbol{i},\boldsymbol{\delta})} \sum_{\sigma=1}^{N} \hat{c}^{\dagger}_{(\boldsymbol{i},\boldsymbol{\delta}),\sigma} \hat{c}_{(\boldsymbol{i},\boldsymbol{\delta}),\sigma}. \quad (189)$$

The generic hopping is taken from Eq. (141) with appropriate boundary conditions given by Eq. (142). The index $\boldsymbol{i}$ runs over the unit cells, $\boldsymbol{\delta}$ over the orbitals in each unit cell and $\sigma$ from 1 to $N$ and encodes the SU(N) symmetry. Note that $N$ corresponds to `N_SUN` in the code. The flavor index is set to unity such that it does not appear in the Hamiltonian. The chemical potential $\mu$ is relevant only for the finite temperature code.

If the variable `Mz` is set to `.True.`, then the code requires `N_SUN` to be even and simulates the following Hamiltonian:

$$\hat{H} = \sum_{(\boldsymbol{i},\boldsymbol{\delta}),(\boldsymbol{j},\boldsymbol{\delta}')} \sum_{\sigma=1}^{N/2} \sum_{s=1,2} T_{(\boldsymbol{i},\boldsymbol{\delta}),(\boldsymbol{j},\boldsymbol{\delta}')} \hat{c}^{\dagger}_{(\boldsymbol{i},\boldsymbol{\delta}),\sigma,s} e^{\frac{2\pi i}{\Phi_0} \int_{\boldsymbol{i}+\boldsymbol{\delta}}^{\boldsymbol{j}+\boldsymbol{\delta}'} \boldsymbol{A}(\boldsymbol{l})d\boldsymbol{l}} \hat{c}_{(\boldsymbol{j},\boldsymbol{\delta}'),\sigma,s}$$
$$- \sum_{\boldsymbol{i}} \sum_{\delta} \frac{U_{\delta}}{N} \left( \sum_{\sigma=1}^{N/2} \left[ \hat{c}^{\dagger}_{(\boldsymbol{i},\boldsymbol{\delta}),\sigma,2} \hat{c}_{(\boldsymbol{i},\boldsymbol{\delta}),\sigma,2} - \hat{c}^{\dagger}_{(\boldsymbol{i},\boldsymbol{\delta}),\sigma,1} \hat{c}_{(\boldsymbol{i},\boldsymbol{\delta}),\sigma,1} \right] \right)^2$$
$$- \mu \sum_{(\boldsymbol{i},\boldsymbol{\delta})} \sum_{\sigma=1}^{N/2} \sum_{s=1,2} \hat{c}^{\dagger}_{(\boldsymbol{i},\boldsymbol{\delta}),\sigma,s} \hat{c}_{(\boldsymbol{i},\boldsymbol{\delta}),\sigma,s}. \quad (190)$$

In this case, the flavor index `N_FL` takes the value 2. Cleary at $N = 2$, both modes correspond to the Hubbard model. For $N$ even and $N > 2$ the models differ. In particular in the latter Hamiltonian the U(N) symmetry is broken down to U(N/2) $\otimes$ U(N/2).

It the variable `Continuous=.T.` then the code will use the generic HS transformation:

$$e^{\alpha \hat{A}^2} = \frac{1}{\sqrt{2\pi}} \int d\phi e^{-\phi^2/2 + \sqrt{2\alpha}\hat{A}} \quad (191)$$

as opposed to the discrete version of Eq. 11. If the Langevin flag is set to false, the code will use the single spin-flip update:

$$\phi \rightarrow \phi + \texttt{Amplitude} \, (\xi - 1/2) \quad (192)$$

where $\xi$ is a random number $\in [0,1]$ and `Amplitude` is defined in the `Fields_mod.F90` module. Since this model class works for all predefined lattices (see Fig. 5) it includes the SU(N) periodic Anderson model on the square and Honeycomb lattices. Finally, we note that the executable for this class is given by `Hubbard.out`.

As an example, we can consider the periodic Anderson model. Here we choose the `Bilayer_square` lattice `Ham_U = Ham_T2 = 0`, `Ham_U2=` $U_f$, `Ham_tperp=` $V$ and `Ham_T=` 1. The pyALF based python script `Hubbard_PAM.py` produces the data shown in Fig. 7 for the L=8 lattice.

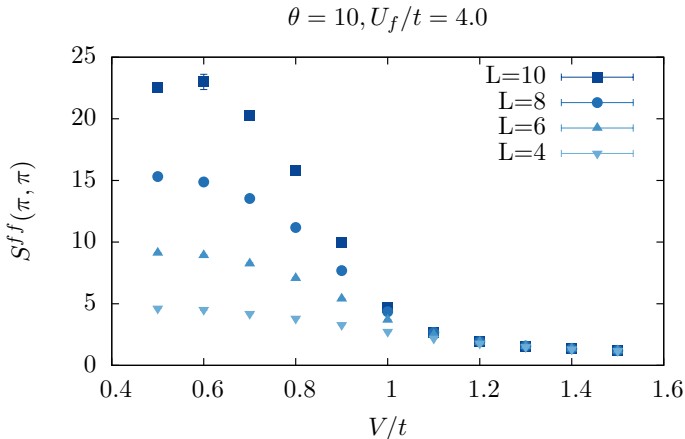

Figure 7: The periodic Anderson model. Here we plot the equal-time spin structure factor of the f-electrons at $\boldsymbol{q} = (\pi, \pi)$. This quantity is found in the file `SpinZ_eqJK`. The pyALF based python script `Hubbard_PAM.py` produces the data shown for the $L = 8$ lattice. One sees that for the chosen value of $U_f/t$ the competition between the RKKY interaction and Kondo screening drives the system through a magnetic order-disorder transition at $V_c/t \simeq 1$ [146].

## 9.2 SU(N) t-V models `Hamiltonian_tV_smod.F90`

The parameter space for this model class reads:

```
&VAR_tV                    !! Variables for the t-V class
ham_T     = 1.d0           ! Hopping parameter
ham_chem  = 0.d0           ! Chemical potential
ham_V     = 0.5d0          ! interaction strength
ham_T2    = 1.d0           ! For bilayer systems
ham_V2    = 0.5d0          ! For bilayer systems
ham_Tperp = 1.d0           ! For bilayer systems
ham_Vperp = 0.5d0          ! For bilayer systems
/
```

In the above `ham_T` and `ham_T2` and `ham_Tperp` correspond to the hopping in the first and second layers respectively and `ham_Tperp` is to the interlayer hopping. The interaction term has an orbital index, `ham_V` for the first and `ham_V2` for the second layers, and `ham_Vperp` for interlayer coupling. Note that we use the same sign conventions here for both the hopping parameters and the interaction strength. This implies a relative minus sign between here and the $U_\delta$ interaction strength of the Hubbard model (see Sec. 9.1). Finally `ham_chem` corresponds to the chemical potential. Let us introduce the operator

$$\hat{b}_{\langle(\boldsymbol{i},\boldsymbol{\delta}),(\boldsymbol{j},\boldsymbol{\delta}')\rangle} = \sum_{\sigma=1}^{N} \hat{c}^{\dagger}_{(\boldsymbol{i},\boldsymbol{\delta}),\sigma} e^{\frac{2\pi i}{\Phi_0} \int_{\boldsymbol{i}+\boldsymbol{\delta}}^{\boldsymbol{j}+\boldsymbol{\delta}'} \boldsymbol{A}(\boldsymbol{l})d\boldsymbol{l}} \hat{c}_{(\boldsymbol{j},\boldsymbol{\delta}'),\sigma} + \text{H.c.} \tag{193}$$

The model is then defined as follows:

$$\hat{H} = \sum_{\langle(\boldsymbol{i},\boldsymbol{\delta}),(\boldsymbol{j},\boldsymbol{\delta}')\rangle} T_{(\boldsymbol{i},\boldsymbol{\delta}),(\boldsymbol{j},\boldsymbol{\delta}')} \hat{b}_{\langle(\boldsymbol{i},\boldsymbol{\delta}),(\boldsymbol{j},\boldsymbol{\delta}')\rangle} + \sum_{\langle(\boldsymbol{i},\boldsymbol{\delta}),(\boldsymbol{j},\boldsymbol{\delta}')\rangle} \frac{V_{(\boldsymbol{i},\boldsymbol{\delta}),(\boldsymbol{j},\boldsymbol{\delta}')}}{N} \left( \hat{b}_{\langle(\boldsymbol{i},\boldsymbol{\delta}),(\boldsymbol{j},\boldsymbol{\delta}')\rangle} \right)^2$$
$$- \mu \sum_{(\boldsymbol{i},\boldsymbol{\delta})} \sum_{\sigma=1}^{N} \hat{c}_{(\boldsymbol{i},\boldsymbol{\delta}),\sigma}^{\dagger} \hat{c}_{(\boldsymbol{i},\boldsymbol{\delta}),\sigma} . \tag{194}$$

The generic hopping is taken from Eq. (141) with appropriate boundary conditions given by Eq. (142). The index $\boldsymbol{i}$ runs over the unit cells, $\boldsymbol{\delta}$ over the orbitals in each unit cell and $\sigma$ from 1 to $N$, encoding the SU(N) symmetry. Note that $N$ corresponds to N_SUN in the code. The flavor index is set to unity such that it does not appear in the Hamiltonian. The chemical potential $\mu$ is relevant only for the finite temperature code. An example showing how to run this model class can be found in the pyALF based Jupyter notebook tV_model.ipynb.

As a concrete example, we can consider the Hamiltonian of the t-V model of SU(N) fermions on the square lattice,

$$\hat{H} = -t \sum_{\langle \boldsymbol{i},\boldsymbol{j}\rangle} \hat{b}_{\langle \boldsymbol{i},\boldsymbol{j}\rangle} - \frac{V}{N} \sum_{\langle \boldsymbol{i},\boldsymbol{j}\rangle} \left( \hat{b}_{\langle \boldsymbol{i},\boldsymbol{j}\rangle} \right)^2 - \mu \sum_{\boldsymbol{i}} \sum_{\sigma=1}^{N} \hat{c}_{\boldsymbol{i},\sigma}^{\dagger} \hat{c}_{\boldsymbol{i},\sigma} , \tag{195}$$

which can be simulated by setting ham_T $= t$, ham_V $= V$, and ham_chem $= \mu$. At half-band filling $\mu = 0$, the sign problem is absent for $V > 0$ and for all values of $N$ [75, 147]. For even values of $N$ no sign problem occurs for $V > 0$ and arbitrary chemical potentials [74].

Note that in the absence of orbital magnetic fields, the model has an $O(2N)$ symmetry. This can be seen by writing the model in a Majorana basis (see e.g. Ref. [21]).

### 9.3  SU(N) Kondo lattice models Hamiltonian_Kondo_smod.F90

The Kondo lattice model we consider is an SU(N) generalization of the SU(2) Kondo-model discussed in [31, 32]. Here we follow the work of Ref. [50]. Let $T^a$ be the $N^2 - 1$ generators of SU(N) that satisfy the normalization condition:

$$\mathrm{Tr}\left[T^a T^b\right] = \frac{1}{2}\delta_{a,b}. \tag{196}$$

For the SU(2) case, $T^a$ corresponds to the $T = \frac{1}{2}\boldsymbol{\sigma}$ with $\boldsymbol{\sigma}$ a vector of the three Pauli spin matrices, Eq. (171). The Hamiltonian is defined on bilayer square or honeycomb lattices, with hopping restricted to the first layer (i.e conduction orbitals $\boldsymbol{c}_i^{\dagger}$) and spins, f-orbitals, on the second layer.

$$\hat{H} = -t \sum_{\langle i,j\rangle} \sum_{\sigma=1}^{N} \left( \hat{c}_{i,\sigma}^{\dagger} e^{\frac{2\pi i}{\Phi_0} \int_i^j \boldsymbol{A}\cdot d\boldsymbol{l}} \hat{c}_{j,\sigma} + \mathrm{H.c.} \right) - \mu \sum_{i,\sigma} \hat{c}_{i,\sigma}^{\dagger} \hat{c}_{i,\sigma}$$
$$+ \frac{U_c}{N} \sum_{i} \left( \hat{n}_i^c - \frac{N}{2} \right)^2 + \frac{2J}{N} \sum_{i,a=1}^{N^2-1} \hat{T}_i^{a,c}\hat{T}_i^{a,f}. \tag{197}$$

In the above, $i$ is a super-index accounting for the unit cell and orbital,

$$\hat{T}_i^{a,c} = \sum_{\sigma,\sigma'=1}^{N} \hat{c}_{i,\sigma}^{\dagger} T_{\sigma,\sigma'}^a \hat{c}_{i,\sigma'}, \quad \hat{T}_i^{a,f} = \sum_{\sigma,\sigma'=1}^{N} \hat{f}_{i,\sigma}^{\dagger} T_{\sigma,\sigma'}^a \hat{f}_{i,\sigma'}, \quad \text{and} \quad \hat{n}_i^c = \sum_{\sigma=1}^{N} \hat{c}_{i,\sigma}^{\dagger} \hat{c}_{i,\sigma}. \tag{198}$$

Finally, the constraint

$$\sum_{\sigma=1}^{N} \hat{f}_{i,\sigma}^{\dagger} \hat{f}_{i,\sigma} \equiv \hat{n}_i^f = \frac{N}{2} \tag{199}$$

holds. Some rewriting has to be carried out so as to implement the model. First, we use the relation:

$$\sum_a T_{\alpha,\beta}^a T_{\alpha',\beta'}^a = \frac{1}{2}\left(\delta_{\alpha,\beta'}\delta_{\alpha',\beta} - \frac{1}{N}\delta_{\alpha,\beta}\delta_{\alpha',\beta'}\right),$$

to show that in the unconstrained Hilbert space,

$$\frac{2J}{N}\sum_{a=1}^{N^2-1} \hat{T}_i^{a,c}\hat{T}_i^{a,f} = -\frac{J}{2N}\sum_i \left(\hat{D}_i^{\dagger}\hat{D}_i + \hat{D}_i\hat{D}_i^{\dagger}\right) + \frac{J}{N}\left(\frac{\hat{n}_i^c}{2} + \frac{\hat{n}_i^f}{2} - \frac{\hat{n}_i^c\hat{n}_i^f}{N}\right)$$

with

$$\hat{D}_i^{\dagger} = \sum_{\sigma=1}^{N} \hat{c}_{i,\sigma}^{\dagger}\hat{f}_{i,\sigma}.$$

In the constrained Hilbert space, $\hat{n}_i^f = N/2$, the above gives:

$$\frac{2J}{N}\sum_{a=1}^{N^2-1} \hat{T}_i^{a,c}\hat{T}_i^{a,f} = -\frac{J}{4N}\left[\left(\hat{D}_i^{\dagger} + \hat{D}_i\right)^2 + \left(i\hat{D}_i^{\dagger} - i\hat{D}_i\right)^2\right] + \frac{J}{4}. \tag{200}$$

The perfect square form complies with the requirements of ALF. We still have to impose the constraint. To do so, we work in the unconstrained Hilbert space and add a Hubbard $U$-term on the f-orbitals. With this addition, the Hamiltonian we simulate reads:

$$\hat{H}_{\text{QMC}} = -t\sum_{\langle i,j\rangle}\sum_{\sigma=1}^{N}\left(\hat{c}_{i,\sigma}^{\dagger} e^{\frac{2\pi i}{\Phi_0}\int_i^j \boldsymbol{A}\cdot d\boldsymbol{l}}\hat{c}_{j,\sigma} + \text{H.c.}\right) - \mu\sum_{i,\sigma}\hat{c}_{i,\sigma}^{\dagger}\hat{c}_{i,\sigma} + \frac{U_c}{N}\sum_i\left(\hat{n}_i^c - \frac{N}{2}\right)^2$$

$$- \frac{J}{4N}\left[\left(\hat{D}_i^{\dagger} + \hat{D}_i\right)^2 + \left(i\hat{D}_i^{\dagger} - i\hat{D}_i\right)^2\right] + \frac{U_f}{N}\sum_i\left(\hat{n}_i^f - \frac{N}{2}\right)^2. \tag{201}$$

The key point for the efficiency of the code, is to see that

$$\left[\hat{H}_{\text{QMC}}, \left(\hat{n}_i^f - \frac{N}{2}\right)^2\right] = 0 \tag{202}$$

such that the constraint is implemented efficiently. In fact, for the finite temperature code at inverse temperature $\beta$, the unphysical Hilbert space is suppressed by a factor $e^{-\beta U_f/N}$.

### The SU(2) case

The SU(2) case is special and allows for a more efficient implementation than the one described above. The key point is that for the SU(2) case, the Hubbard term is related to the fermion parity,

$$\left(\hat{n}_i^f - 1\right)^2 = \frac{(-1)^{\hat{n}_i^f} + 1}{2} \tag{203}$$

such that we can omit the *current*-term $\left(i\hat{D}_i^\dagger - i\hat{D}_i\right)^2$ without violating Eq. (202). As in Refs. [31, 32, 148], the Hamiltonian that one will simulate reads:

$$\hat{\mathcal{H}} = \underbrace{-t \sum_{\langle i,j\rangle,\sigma} \left(\hat{c}_{i,\sigma}^\dagger e^{\frac{2\pi i}{\Phi_0}\int_i^j \boldsymbol{A}\cdot d\boldsymbol{l}} \hat{c}_{j,\sigma} + \text{H.c.}\right) + \frac{U_c}{2}\sum_i (\hat{n}_i^c - 1)^2}_{\equiv \hat{\mathcal{H}}_{tU_c}}$$

$$-\frac{J}{4}\sum_i \left(\sum_\sigma \hat{c}_{i,\sigma}^\dagger \hat{f}_{i,\sigma} + \hat{f}_{i,\sigma}^\dagger \hat{c}_{i,\sigma}\right)^2 + \underbrace{\frac{U_f}{2}\sum_i \left(\hat{n}_i^f - 1\right)^2}_{\equiv \hat{\mathcal{H}}_{U_f}}. \quad (204)$$

The relation to the Kondo lattice model follows from expanding the square of the hybridization to obtain:

$$\hat{\mathcal{H}} = \hat{\mathcal{H}}_{tU_c} + J\sum_{\boldsymbol{i}} \left(\hat{\boldsymbol{S}}_{\boldsymbol{i}}^c \cdot \hat{\boldsymbol{S}}_{\boldsymbol{i}}^f + \hat{\eta}_{\boldsymbol{i}}^{z,c}\cdot\hat{\eta}_{\boldsymbol{i}}^{z,f} - \hat{\eta}_{\boldsymbol{i}}^{x,c}\cdot\hat{\eta}_{\boldsymbol{i}}^{x,f} - \hat{\eta}_{\boldsymbol{i}}^{y,c}\cdot\hat{\eta}_{\boldsymbol{i}}^{y,f}\right) + \hat{\mathcal{H}}_{U_f}, \quad (205)$$

where the $\eta$-operators relate to the spin-operators via a particle-hole transformation in one spin sector:

$$\hat{\eta}_{\boldsymbol{i}}^\alpha = \hat{P}^{-1}\hat{S}_{\boldsymbol{i}}^\alpha\hat{P} \text{ with } \hat{P}^{-1}\hat{c}_{\boldsymbol{i},\uparrow}\hat{P} = (-1)^{i_x+i_y}\hat{c}_{\boldsymbol{i},\uparrow}^\dagger \text{ and } \hat{P}^{-1}\hat{c}_{\boldsymbol{i},\downarrow}\hat{P} = \hat{c}_{\boldsymbol{i},\downarrow}. \quad (206)$$

Since the $\hat{\eta}^f$ and $\hat{S}^f$ operators do not alter the parity $[(-1)^{\hat{n}_{\boldsymbol{i}}^f}]$ of the $f$-sites,

$$\left[\hat{\mathcal{H}}, \hat{\mathcal{H}}_{U_f}\right] = 0. \quad (207)$$

Thereby, and for positive values of $U$, doubly occupied or empty $f$-sites – corresponding to even parity sites – are suppressed by a Boltzmann factor $e^{-\beta U_f/2}$ in comparison to odd parity sites. Thus, essentially, choosing $\beta U_f$ adequately allows one to restrict the Hilbert space to odd parity $f$-sites. In this Hilbert space, $\hat{\eta}^{x,f} = \hat{\eta}^{y,f} = \hat{\eta}^{z,f} = 0$ such that the Hamiltonian (204) reduces to the Kondo lattice model.

### QMC implementation

The name space for this model class reads:

```
&VAR_Kondo                  !! Variables for the Kondo  class
ham_T     = 1.d0            ! Hopping parameter
ham_chem  = 0.d0            ! Chemical potential
ham_Uc    = 0.d0            ! Hubbard interaction  on  c-orbitals Uc
ham_Uf    = 2.d0            ! Hubbard interaction  on  f-orbials  Uf
ham_JK    = 2.d0            ! Kondo Coupling  J
/
```

Aside from the usual observables we have included the scalar observable `Constraint_scal` that measures

$$\left\langle \sum_i \left(\hat{n}_i^f - \frac{N}{2}\right)^2 \right\rangle. \quad (208)$$

$U_f$ has to be chosen large enough such that the above quantity vanishes within statistical uncertainty. For the square lattice, Fig. 8 plots the aforementioned quantity as a function of $U_f$ for the SU(2) model. As apparent $\left\langle \sum_i \left(\hat{n}_i^f - N/2\right)^2 \right\rangle \propto e^{-\beta U_f/2}$.

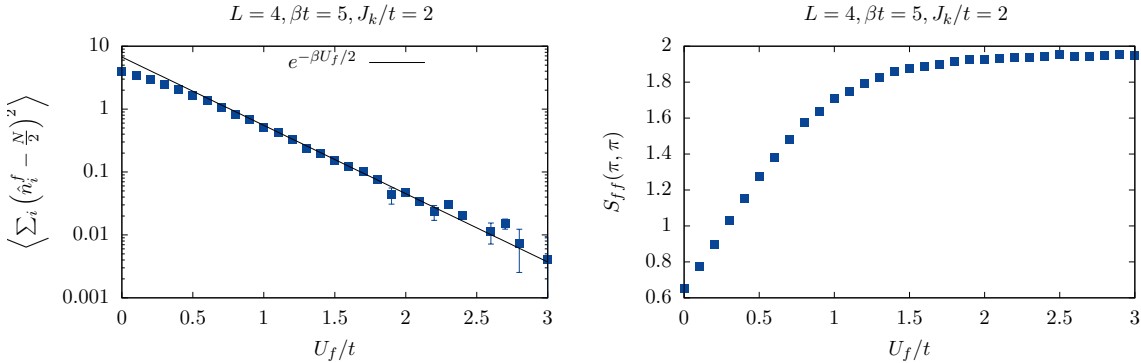

Figure 8: Left: Suppression of charge fluctuations of the f-orbitals as a function of $U_f$. Right: When charge fluctuations on the f-orbitals vanish, quantities such as the Fourier transform of the $f$ spin-spin correlations at $\boldsymbol{q} = (\pi, \pi)$ converge to their KLM value. Typically, for the SU(2) case, $\beta U_f > 10$ suffices to reach convergent results. The pyALF script used to produce the data of the plot can be found in `Kondo.py`

### 9.4  Models with long range Coulomb interactions `Hamiltonian_LRC_smod.F90`

The model we consider here is defined for `N_FL=1`, arbitrary values of `N_SUN` and all the predefined lattices. It reads:

$$\hat{H} = \sum_{i,j} \sum_{\sigma=1}^{N} T_{i,j} \hat{c}^{\dagger}_{i,\sigma} e^{\frac{2\pi i}{\Phi_0} \int_i^j \boldsymbol{A}(\boldsymbol{l})d\boldsymbol{l}} \hat{c}_{j,\sigma} + \frac{1}{N} \sum_{i,j} \left(\hat{n}_i - \frac{N}{2}\right) V_{i,j} \left(\hat{n}_j - \frac{N}{2}\right) - \mu \sum_i \hat{n}_i. \quad (209)$$

In the above, $i = (\boldsymbol{i}, \boldsymbol{\delta}_i)$ and $j = (\boldsymbol{j}, \boldsymbol{\delta}_j)$ are super-indices encoding the unit-cell and orbital and $\hat{n}_i = \sum_{\sigma=1}^N \hat{c}^{\dagger}_{i,\sigma} \hat{c}_{i,\sigma}$ For simplicity, the interaction is specified by two parameters, $U$ and $\alpha$ that monitor the strength of the onsite interaction and the magnitude of the Coulomb tail respectively:

$$V_{i,j} \equiv V(\boldsymbol{i} + \boldsymbol{\delta}_i, \boldsymbol{j} + \boldsymbol{\delta}_j) = U \begin{cases} 1 & \text{if } i = j \\ \frac{\alpha\, d_{\min}}{||\boldsymbol{i}-\boldsymbol{j}+\boldsymbol{\delta}_i-\boldsymbol{\delta}_j||} & \text{otherwise .} \end{cases} \quad (210)$$

Here $d_{\min}$ is the minimal distance between two orbitals. On a torus, some care has be taken in defining the distance. Namely, with the lattice size given by the vectors $\boldsymbol{L}_1$ and $\boldsymbol{L}_2$ (see Sec. 8.1),

$$||\boldsymbol{i}|| = \min_{n_1,n_2 \in \mathbb{Z}} |\boldsymbol{i} - n_1 \boldsymbol{L}_1 - n_2 \boldsymbol{L}_2|. \quad (211)$$

The implementation of the model follows Ref. [51], but supports various lattice geometries. We use the following HS decomposition:

$$e^{-\Delta\tau \hat{H}_V} \propto \int \prod_i d\phi_i e^{-\frac{N\Delta\tau}{4} \sum_{i,j} \phi_i V_{i,j}^{-1} \phi_j - \sum_i i\Delta\tau \phi_i \left(\hat{n}_i - \frac{N}{2}\right)}, \quad (212)$$

where $\phi_i$ is a real variable, $V$ is symmetric and, importantly, has to be positive definite for the Gaussian integration to be defined. The partition function reads:

$$Z \propto \int \prod_i d\phi_{i,\tau} \overbrace{e^{-\frac{N\Delta\tau}{4}\sum_{i,j}\phi_{i,\tau}V_{i,j}^{-1}\phi_{j,\tau}}}^{W_B(\phi)} \mathrm{Tr}\underbrace{\left[\prod_\tau e^{-\Delta\tau\hat{H}_T}e^{-\sum_i i\Delta\tau\phi_{i,\tau}\left(\hat{n}_i-\frac{N}{2}\right)}\right]}_{W_F(\phi)}, \tag{213}$$

such that the weight splits into bosonic and fermionic parts.

For the update, it is convenient to work in a basis where $V$ is diagonal:

$$\mathrm{Diag}\left(\lambda_1,\cdots,\lambda_{\mathtt{Ndim}}\right) = O^T V O \tag{214}$$

with $O^T O = 1$ and define:

$$\eta_{i,\tau} = \sum_j O_{i,j}^T \phi_{j,\tau}. \tag{215}$$

On a given time slice $\tau_u$ we propose a new field configuration with the probability:

$$T^0(\eta \to \eta') = \begin{cases} \prod_i \left[PP_B(\eta'_{i,\tau_u}) + (1-P)\delta(\eta_{i,\tau_u} - \eta'_{i,\tau_u})\right] & \text{for } \tau = \tau_u \\ \delta(\eta_{i,\tau} - \eta'_{i,\tau}) & \text{for } \tau \neq \tau_u \end{cases} \tag{216}$$

where

$$P_B(\eta_{i,\tau}) \propto e^{-\frac{N\Delta\tau}{4\lambda_i}\eta_{i,\tau}^2}, \tag{217}$$

$P \in [0,1]$ and $\delta$ denotes the Dirac $\delta$-function. That is, we carry out simple sampling of the field with probability $P$ and leave the field unchanged with probability $(1-P)$. $P$ is a free parameter that does not change the final result but that allows one to adjust the acceptance. We then use the Metropolis-Hasting acceptance-rejection scheme and accept the move with probability

$$\min\left(\frac{T^0(\eta' \to \eta)W_B(\eta')W_F(\eta')}{T^0(\eta \to \eta')W_B(\eta)W_F(\eta)}, 1\right) = \min\left(\frac{W_F(\eta')}{W_F(\eta)}, 1\right), \tag{218}$$

where

$$W_B(\eta) = e^{-\frac{N\Delta\tau}{4}\sum_{i,\tau}\eta_{i,\tau}^2/\lambda_i} \quad \text{and} \quad W_F(\eta) = \mathrm{Tr}\left[\prod_\tau e^{-\Delta\tau\hat{H}_T}e^{-\sum_{i,j}i\Delta\tau O_{i,j}\eta_{j,\tau}\left(\hat{n}_i-\frac{N}{2}\right)}\right]. \tag{219}$$

Since a local change on a single time slice in the $\eta$ basis corresponds to a non-local space update in the $\phi$ basis, we use the routine for global updates in space to carry out the update (see Sec. 2.2.3).

**QMC implementation**

The name space for this model class reads:

```
&VAR_LRC                    !! Variables for the  Long Range Coulomb class
ham_T          = 1.0        ! Specifies the hopping and chemical potential
ham_T2         = 1.0        ! For bilayer systems
ham_Tperp      = 1.0        ! For bilayer systems
ham_chem       = 1.0        ! Chemical potential
ham_U          = 4.0        ! On-site interaction
ham_alpha      = 0.1        ! Coulomb tail magnitude
Percent_change = 0.1        ! Parameter P
/
```

By setting $\alpha$ to zero we can test this code against the Hubbard code. For a $4 \times 4$ square lattice at $\beta t = 5$, $U/t = 4$, and half-band filling, `Hamiltonian_Hubbard_smod.F90` gives $E = -13.1889 \pm 0.0017$ and `Hamiltonian_LRC_smod.F90`, $E = -13.199 \pm 0.040$. Note that for the Hubbard code we have used the default `Mz = .True.`. This option breaks SU(2) spin symmetry for a given HS configuration, but produces very precise values of the energy. On the other hand, the LRC code is an SU(2) invariant code (as would be choosing `Mz = .False.`) and produces more fluctuations in the double occupancy. This partly explains the difference in error bars between the two codes. To produce this data, one can run the pyALF python script `LRC.py`.

### 9.5 $\mathbf{Z}_2$ lattice gauge theories coupled to fermion and $\mathbf{Z}_2$ matter `Hamiltonian_-Z2_smod.F90`

The Hamiltonian we will consider here reads

$$
\begin{aligned}
\hat{H} = &-t_{\mathbf{Z}_2} \sum_{\langle \boldsymbol{i},\boldsymbol{j} \rangle, \sigma} \hat{\sigma}^z_{\langle \boldsymbol{i},\boldsymbol{j} \rangle} \left( \hat{\Psi}^\dagger_{\boldsymbol{i},\sigma} \hat{\Psi}_{\boldsymbol{j},\sigma} + \text{H.c.} \right) - \mu \sum_{\boldsymbol{i},\sigma} \hat{\Psi}^\dagger_{\boldsymbol{i},\sigma} \hat{\Psi}_{\boldsymbol{i},\sigma} - g \sum_{\langle \boldsymbol{i},\boldsymbol{j} \rangle} \hat{\sigma}^x_{\langle \boldsymbol{i},\boldsymbol{j} \rangle} \\
&+ K \sum_{\square} \prod_{\langle \boldsymbol{i},\boldsymbol{j} \rangle \in \partial\square} \hat{\sigma}^z_{\langle \boldsymbol{i},\boldsymbol{j} \rangle} + J \sum_{\langle \boldsymbol{i},\boldsymbol{j} \rangle} \hat{\tau}^z_{\boldsymbol{i}} \hat{\sigma}^z_{\langle \boldsymbol{i},\boldsymbol{j} \rangle} \hat{\tau}^z_{\boldsymbol{j}} - h \sum_{\boldsymbol{i}} \hat{\tau}^x_{\boldsymbol{i}} \\
&- t \sum_{\langle \boldsymbol{i},\boldsymbol{j} \rangle, \sigma} \hat{\tau}^z_{\boldsymbol{i}} \hat{\tau}^z_{\boldsymbol{j}} \left( \hat{\Psi}^\dagger_{\boldsymbol{i},\sigma} \hat{\Psi}_{\boldsymbol{j},\sigma} + \text{H.c.} \right) + \frac{U}{N} \sum_{\boldsymbol{i}} \left[ \sum_\sigma \left( \hat{\Psi}^\dagger_{\boldsymbol{i},\sigma} \hat{\Psi}_{\boldsymbol{i},\sigma} - 1/2 \right) \right]^2.
\end{aligned}
\tag{220}
$$

The model is defined on a square lattice, and describes fermions,

$$
\left\{ \hat{\Psi}^\dagger_{\boldsymbol{i},\sigma}, \hat{\Psi}_{\boldsymbol{j},\sigma'} \right\} = \delta_{\boldsymbol{i},\boldsymbol{j}} \delta_{\sigma,\sigma'}, \ \left\{ \hat{\Psi}_{\boldsymbol{i},\sigma}, \hat{\Psi}_{\boldsymbol{j},\sigma'} \right\} = 0,
\tag{221}
$$

coupled to bond gauge fields,

$$
\hat{\sigma}^z_{\langle \boldsymbol{i},\boldsymbol{j} \rangle} = \begin{bmatrix} 1 & 0 \\ 0 & -1 \end{bmatrix}, \hat{\sigma}^x_{\langle \boldsymbol{i},\boldsymbol{j} \rangle} = \begin{bmatrix} 0 & 1 \\ 1 & 0 \end{bmatrix}, \left\{ \hat{\sigma}^z_{\langle \boldsymbol{i},\boldsymbol{j} \rangle}, \hat{\sigma}^x_{\langle \boldsymbol{i}',\boldsymbol{j}' \rangle} \right\} = 2 \left( 1 - \delta_{\langle \boldsymbol{i},\boldsymbol{j} \rangle, \langle \boldsymbol{i}',\boldsymbol{j}' \rangle} \right) \hat{\sigma}^z_{\langle \boldsymbol{i},\boldsymbol{j} \rangle} \hat{\sigma}^x_{\langle \boldsymbol{i}',\boldsymbol{j}' \rangle}
\tag{222}
$$

and $\mathbf{Z}_2$ matter fields:

$$
\hat{\tau}^z_{\boldsymbol{i}} = \begin{bmatrix} 1 & 0 \\ 0 & -1 \end{bmatrix}, \quad \hat{\tau}^x_{\boldsymbol{i}} = \begin{bmatrix} 0 & 1 \\ 1 & 0 \end{bmatrix}, \quad \{ \hat{\tau}^z_{\boldsymbol{i}}, \hat{\tau}^x_{\boldsymbol{i}'} \} = 2 \left( 1 - \delta_{\boldsymbol{i},\boldsymbol{i}'} \right) \hat{\tau}^z_{\boldsymbol{i}} \hat{\tau}^x_{\boldsymbol{i}'}.
\tag{223}
$$

Fermions, gauge fields and $\mathbf{Z}_2$ matter fields commute with each other.

Importantly, the model has a local $\mathbf{Z}_2$ symmetry. Consider:

$$
\hat{Q}_{\boldsymbol{i}} = (-1)^{\sum_\sigma \hat{\Psi}^\dagger_{\boldsymbol{i},\sigma} \hat{\Psi}_{\boldsymbol{i},\sigma}} \ \hat{\tau}^x_{\boldsymbol{i}} \ \hat{\sigma}^x_{\boldsymbol{i},\boldsymbol{i}+\boldsymbol{a}_x} \hat{\sigma}^x_{\boldsymbol{i},\boldsymbol{i}-\boldsymbol{a}_x} \hat{\sigma}^x_{\boldsymbol{i},\boldsymbol{i}+\boldsymbol{a}_y} \hat{\sigma}^x_{\boldsymbol{i}}.
\tag{224}
$$

One can then show that $\hat{Q}^2_{\boldsymbol{i}} = 1$ and that

$$
\left[ \hat{Q}_{\boldsymbol{i}}, \hat{H} \right] = 0.
\tag{225}
$$

The above allows us to assign $\mathbf{Z}_2$ charges to the operators. Since $\left\{ \hat{Q}_{\boldsymbol{i}}, \hat{\Psi}^\dagger_{\boldsymbol{i},\sigma} \right\} = 0$ we can assign a $\mathbf{Z}_2$ charge to the fermions. Equivalently $\hat{\tau}^z_{\boldsymbol{i}}$ has a $\mathbf{Z}_2$ charge and $\hat{\sigma}^z_{\boldsymbol{i},\boldsymbol{j}}$ carries $\mathbf{Z}_2$ charges at its ends. Since the total fermion number is conserved, we can assign an electric charge to the fermions. Finally, the model has an SU(N) color symmetry. In fact, at zero chemical potential and $U = 0$, the symmetry is enhanced to $O(2N)$ [21]. Aspects of this Hamiltonian were investigated in Refs. [21, 25, 26, 28–30] and we refer the interested user to these papers for a discussion of the phases and phase transitions supported by the model.

### QMC implementation

The name space for this model class reads:

```
&VAR_Z2_Matter                !! Variables for the Z_2 class
ham_T           = 1.0         ! Hopping for fermions
ham_TZ2         = 1.0         ! Hopping for orthogonal fermions
ham_chem        = 0.0         ! Chemical potential for fermions
ham_U           = 0.0         ! Hubbard for fermions
Ham_J           = 1.0         ! Hopping Z2 matter fields
Ham_K           = 1.0         ! Plaquette term for gauge fields
Ham_h           = 1.0         ! sigma^x-term for matter
Ham_g           = 1.0         ! tau^x-term for gauge
Dtau            = 0.1d0       ! Thereby Ltrot=Beta/dtau
Beta            = 10.d0       ! Inverse temperature
Projector       = .False.     ! To enable projective code
Theta           = 10.0        ! Projection parameter
/
```

We note that the implementation is such that if `Ham_T=0` (`Ham_TZ2=0`) then all the terms involving the matter field ($\mathbf{Z}_2$ gauge field) are automatically set to zero. We warn the user that autocorrelation and warmup times can be large for this model class. At this point, the model is only implemented for the square lattice and does not support a symmetric Trotter decomposition.

The key point to implement the model is to define a new bond variable:

$$\hat{\mu}^z_{\langle \boldsymbol{i}, \boldsymbol{j} \rangle} = \hat{\tau}^z_{\boldsymbol{i}} \hat{\tau}^z_{\boldsymbol{j}}. \tag{226}$$

By construction, the $\hat{\mu}^z_{\langle \boldsymbol{i}, \boldsymbol{j} \rangle}$ bond variables have a zero flux constraint:

$$\hat{\mu}^z_{\langle \boldsymbol{i}, \boldsymbol{i}+\boldsymbol{a}_x \rangle} \hat{\mu}^z_{\langle \boldsymbol{i}+\boldsymbol{a}_x, \boldsymbol{i}+\boldsymbol{a}_x+\boldsymbol{a}_y \rangle} \hat{\mu}^z_{\langle \boldsymbol{i}+\boldsymbol{a}_x+\boldsymbol{a}_y, \boldsymbol{i}+\boldsymbol{a}_y \rangle} \hat{\mu}^z_{\langle \boldsymbol{i}+\boldsymbol{a}_y, \boldsymbol{i} \rangle} = 1. \tag{227}$$

Consider a basis where $\hat{\mu}^z_{\langle \boldsymbol{i}, \boldsymbol{j} \rangle}$ and $\hat{\tau}^z_{\boldsymbol{i}}$ are diagonal with eigenvalues $\mu_{\langle \boldsymbol{i}, \boldsymbol{j} \rangle}$ and $\tau_{\boldsymbol{i}}$ respectively. The map from $\{\tau_{\boldsymbol{i}}\}$ to $\{\mu_{\langle \boldsymbol{i}, \boldsymbol{j} \rangle}\}$ is unique. The reverse however is valid only up to a global sign. To pin down this sign (and thereby the relative signs between different time slices) we store the fields $\mu_{\langle \boldsymbol{i}, \boldsymbol{j} \rangle}$ at every time slice as well as the value of the Ising field at a reference site $\tau_{\boldsymbol{i}=\boldsymbol{0}}$. Within the ALF, this can be done by adding a dummy operator in the `Op_V` list to carry this degree of freedom. With this extra degree of freedom we can switch between the two representations without loosing any information. To compute the Ising part of the action it is certainly more transparent to work with the $\{\tau_{\boldsymbol{i}}\}$ variables. For the fermion determinant, the $\{\mu_{\langle \boldsymbol{i}, \boldsymbol{j} \rangle}\}$ are more convenient.

Since flipping $\hat{\tau}^z_{\boldsymbol{i}}$ amounts to changing the sign of the four bond variables emanating from site $\boldsymbol{i}$, the identity:

$$\hat{\tau}^x_{\boldsymbol{i}} = \hat{\mu}^x_{\boldsymbol{i}, \boldsymbol{i}+\boldsymbol{a}_x} \hat{\mu}^x_{\boldsymbol{i}+\boldsymbol{a}_x, \boldsymbol{i}+\boldsymbol{a}_x+\boldsymbol{a}_y} \hat{\mu}^x_{\boldsymbol{i}+\boldsymbol{a}_x+\boldsymbol{a}_y, \boldsymbol{i}+\boldsymbol{a}_y} \tag{228}$$

holds. Note that $\left\{ \hat{\mu}^z_{\langle \boldsymbol{i}, \boldsymbol{j} \rangle}, \hat{\mu}^x_{\langle \boldsymbol{i}', \boldsymbol{j}' \rangle} \right\} = 2 \left( 1 - \delta_{\langle \boldsymbol{i}, \boldsymbol{j} \rangle, \langle \boldsymbol{i}', \boldsymbol{j}' \rangle} \right) \hat{\mu}^z_{\langle \boldsymbol{i}, \boldsymbol{j} \rangle} \hat{\mu}^x_{\langle \boldsymbol{i}', \boldsymbol{j}' \rangle}$, such that applying $\hat{\mu}^x_{\langle \boldsymbol{i}, \boldsymbol{j} \rangle}$ on an eigenstate of $\hat{\mu}^z_{\langle \boldsymbol{i}, \boldsymbol{j} \rangle}$ flips the field.

The model can then be written as:

$$\hat{H} = - t_{\mathbf{Z}_2} \sum_{\langle \boldsymbol{i},\boldsymbol{j}\rangle,\sigma} \hat{\sigma}^z_{\langle \boldsymbol{i},\boldsymbol{j}\rangle} \left(\hat{\Psi}^\dagger_{\boldsymbol{i},\sigma}\hat{\Psi}_{\boldsymbol{j},\sigma} + \text{H.c.}\right) - \mu \sum_{\boldsymbol{i},\sigma} \hat{\Psi}^\dagger_{\boldsymbol{i},\sigma}\hat{\Psi}_{\boldsymbol{i},\sigma} - g \sum_{\langle \boldsymbol{i},\boldsymbol{j}\rangle} \hat{\sigma}^x_{\langle \boldsymbol{i},\boldsymbol{j}\rangle} + K \sum_{\square} \prod_{\langle \boldsymbol{i},\boldsymbol{j}\rangle \in \partial\square} \hat{\sigma}^z_{\langle \boldsymbol{i},\boldsymbol{j}\rangle}$$

$$+ J \sum_{\langle \boldsymbol{i},\boldsymbol{j}\rangle} \hat{\mu}^z_{\langle \boldsymbol{i},\boldsymbol{j}\rangle}\hat{\sigma}^z_{\langle \boldsymbol{i},\boldsymbol{j}\rangle} - h \sum_{\boldsymbol{i}} \hat{\mu}^x_{\boldsymbol{i},\boldsymbol{i}+\boldsymbol{a}_x}\hat{\mu}^x_{\boldsymbol{i}+\boldsymbol{a}_x,\boldsymbol{i}+\boldsymbol{a}_x+\boldsymbol{a}_y}\hat{\mu}^x_{\boldsymbol{i}+\boldsymbol{a}_x+\boldsymbol{a}_y,\boldsymbol{i}+\boldsymbol{a}_y}\hat{\mu}^x_{\boldsymbol{i}+\boldsymbol{a}_y,\boldsymbol{i}}$$

$$- t \sum_{\langle \boldsymbol{i},\boldsymbol{j}\rangle,\sigma} \hat{\mu}^z_{\boldsymbol{i},\boldsymbol{j}} \left(\hat{\Psi}^\dagger_{\boldsymbol{i},\sigma}\hat{\Psi}_{\boldsymbol{j},\sigma} + \text{H.c.}\right) + \frac{U}{N} \sum_{\boldsymbol{i}} \left[\sum_\sigma (\hat{\Psi}^\dagger_{\boldsymbol{i},\sigma}\hat{\Psi}_{\boldsymbol{i},\sigma} - 1/2)\right]^2 \tag{229}$$

subject to the constraint of Eq. (227).

To formulate the Monte Carlo, we work in a basis in which $\hat{\mu}^z_{\langle \boldsymbol{i},\boldsymbol{j}\rangle}$, $\hat{\tau}^z_{\mathbf{0}}$ and $\hat{\sigma}^z_{\langle \boldsymbol{i},\boldsymbol{j}\rangle}$ are diagonal:

$$\hat{\mu}^z_{\langle \boldsymbol{i},\boldsymbol{j}\rangle}|\underline{s}\rangle = \mu_{\langle \boldsymbol{i},\boldsymbol{j}\rangle}|\underline{s}\rangle, \quad \hat{\sigma}^z_{\langle \boldsymbol{i},\boldsymbol{j}\rangle}|\underline{s}\rangle = \sigma_{\langle \boldsymbol{i},\boldsymbol{j}\rangle}|\underline{s}\rangle, \quad \hat{\tau}^z_{\mathbf{0}}|\underline{s}\rangle = \tau_{\mathbf{0}}|\underline{s}\rangle \tag{230}$$

with $\underline{s} = \left(\{\mu_{\langle \boldsymbol{i},\boldsymbol{j}\rangle}\}, \{\sigma_{\langle \boldsymbol{i},\boldsymbol{j}\rangle}\}, \tau_{\mathbf{0}}\right)$. In this basis,

$$Z = \sum_{\underline{s}_1,\cdots,\underline{s}_{L_\tau}} e^{-S_0(\{\underline{s}_\tau\})}\text{Tr}_F\left[\prod_{\tau=1}^{L_\tau} e^{-\Delta\tau \hat{H}_F(\underline{s}_\tau)}\right], \tag{231}$$

where

$$S_0(\{\underline{s}_\tau\}) = -\ln\left[\prod_{\tau=1}^{L_\tau} \langle \underline{s}_{\tau+1}|e^{-\Delta\tau \hat{H}_I}|\underline{s}_\tau\rangle\right],$$

$$\hat{H}_I = -g\sum_{\langle \boldsymbol{i},\boldsymbol{j}\rangle} \hat{\sigma}^x_{\langle \boldsymbol{i},\boldsymbol{j}\rangle} + K\sum_{\square}\prod_{\langle \boldsymbol{i},\boldsymbol{j}\rangle \in \partial\square} \hat{\sigma}^z_{\langle \boldsymbol{i},\boldsymbol{j}\rangle} + J\sum_{\langle \boldsymbol{i},\boldsymbol{j}\rangle} \hat{\mu}^z_{\langle \boldsymbol{i},\boldsymbol{j}\rangle}\hat{\sigma}^z_{\langle \boldsymbol{i},\boldsymbol{j}\rangle}$$

$$- h\sum_{\boldsymbol{i}} \hat{\mu}^x_{\boldsymbol{i},\boldsymbol{i}+\boldsymbol{a}_x}\hat{\mu}^x_{\boldsymbol{i}+\boldsymbol{a}_x,\boldsymbol{i}+\boldsymbol{a}_x+\boldsymbol{a}_y}\hat{\mu}^x_{\boldsymbol{i}+\boldsymbol{a}_x+\boldsymbol{a}_y,\boldsymbol{i}+\boldsymbol{a}_y}$$

and

$$\hat{H}_F(\underline{s}) = - t_{\mathbf{Z}_2} \sum_{\langle \boldsymbol{i},\boldsymbol{j}\rangle,\sigma} \sigma_{\langle \boldsymbol{i},\boldsymbol{j}\rangle}\left(\hat{\Psi}^\dagger_{\boldsymbol{i},\sigma}\hat{\Psi}_{\boldsymbol{j},\sigma} + \text{H.c.}\right) - \mu\sum_{\boldsymbol{i},\sigma} \hat{\Psi}^\dagger_{\boldsymbol{i},\sigma}\hat{\Psi}_{\boldsymbol{i},\sigma}$$

$$- t \sum_{\langle \boldsymbol{i},\boldsymbol{j}\rangle,\sigma} \mu_{\boldsymbol{i},\boldsymbol{j}}\left(\hat{\Psi}^\dagger_{\boldsymbol{i},\sigma}\hat{\Psi}_{\boldsymbol{j},\sigma} + \text{H.c.}\right) + \frac{U}{N}\sum_{\boldsymbol{i}}\left[\sum_\sigma (\hat{\Psi}^\dagger_{\boldsymbol{i},\sigma}\hat{\Psi}_{\boldsymbol{i},\sigma} - 1/2)\right]^2.$$

In the above, $|\underline{s}_{L_\tau+1}\rangle = |\underline{s}_1\rangle$. With a further HS transformation of the Hubbard term (see Sec. 8.3.1) the model is readily implemented in the ALF. Including this HS field, $l$, [see Eq. (11)] yields the configuration space:

$$C = \left(\{\mu_{\langle \boldsymbol{i},\boldsymbol{j}\rangle,\tau}\}, \{\sigma_{\langle \boldsymbol{i},\boldsymbol{j}\rangle,\tau}\}, \{\tau_{\mathbf{0},\tau}\}, \{l_{\boldsymbol{i},\tau}\}\right) \tag{232}$$

where the variables $\mu$, $\tau$ and $\sigma$ take the values $\pm 1$ and $l$ the values $\pm 1, \pm 2$.

The initial configuration as well as the moves have to respect the zero flux constraint of Eq. (227). Therefore, single spin flips of the $\mu$ fields are prohibited and the minimal move one can carry out on a given time slice is the following. We randomly choose a site $\boldsymbol{i}$ and propose a move where: $\mu_{\boldsymbol{i},\boldsymbol{i}+\boldsymbol{a}_x} \to -\mu_{\boldsymbol{i},\boldsymbol{i}+\boldsymbol{a}_x}$, $\mu_{\boldsymbol{i},\boldsymbol{i}-\boldsymbol{a}_x} \to -\mu_{\boldsymbol{i},\boldsymbol{i}-\boldsymbol{a}_x}$, $\mu_{\boldsymbol{i},\boldsymbol{i}+\boldsymbol{a}_y} \to -\mu_{\boldsymbol{i},\boldsymbol{i}+\boldsymbol{a}_y}$ and $\mu_{\boldsymbol{i},\boldsymbol{i}-\boldsymbol{a}_y} \to -\mu_{\boldsymbol{i},\boldsymbol{i}-\boldsymbol{a}_y}$. One can carry out such moves by using the global move in real space option presented in Sec. 2.2.3 and 5.7.1.

### 9.5.1 Projective approach

The program also supports a zero temperature implementation. Our choice of the trial wave function does not break any symmetries of the model and reads:

$$|\Psi_T\rangle = |\Psi_T^F\rangle \otimes_{\langle i,j\rangle} |+\rangle_{\langle i,j\rangle} \otimes_i |+\rangle_i. \tag{233}$$

For the fermion part we use a Fermi sea with small dimerization to avoid the negative sign problem at half-filling (see Sec. 7.5). For the Ising part the trial wave function is diagonal in the $\hat{\sigma}_{\langle i,j\rangle}^x$ and $\hat{\tau}_i^x$ operators:

$$\hat{\sigma}_{\langle i,j\rangle}^x |+\rangle_{\langle i,j\rangle} = |+\rangle_{\langle i,j\rangle} \quad \text{and} \quad \hat{\tau}_i^x |+\rangle_i = |+\rangle_i. \tag{234}$$

An alternative choice would be to choose a charge density wave fermionic trial wave function. This violates the partial particle-hole symmetry of the model at $U = \mu = 0$ and effectively imposes the constraint $\hat{Q}_i = 1$.

### 9.5.2 Observables

Apart from the standard observables discussed in Sec. 8.4 the code computes additionally

$$\langle \hat{\sigma}_{\langle i,j\rangle}^x \rangle \quad \text{and} \quad \langle \hat{\tau}_j^x \rangle,$$

which are written to file X_scal;

$$\left\langle \hat{\sigma}_{\langle i,i+a_x\rangle}^z \hat{\sigma}_{\langle i+a_x,i+a_x+a_y\rangle}^z \hat{\sigma}_{\langle i+a_x+a_y,i+a_y\rangle}^z \hat{\sigma}_{\langle i+a_y,i\rangle}^z \right\rangle$$

and

$$\left\langle \hat{\mu}_{\langle i,i+a_x\rangle}^z \hat{\mu}_{\langle i+a_x,i+a_x+a_y\rangle}^z \hat{\mu}_{\langle i+a_x+a_y,i+a_y\rangle}^z \hat{\mu}_{\langle i+a_y,i\rangle}^z \right\rangle,$$

written to file Flux_scal; and also $\langle \hat{Q}_i \rangle$ (file Q_scal). Note that the flux over a plaquette of the $\hat{\mu}_{\langle i,j\rangle}^z$ is equal to unity by construction so that this observable provides a sanity check. The file Q_eq contains the two-point correlation $\langle \hat{Q}_i \hat{Q}_j \rangle - \langle \hat{Q}_i \rangle \langle \hat{Q}_j \rangle$ and Greenf_eq the equal-time fermion Green function $\langle \hat{\tau}_i^z \hat{\Psi}_{i,\sigma}^\dagger \hat{\tau}_j^z \hat{\Psi}_{j,\sigma} \rangle$.

### 9.5.3 A test case: $\mathbf{Z}_2$ slave spin formulation of the SU(2) Hubbard model

In this subsection, we demonstrate that the code can be used to simulate the attractive Hubbard model in the $\mathbf{Z}_2$-slave spin formulation [149]:

$$\hat{H} = -t \sum_{\langle i,j\rangle,\sigma} \hat{c}_{i,\sigma}^\dagger \hat{c}_{j,\sigma} - U \sum_i (\hat{n}_{i,\uparrow} - 1/2)(\hat{n}_{i,\downarrow} - 1/2). \tag{235}$$

In the $\mathbf{Z}_2$ slave spin representation, the physical fermion, $\hat{c}_{i,\sigma}$, is fractionalized into an Ising spin carrying $\mathbf{Z}_2$ charge and a fermion, $\hat{\Psi}_{i,\sigma}$, carrying $\mathbf{Z}_2$ and global $U(1)$ charge:

$$\hat{c}_{i,\sigma}^\dagger = \hat{\tau}_i^z \hat{\Psi}_{i,\sigma}^\dagger. \tag{236}$$

To ensure that we remain in the correct Hilbert space, the constraint:

$$\hat{\tau}_i^x - (-1)^{\sum_\sigma \hat{\Psi}_{i,\sigma}^\dagger \hat{\Psi}_{i,\sigma}} = 0 \tag{237}$$

has to be imposed locally. Since $(\tau_i^x)^2 = 1$, the latter is equivalent to

$$\hat{Q}_{\boldsymbol{i}} = \tau_{\boldsymbol{i}}^x (-1)^{\sum_\sigma \hat{\Psi}_{i,\sigma}^\dagger \hat{\Psi}_{i,\sigma}} = 1. \tag{238}$$

Using

$$(-1)^{\sum_\sigma \hat{\Psi}_{i,\sigma}^\dagger \hat{\Psi}_{i,\sigma}} = \prod_\sigma (1 - 2\hat{\Psi}_{\boldsymbol{i},\sigma}^\dagger \hat{\Psi}_{\boldsymbol{i},\sigma}) = 4 \prod_\sigma (\hat{c}_{\boldsymbol{i},\sigma}^\dagger \hat{c}_{\boldsymbol{i},\sigma} - 1/2), \tag{239}$$

the $\mathbf{Z}_2$ slave spin representation of the Hubbard model now reads:

$$\hat{H}_{\mathbf{Z}_2} = -t \sum_{\langle \boldsymbol{i},\boldsymbol{j} \rangle, \sigma} \hat{\tau}_{\boldsymbol{i}}^z \hat{\tau}_{\boldsymbol{j}}^z \hat{\Psi}_{\boldsymbol{i},\sigma}^\dagger \hat{\Psi}_{\boldsymbol{j},\sigma} - \frac{U}{4} \sum_{\boldsymbol{i}} \hat{\tau}_{\boldsymbol{i}}^x. \tag{240}$$

Importantly, the constraint commutes with Hamiltonian:

$$\left[ \hat{H}_{\mathbf{Z}_2}, \hat{Q}_{\boldsymbol{i}} \right] = 0. \tag{241}$$

Hence one can foresee that the constraint will be dynamically imposed (we expect a finite-temperature Ising phase transition below which $\hat{Q}_{\boldsymbol{i}}$ orders) and that at $T = 0$ on a finite lattice both models should give the same results.

A test run for the $8 \times 8$ lattice at $U/t = 4$ and $\beta t = 40$ gives:

| k | $\langle n_k \rangle_H$ | $\langle n_k \rangle_{H_{\mathbf{Z}_2}}$ |
|---|---|---|
| $(0,0)$ | $1.93348548 \pm 0.00011322$ | $1.93333895 \pm 0.00010405$ |
| $(\pi/4, \pi/4)$ | $1.90120688 \pm 0.00014854$ | $1.90203726 \pm 0.00017943$ |
| $(\pi/2, \pi/2)$ | $0.99942957 \pm 0.00091377$ | $1.00000000 \pm 0.00000000$ |
| $(3\pi/4, 3\pi/4)$ | $0.09905425 \pm 0.00015940$ | $0.09796274 \pm 0.00017943$ |
| $(\pi, \pi)$ | $0.06651452 \pm 0.00011321$ | $0.06666105 \pm 0.00010405$ |

Here a Trotter time step of $\Delta\tau t = 0.05$ was used in order to minimize the systematic error which should be different between the two codes. The Hamiltonian is invariant under a partial particle-hole transformation (see Ref. [21]). Since $\hat{Q}_{\boldsymbol{i}}$ is odd under this transformation, $\langle \hat{Q}_{\boldsymbol{i}} \rangle = 0$. To asses whether the constraint is well imposed, the code, for this special case, computes the correlation function:

$$S_Q(\boldsymbol{q}) = \sum_{\boldsymbol{i}} \langle \hat{Q}_{\boldsymbol{i}} \hat{Q}_{\boldsymbol{0}} \rangle. \tag{242}$$

For the above run we obtain $S_Q(\boldsymbol{q} = \boldsymbol{0}) = 63.4 \pm 1.7$ which, for this $8 \times 8$ lattice, complies with a ferromagnetic ordering of the Ising $\hat{Q}_{\boldsymbol{i}}$ variables. The pyALF python script that produces this data can be found in `Z2_Matter.py`. This code was used in Refs. [28, 29].

## 10  Maximum Entropy

If we want to compare the data we obtain from Monte Carlo simulations with experiments, we must extract spectral information from the imaginary-time output. This can be achieved

through the maximum entropy method (MaxEnt), which generically computes the image $A(\omega)$ for a given data set $g(\tau)$ and kernel $K(\tau, \omega)$:

$$g(\tau) = \int_{\omega_{\text{start}}}^{\omega_{\text{end}}} d\omega K(\tau, \omega) A(\omega). \tag{243}$$

The ALF package includes a standard implementation of the stochastic MaxEnt, as formulated in the article of K. Beach [97], in the module `Libraries/Modules/maxent_stoch_mod.F90`. Its wrapper is found in `Analysis/Max_SAC.F90` and the Green function is read from the output of the `cov_tau.F90` analysis program.

## 10.1   General setup

The stochastic MaxEnt is essentially a parallel-tempering Monte Carlo simulation. For a discrete set of $\tau_i$ points, $i \in 1 \cdots n$, the goodness-of-fit functional, which we take as the energy reads

$$\chi^2(A) = \sum_{i,j=1}^{n} \left[ g(\tau_i) - \overline{g(\tau_i)} \right] C^{-1}(\tau_i, \tau_j) \left[ g(\tau_j) - \overline{g(\tau_j)} \right], \tag{244}$$

with $\overline{g(\tau_i)} = \int d\omega K(\tau_i, \omega) A(\omega)$ and $C$ the covariance matrix. The set of $N_\alpha$ inverse temperatures considered in the parallel tempering is given by $\alpha_m = \alpha_{st} R^m$, for $m = 1 \cdots N_\alpha$ and a constant $R$. The phase space corresponds to all possible spectral functions satisfying a given sum rule and the required positivity. Finally, the partition function reads $Z = \int \mathcal{D}A \, e^{-\alpha \chi^2(A)}$ [97], such that for a given "inverse temperature" $\alpha$, the image is given by:

$$\langle A(\omega) \rangle = \frac{\int \mathcal{D}A \, e^{-\alpha \chi^2(A)} A(\omega)}{\int \mathcal{D}A \, e^{-\alpha \chi^2(A)}}. \tag{245}$$

In the code, the spectral function is parametrized by a set of $N_\gamma$ Dirac $\delta$ functions:

$$A(\omega) = \sum_{i=1}^{N_\gamma} a_i \delta (\omega - \omega_i). \tag{246}$$

To produce a histogram of $A(\omega)$ we divide the frequency range in `Ndis` intervals.

Besides the parameters included in the namelist `VAR_Max_Stoch` set in the file `parameters` (see Sec. 5.7), also the variable `N_cov`, from the namelist `VAR_errors`, is required to run the maxent code. Recalling: `N_cov = 1` (`N_cov = 0`) sets that the covariance will (will not) be taken into account.

### Input files

In addition to the aforementioned parameter file, the MaxEnt program requires the output of the analysis of the time-displaced functions. The program `Anaylsis/ana.out` (see Sec. 6.3) generates, for each $k$-point, a directory named `Variable_name_kx_ky`. In this directory the file `g_kx_ky` contains the required information for the MaxEnt code, which is formatted as follows:

```
<# of tau-points>  <# of bins >  <beta>  <Norb>  <Channel>
do tau = 1, # of tau-points
```

```
    τ,    Σ_α⟨S_{α,α}^{(corr)}(k,τ)⟩,    error
enddo
do tau1 = 1, # of tau-points
  do tau2 = 1, # of tau-points
      C(τ₁, τ₂)
  enddo
enddo
```

**Output files**

The code produces the following output files:

- The files `Aom_n` contains the average spectral function at inverse temperature $\alpha_n$. This corresponds to $\langle A_n(\omega) \rangle = \frac{1}{Z} \int \mathcal{D}A(\omega) \, e^{-\alpha_n \chi^2(A)} A(\omega)$. The file contains three columns: $\omega$, $\langle A_n(\omega) \rangle$, and $\Delta \langle A_n(\omega) \rangle$.

- The files `Aom_ps_n` contain the average image over the inverse temperatures $\alpha_n$ to $\alpha_{N_\gamma}$, see Ref. [97] for more details. Its first three columns have the same meaning as for the files `Aom_n`.

- The file `Green` contains the Green function, obtained from the spectral function through

$$ G(\omega) = -\frac{1}{\pi} \int d\Omega \frac{A(\Omega)}{\omega - \Omega + i\delta} , \tag{247} $$

  where $\delta = \Delta\omega = (\omega_{\text{end}} - \omega_{\text{start}})/\texttt{Ndis}$ and the image corresponds to that of the file `Aom_ps_n` with $n = N_\alpha - 10$. The first column of the `Green` file is a place holder for post-processing. The last three columns correspond to $\omega, \mathrm{Re}\, G(\omega), -\mathrm{Im}\, G(\omega)/\pi$.

- One of the most important output files is `energies`, which lists $\alpha_n, \langle \chi^2 \rangle, \Delta \langle \chi^2 \rangle$.

- `best_fit` gives the values of $a_i$ and $\omega_i$ (recall that $A(\omega) = \sum_{i=1}^{N_\gamma} a_i \delta(\omega - \omega_i)$) corresponding to the last configuration of the lowest temperature run.

- The file `data_out` facilitates crosschecking. It lists $\tau$, $g(\tau)$, $\Delta g(\tau)$, and $\int d\omega K(\tau, \omega) A(\omega)$, where the image corresponds to the best fit (i.e. the lowest temperature). This data should give an indication of how good the fit actually is. Note that `data_out` contains only the data points that have passed the tolerance test.

- Two dump files are also generated, `dump_conf` and `dump_Aom`. Since the MaxEnt is a Monte Carlo code, it is possible to improve the data by continuing a previous simulation. The data in the dump files allow you to do so. These files are only generated if the variable `checkpoint` is set to `.true.`.

The essential question is: Which image should one use? There is no ultimate answer to this question in the context of the stochastic MaxEnt. The only rule of thumb is to consider temperatures for which the $\chi^2$ is comparable to the number of data points.

## 10.2   Single-particle quantities: `Channel=P`

For the single-particle Green function,

$$\langle \hat{c}_k(\tau)\hat{c}_k^\dagger(0)\rangle = \int \mathrm{d}\omega K_p(\tau,\omega)A_p(k,\omega), \tag{248}$$

with

$$K_p(\tau,\omega) = \frac{1}{\pi}\frac{e^{-\tau\omega}}{1+e^{-\beta\omega}} \tag{249}$$

and, in the Lehmann representation,

$$A_p(k,\omega) = \frac{\pi}{Z}\sum_{n,m}e^{-\beta E_n}\left(1+e^{-\beta\omega}\right)|\langle n|c_n|m\rangle|^2\delta\left(E_m - E_n - \omega\right). \tag{250}$$

Here $\left(\hat{H} - \mu\hat{N}\right)|n\rangle = E_n|n\rangle$. Note that $A_p(k,\omega) = -\operatorname{Im}G^{\mathrm{ret}}(k,\omega)$, with

$$G^{\mathrm{ret}}(k,\omega) = -i\int \mathrm{d}t\Theta(t)e^{i\omega t}\langle\{\hat{c}_k(t),\hat{c}_k^\dagger(0)\}\rangle. \tag{251}$$

Finally the sum rule reads

$$\int \mathrm{d}\omega A_p(k,\omega) = \pi\langle\{\hat{c}_k,\hat{c}_k^\dagger\}\rangle = \pi\left(\langle\hat{c}_k(\tau=0)\hat{c}_k^\dagger(0)\rangle + \langle\hat{c}_k(\tau=\beta)\hat{c}_k^\dagger(0)\rangle\right). \tag{252}$$

Using the `Max_Sac.F90` with `Channel="P"` will load the above kernel in the MaxEnt library. In this case the back transformation is set to unity. Note that for each configuration of fields we have $\langle\langle\hat{c}_k(\tau=0)\hat{c}_k^\dagger(0)\rangle\rangle_C + \langle\langle\hat{c}_k(\tau=\beta)\hat{c}_k^\dagger(0)\rangle\rangle_C = \langle\langle\{\hat{c}_k,\hat{c}_k^\dagger\}\rangle\rangle_C = 1$, hence, if both the $\tau=0$ and $\tau=\beta$ data points are included, the covariance matrix will have a zero eigenvalue and the $\chi^2$ measure is not defined. Therefore, for the particle channel the program omits the $\tau=\beta$ data point. There are special particle-hole symmetric cases where the $\tau=0$ data point shows no fluctuations – in such cases the code omits the $\tau=0$ data point as well.

## 10.3   Particle-hole quantities: `Channel=PH`

**Imaginary-time formulation**

For particle-hole quantities such as spin-spin or charge-charge correlations, the kernel reads

$$\langle\hat{S}(q,\tau)\hat{S}(-q,0)\rangle = \frac{1}{\pi}\int \mathrm{d}\omega\frac{e^{-\tau\omega}}{1-e^{-\beta\omega}}\chi''(q,\omega). \tag{253}$$

This follows directly from the Lehmann representation

$$\chi''(q,\omega) = \frac{\pi}{Z}\sum_{n,m}e^{-\beta E_n}|\langle n|\hat{S}(q)|m\rangle|^2\delta(\omega + E_n - E_m)\left(1 - e^{-\beta\omega}\right). \tag{254}$$

Since the linear response to a hermitian perturbation is real, $\chi''(q,\omega) = -\chi''(-q,-\omega)$ and hence $\langle\hat{S}(q,\tau)\hat{S}(-q,0)\rangle$ is a symmetric function around $\beta = \tau/2$ for systems with inversion symmetry – the ones we consider here. When `Channel=PH` the analysis program `ana.out` uses this symmetry to provide an improved estimator.

The stochastic MaxEnt requires a sum rule, and hence the kernel and image have to be adequately redefined. Let us consider $\coth(\beta\omega/2)\chi''(q,\omega)$. For this quantity, we have the sum rule, since

$$\int d\omega \coth(\beta\omega/2)\chi''(q,\omega) = 2\pi\langle\hat{S}(q,\tau=0)\hat{S}(-q,0)\rangle, \tag{255}$$

which is just the first point in the data. Therefore,

$$\langle\hat{S}(q,\tau)\hat{S}(-q,0)\rangle = \int d\omega \underbrace{\frac{1}{\pi}\frac{e^{-\tau\omega}}{1-e^{-\beta\omega}}\tanh(\beta\omega/2)}_{K_{pp}(\tau,\omega)}\underbrace{\coth(\beta\omega/2)\chi''(q,\omega)}_{A(\omega)} \tag{256}$$

and one computes $A(\omega)$. Note that since $\chi''$ is an odd function of $\omega$ one restricts the integration range to positive values of $\omega$. Hence:

$$\langle\hat{S}(q,\tau)\hat{S}(-q,0)\rangle = \int_0^\infty d\omega \underbrace{(K(\tau,\omega)+K(\tau,-\omega))}_{K_{ph}(\tau,\omega)}A(\omega). \tag{257}$$

In the code, $\omega_{\text{start}}$ is set to zero by default and the kernel $K_{ph}$ is defined in the routine XKER_ph.

In general, one would like to produce the dynamical structure factor that gives the susceptibility according to

$$S(q,\omega) = \chi''(q,\omega)/\left(1-e^{-\beta\omega}\right). \tag{258}$$

In the code, the routine BACK_TRANS_ph transforms the image $A$ to the desired quantity:

$$S(q,\omega) = \frac{A(\omega)}{1+e^{-\beta\omega}}. \tag{259}$$

**Matsubara-frequency formulation**

The ALF library uses imaginary time. It is, however, possible to formulate the MaxEnt in Matsubara frequencies. Consider:

$$\chi(q,i\Omega_m) = \int_0^\beta d\tau e^{i\Omega_m\tau}\langle\hat{S}(q,\tau)\hat{S}(-q,0)\rangle = \frac{1}{\pi}\int d\omega\frac{\chi''(q,\omega)}{\omega-i\Omega_m}. \tag{260}$$

Using the fact that $\chi''(q,\omega) = -\chi''(-q,-\omega) = -\chi''(q,-\omega)$ one obtains

$$\begin{aligned}
\chi(q,i\Omega_m) &= \frac{1}{\pi}\int_0^\infty d\omega\left(\frac{1}{\omega-i\Omega_m}-\frac{1}{-\omega-i\Omega_m}\right)\chi''(q,\omega)\\
&= \frac{2}{\pi}\int_0^\infty d\omega\frac{\omega^2}{\omega^2+\Omega_m^2}\frac{\chi''(q,\omega)}{\omega}\\
&\equiv \int_0^\infty d\omega K(\omega,i\Omega_m)A(q,\omega),
\end{aligned} \tag{261}$$

with

$$K(\omega,i\Omega_m) = \frac{\omega^2}{\omega^2+\Omega_m^2} \quad \text{and} \quad A(q,\omega) = \frac{2}{\pi}\frac{\chi''(q,\omega)}{\omega}. \tag{262}$$

The above definitions produce an image that satisfies the sum rule:

$$\int_0^\infty d\omega A(q,\omega) = \frac{1}{\pi}\int_{-\infty}^\infty d\omega\frac{\chi''(q,\omega)}{\omega} \equiv \chi(q,i\Omega_m=0). \tag{263}$$

## 10.4    Particle-Particle quantities: `Channel=PP`

Similarly to the particle-hole channel, the particle-particle channel is also a bosonic correlation function. Here, however, we do not assume that the imaginary time data is symmetric around the $\tau = \beta/2$ point. We use the kernel $K_{pp}$ defined in Eq. (256) and consider the whole frequency range. The back transformation yields

$$\frac{\chi''(\omega)}{\omega} = \frac{\tanh(\beta\omega/2)}{\omega} A(\omega). \tag{264}$$

## 10.5    Zero-temperature, projective code: `Channel=T0`

In the zero temperature limit, the spectral function associated to an operator $\hat{O}$ reads:

$$A_o(\omega) = \pi \sum_n |\langle n|\hat{O}|0\rangle|^2 \delta(E_n - E_0 - \omega), \tag{265}$$

such that

$$\langle 0|\hat{O}^\dagger(\tau)\hat{O}(0)|0\rangle = \int \mathrm{d}\omega K_0(\tau,\omega)A_0(\omega), \tag{266}$$

with

$$K_0(\tau,\omega) = \frac{1}{\pi}e^{-\tau\omega}. \tag{267}$$

The zeroth moment of the spectral function reads

$$\int \mathrm{d}\omega A_o(\omega) = \pi\langle 0|\hat{O}^\dagger(0)\hat{O}(0)|0\rangle, \tag{268}$$

and hence corresponds to the first data point.

In the zero-temperature limit one does not distinguish between particle, particle-hole, or particle-particle channels. Using the `Max_Sac.F90` with `Channel="T0"` loads the above kernel in the MaxEnt library. In this case the back transformation is set to unity. The code will also cut-off the tail of the imaginary time correlation function if the relative error is greater that the variable `Tolerance`.

## 10.6    Dynamics of the one-dimensional half-filled Hubbard model

To conclude this section, we show the example of the one-dimensional Hubbard model, which is known to show spin-charge separation (see Ref. [150] and references therein). The data of Fig. 9 was produced with the pyALF python script `Hubbard_1D.py`, and the spectral function plots with the bash script `Spectral.sh`.

# 11    Conclusions and Future Directions

In its present form, the auxiliary-field QMC code of the ALF project allows us to simulate a large class of non-trivial models, both efficiently and at minimal programming cost. The package contains many advanced functionalities, including a projective formulation, various updating schemes, better control of Trotter errors, predefined structures that facilitate reuse, a large class of models, continuous fields and, finally, stochastic analytical continuation code.

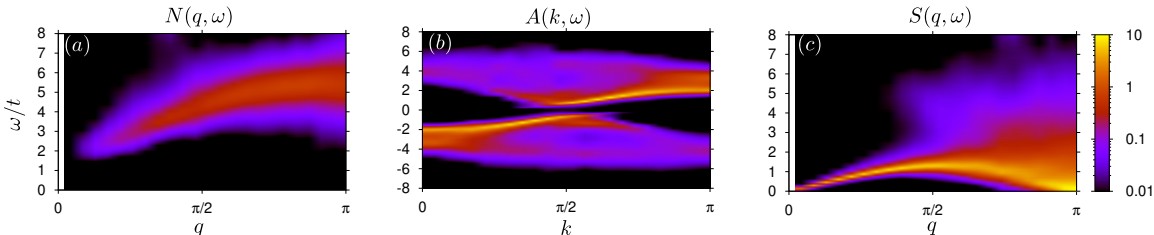

Figure 9: Dynamics of the one-dimensional half-filled Hubbard model on a 46-site chain, with U/t=4 and $\beta t = 10$. (a) Dynamical charge structure factor, (b) single particle spectral function and (c) dynamical spin structure factor. Data obtained using the pyALF python script `Hubbard_1D.py`, considering 400 bins of 200 sweeps each and taking into account the covariance matrix for the MaxEnt. The parameters for the MaxEnt that differ from the default values are also listed in the python script.

Also the usability of the code has been continuously improved. In particular the pyALF project provides a Python interface to the ALF which substantially facilitates running the code for established models. This ease of use renders ALF a powerful tool to for benchmarking new algorithms.

There are further capabilities that we would like to see in future versions of ALF. Introducing time-dependent Hamiltonians, for instance, will require some rethinking, but will allow, for example, to access entanglement properties of interacting fermionic systems [61–63]. Moreover, the auxiliary field approach is not the only method to simulate fermionic systems. It would be desirable to include additional lattice fermion algorithms such as the CT-INT [89, 151]. Lastly, at the more technical level, improved IO (e.g., HDF5 support), post-processing, object oriented programming, as well as increased compatibility with other software projects are all certainly improvements to look forward to.

## Acknowledgments

We are very grateful to B. Danu, S. Beyl, M. Hohenadler, M. Raczkowski, T. Sato, M. Ulybyshev, Z. Wang, and M. Weber for their constant support during the development of this project. We equally thank G. Hager, M. Wittmann, and G. Wellein for useful discussions and overall support. FFA would also like to thank T. Lang and Z. Y. Meng for developments of the auxiliary field code as well as to T. Grover. MB, FFA and FG thank the Bavarian Competence Network for Technical and Scientific High Performance Computing (KONWIHR) for financial support. FG, JH, and JS thank the SFB-1170 for financial support under projects Z03 and C01. F.P.T is funded by the Deutsche Forschungsgemeinschaft (DFG, German Research Foundation) – project number 414456783. JSEP thanks the DFG for financial support under the project AS120/14-1, dedicated to the further development of the ALF library. Part of the optimization of the code was carried out during the Porting and Tuning Workshop 2016 offered by the Forschungszentrum Jülich. Calculations performed to extensively test this package were carried out both on SuperMUC-NG at the Leibniz Supercomputing Centre and on JURECA [152] at the Jülich Supercomputing Centre. We thank both institutions for the generous allocation of computing time.

# A    Practical implementation of Wick decomposition of $2n$-point correlation functions of two imaginary times

In this Appendix we briefly outline how to compute $2n$ point correlation functions of the form:

$$
\lim_{\epsilon \to 0} \sum_{\sigma_1, \sigma'_1, \cdots, \sigma_n, \sigma'_n, s_1, s'_1 \cdots s_n, s'_n} f(\sigma_1, \sigma'_1, \cdots, \sigma_n, \sigma'_n, s_1, s'_1 \cdots s_n, s'_n)
$$

$$
\langle\langle \mathcal{T} \left( \hat{c}^\dagger_{x_1, \sigma_1, s_1}(\tau_{1,\epsilon}) \hat{c}_{x'_1, \sigma'_1, s'_1}(\tau'_{1,\epsilon}) - a_1 \right) \cdots \left( \hat{c}^\dagger_{x_n, \sigma_n, s_n}(\tau_{n,\epsilon}) \hat{c}_{x'_n, \sigma'_n, s'_m}(\tau'_{n,\epsilon}) - a_n \right) \rangle\rangle_C. \quad (269)
$$

Here, $\sigma$ is a color index and $s$ a flavor index such that

$$
\langle\langle \mathcal{T} \hat{c}^\dagger_{x,\sigma,s}(\tau) \hat{c}_{x',\sigma',s'}(\tau') \rangle\rangle_C = \langle\langle \mathcal{T} \hat{c}^\dagger_{x,s}(\tau) \hat{c}_{x',s}(\tau') \rangle\rangle_C \, \delta_{s,s'} \delta_{\sigma,\sigma'}. \quad (270)
$$

That is, the single-particle Green function is diagonal in the flavor index and color independent. To define the time ordering we will assume that all times differ but that $\lim_{\epsilon \to 0} \tau_{n,\epsilon}$ as well as $\lim_{\epsilon \to 0} \tau'_{n,\epsilon}$ take the values 0 or $\tau$. Let

$$
G_s(I, J) = \lim_{\epsilon \to 0} \langle\langle \mathcal{T} c^\dagger_{x_I, s}(\tau_{I,\epsilon}) c_{x'_J, s}(\tau'_{J,\epsilon}) \rangle\rangle_C. \quad (271)
$$

The $G_s(I, J)$ are uniquely defined by the time-displaced correlation functions that enter the `ObserT` routine in the Hamiltonian files. They are defined in Eq. (139) and read:

$$
\begin{aligned}
\texttt{GTO(x,y,s)} &= \langle\langle \hat{c}_{x,s}(\tau) \hat{c}^\dagger_{y,s}(0) \rangle\rangle_C = \langle\langle \mathcal{T} \hat{c}_{x,s}(\tau) \hat{c}^\dagger_{y,s}(0) \rangle\rangle_C \\
\texttt{GOT(x,y,s)} &= -\langle\langle \hat{c}^\dagger_{y,s}(\tau) \hat{c}_{x,s}(0) \rangle\rangle_C = \langle\langle \mathcal{T} \hat{c}_{x,s}(0) \hat{c}^\dagger_{y,s}(\tau) \rangle\rangle_C \\
\texttt{GOO(x,y,s)} &= \langle\langle \hat{c}_{x,s}(0) \hat{c}^\dagger_{y,s}(0) \rangle\rangle_C \\
\texttt{GTT(x,y,s)} &= \langle\langle \hat{c}_{x,s}(\tau) \hat{c}^\dagger_{y,s}(\tau) \rangle\rangle_C.
\end{aligned} \quad (272)
$$

For instance, let $\tau_{I,\epsilon} > \tau'_{J,\epsilon}$ and $\lim_{\epsilon \to 0} \tau_{I,\epsilon} = \lim_{\epsilon \to 0} \tau'_{J,\epsilon} = \tau$. Then

$$
G_s(I, J) = \langle\langle c^\dagger_{x_I, s}(\tau) c_{x'_J, s}(\tau) \rangle\rangle_C = \delta_{x_I, x'_J} - GTT(x'_J, x_I, s). \quad (273)
$$

Using the formulation of Wick's theorem of Eq. (23), Eq. (269) reads:

$$
\sum_{\sigma_1, \sigma'_1, \cdots, \sigma_n, \sigma'_n, s_1, s'_1 \cdots s_n, s'_n} f(\sigma_1, \sigma'_1, \cdots, \sigma_n, \sigma'_n, s_1, s'_1 \cdots s_n, s'_n) \quad (274)
$$

$$
\det \begin{bmatrix}
G_{s_1}(1,1)\delta_{s_1,s'_1}\delta_{\sigma_1,\sigma'_1} - \alpha_1 & G_{s_1}(1,2)\delta_{s_1,s'_2}\delta_{\sigma_1,\sigma'_2} & \cdots & G_{s_1}(1,n)\delta_{s_1,s'_n}\delta_{\sigma_1,\sigma'_n} \\
G_{s_2}(2,1)\delta_{s_2,s'_1}\delta_{\sigma_2,\sigma'_1} & G_{s_2}(2,2)\delta_{s_2,s'_2}\delta_{\sigma_2,\sigma'_2} - \alpha_2 & \cdots & G_{s_2}(2,n)\delta_{s_2,s'_n}\delta_{\sigma_2,\sigma'_n} \\
\vdots & \vdots & \ddots & \vdots \\
G_{s_n}(n,1)\delta_{s_n,s'_1}\delta_{\sigma_n,\sigma'_1} & G_{s_n}(n,2)\delta_{s_n,s'_2}\delta_{\sigma_n,\sigma'_2} & \cdots & G_{s_n}(n,n)\delta_{s_n,s'_n}\delta_{\sigma_n,\sigma'_n} - \alpha_n
\end{bmatrix}.
$$

The symbolic evaluation of the determinant as well as the sum over the color and flavor indices can be carried out with Mathematica. This produces a long expression in terms of the functions $G(I, J, s)$ that can then be included in the code. The Mathematica notebooks that we use can be found in the directory `Mathematica` of the ALF directory. As an open source alternative to Mathematica, the user can consider the Sympy Python library.

# B  Performance, memory requirements and parallelization

As mentioned in the introduction, the auxiliary field QMC algorithm scales linearly in inverse temperature $\beta$ and as a cube in the volume $N_{\mathrm{dim}}$. Using fast updates, a single spin flip requires $(N_{\mathrm{dim}})^2$ operations to update the Green function upon acceptance. As there are $L_{\mathrm{Trotter}} \times N_{\mathrm{dim}}$ spins to be visited, the total computational cost for one sweep is of the order of $\beta(N_{\mathrm{dim}})^3$. This operation alongside QR-decompositions required for stabilization dominates the performance, see Fig. 10. A profiling analysis of our code shows that 80-90% of the CPU time is spend in ZGEMM calls of the BLAS library provided in the MKL package by Intel. Consequently, the single-core performance is next to optimal.

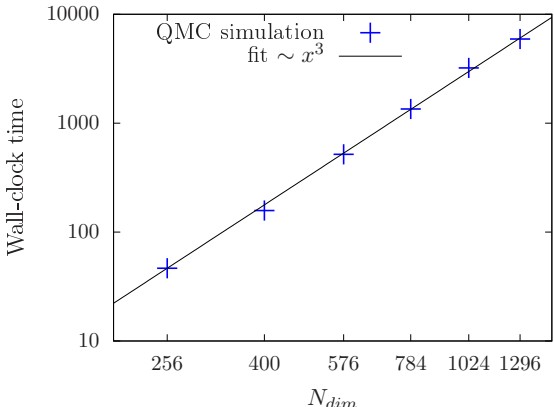

Figure 10: Volume scaling behavior of the auxiliary field QMC code of the ALF project on SuperMUC (phase 2/Haswell nodes) at the LRZ in Munich. The number of sites $N_{\mathrm{dim}}$ corresponds to the system volume. The plot confirms that the leading scaling order is due to matrix multiplications such that the runtime is dominated by calls to ZGEMM.

For the implementation which scales linearly in $\beta$, one has to store $2 \times N_{fl} \times L_{\mathrm{Trotter}}/\texttt{NWrap}$ intermediate propagation matrices of dimension $N_{\mathrm{dim}} \times N_{\mathrm{dim}}$. Hence the memory cost scales as $\beta N_{\mathrm{dim}}^2$ and for large lattices and/or low temperatures this dominates the total memory requirements that can exceed 2 GB memory for a sequential version.

The above estimates of $\beta N_{\mathrm{dim}}^3$ for CPU time and $\beta N_{\mathrm{dim}}^2$ for memory implicitly assume Hamiltonians where the interaction is a sum of local terms. Recently Landau level projection schemes for the regularization of continuum field theories have been introduced in the realm of the auxiliary field QMC algorithm [70,71]. In this case the interaction is not local, such that the matrices stored in the `Op_V` array of `Observable` type are of dimension of $N_{\mathrm{dim}}$. Since the dimension of the `Op_V` array scales as $N_{\mathrm{dim}}$, the memory requirement scales as $N_{\mathrm{dim}}^3$. In these algorithms, a single field couples to a $N_{\mathrm{dim}} \times N_{\mathrm{dim}}$ matrix, such that updating it scales as $N_{\mathrm{dim}}^3$. Furthermore, and as mentioned in Sec. 2.3, for non-local Hamiltonians the Trotter time step has to be scaled as $1/N_{\mathrm{dim}}$ so as to maintain a constant systematic error. Taking all of this into account, yields a CPU time that scales as $\beta N_{\mathrm{dim}}^5$. Hence this approach is expensive both in memory and CPU time.

At the heart of Monte Carlo schemes lies a random walk through the given configuration space. This is easily parallelized via MPI by associating one random walker to each MPI task.

For each task, we start from a random configuration and have to invest the autocorrelation time $T_{\text{auto}}$ to produce an equilibrated configuration. Additionally we can also profit from an OpenMP parallelized version of the BLAS/LAPACK library for an additional speedup, which also effects equilibration overhead $N_{\text{MPI}} \times T_{\text{auto}}/N_{\text{OMP}}$, where $N_{\text{MPI}}$ is the number of cores and $N_{\text{OMP}}$ the number of OpenMP threads. For a given number of independent measurements $N_{\text{meas}}$, we therefore need a wall-clock time given by

$$T = \frac{T_{\text{auto}}}{N_{\text{OMP}}} \left( 1 + \frac{N_{\text{meas}}}{N_{\text{MPI}}} \right) . \tag{275}$$

As we typically have $N_{\text{meas}}/N_{\text{MPI}} \gg 1$, the speedup is expected to be almost perfect, in accordance with the performance test results for the auxiliary field QMC code on SuperMUC (see Fig. 11 (left)).

For many problem sizes, 2 GB memory per MPI task (random walker) suffices such that we typically start as many MPI tasks as there are physical cores per node. Due to the large amount of CPU time spent in MKL routines, we do not profit from the hyper-threading option. For large systems, the memory requirement increases and this is tackled by increasing the amount of OpenMP threads to decrease the stress on the memory system and to simultaneously reduce the equilibration overhead (see Fig. 11 (right)). For the displayed speedup, it was crucial to pin the MPI tasks as well as the OpenMP threads in a pattern which keeps the threads as compact as possible to profit from a shared cache. This also explains the drop in efficiency from 14 to 28 threads where the OpenMP threads are spread over both sockets.

We store the field configurations of the random walker as checkpoints, such that a long simulation can be easily split into several short simulations. This procedure allows us to take advantage of chained jobs using the dependency chains provided by the batch system.

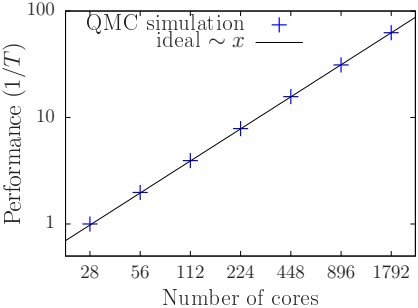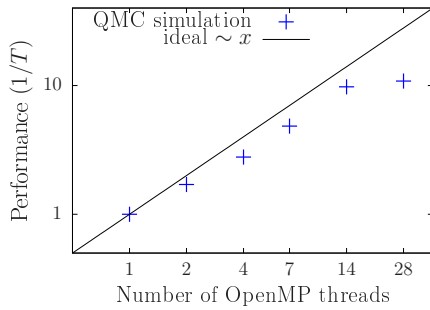

Figure 11: MPI (left) and OpenMP (right) scaling behavior of the auxiliary field QMC code of the ALF project on SuperMUC (phase 2/Haswell nodes) at the LRZ in Munich. The MPI performance data was normalized to 28 cores and was obtained using a problem size of $N_{\mathrm{dim}} = 400$. This is a medium to small system size that is the least favorable in terms of MPI synchronization effects. The OpenMP performance data was obtained using a problem size of $N_{\mathrm{dim}} = 1296$. Employing 2 and 4 OpenMP threads introduces some synchronization/management overhead such that the per-core performance is slightly reduced, compared to the single thread efficiency. Further increasing the amount of threads to 7 and 14 keeps the efficiency constant. The drop in performance of the 28 thread configuration is due to the architecture as the threads are now spread over both sockets of the node. To obtain the above results, it was crucial to pin the processes in a fashion that keeps the OpenMP threads as compact as possible.

## C    Licenses and Copyrights

The ALF code is provided as an open source software such that it is available to all and we hope that it will be useful. If you benefit from this code we ask that you acknowledge the ALF collaboration as mentioned on our website https://alf.physik.uni-wuerzburg.de. The git repository at https://git.physik.uni-wuerzburg.de/ALF/ALF gives us the tools to create a small but vibrant community around the code and provides a suitable entry point for future contributors and future developments. The website is also the place where the original source files can be found. Its public release make it necessary to add copyright headers to our source code, which is licensed under a GPL license to keep the source as well as any future work in the community. And the Creative Commons licenses are a good way to share our documentation and it is also well accepted by publishers. Therefore this document is licensed to you under a CC-BY-SA license. This means you can share it and redistribute it as long as you cite the original source and license your changes under the same license. The details are in the file `license.CCBYSA`, which you should have received with this documentation. To express our desire for a proper attribution we decided to make this a visible part of the license. To that end we have exercised the rights of section 7 of GPL version 3 and have amended the license terms with an additional paragraph that expresses our wish that if an author has benefited from this code that he/she should consider giving back a citation as specified on alf.physik.uni-wuerzburg.de. This is not something that is meant to restrict your freedom of use, but something that we strongly expect to be good scientific conduct. The original GPL license can be found in the file `license.GPL` and the additional terms can

be found in `license.additional`. In favour to our users, the ALF code contains part of the Lapack implementation version 3.6.1 from http://www.netlib.org/lapack. Lapack is licensed under the modified BSD license whose full text can be found in `license.BSD`.

With that being said, we hope that the ALF code will prove to you to be a suitable and high-performance tool that enables you to perform quantum Monte Carlo studies of solid state models of unprecedented complexity.

The ALF project's contributors.

## COPYRIGHT

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
