# Peer review of "The ALF (Algorithms for Lattice Fermions) project release 2.4. Documentation for the auxiliary-field quantum Monte Carlo code"

_SciPost Physics Codebases, doi:SciPost Phys. Codebases 1-r2.4 (2025) , SciPost Phys. Codebases 1-v2.4 (2025)_

## Round 2 · Referee Report · Anonymous (Referee 1) · 2021-5-15

Strengths

This manuscript presents a documentation for Auxiliary Field Monte Carlo simulation codes of many-fermion problems developed by the authors. This is one of the world's leading research groups for doing these types of simulations. These are state of the art methods and have the capability of addressing many contemporary problems in many-body physics.

Making these well documented codes public would greatly enhance their utility and be a great service to the condensed matter community. In addition to worked examples and tutorials the document also presents pedagogical introduction to the Quantum Monte Carlo methods and their variations.

Weaknesses

None

Report

This manuscript is well suited to this journal. I would recommend publication of the manuscript without further changes.

---

## Round 7 · Referee Report · Anonymous (Referee 1) · 2023-8-23

Strengths

1- Very thorough manuscript 2-Highly impactful project

Weaknesses

None found by reviewer

Report

This manuscripts presents the documentation for the ALF (Algorithms for Lattice Fermions) project, release 2.4. The work presented here is comprehensive and leading edge in the field.

A previous version 2.0 has already been published through SciPost in August 2022. The reviewer is not aware of SciPost policy on different release on the same project, however significant progress seem has been made through the releases as reflected in the manuscript as well as the changelog (https://git.physik.uni-wuerzburg.de/ALF/ALF/-/blob/ALF-2.4/CHANGELOG.md).

If the journal policy permits, the most updated documentation is worth publishing, especially for such a quickly-developing field.

---

## Round 7 · Referee Report · Anonymous (Referee 2) · 2023-8-25

Strengths

See report

Weaknesses

See report

Report

The ALF library provides very welcome state-of-the-art code of great value for the community, with many successful usecases already documented from the previous release. The authors now provide a substantially updated code, along with an extended documentation that continues to give a well-balanced description of both code features and corresponding descriptions of the relevant algorithms, including also the relevant original literature.

The only major point I noticed is that it was pointed out recently (in arXiv:2303.14326) that previous implementations of the Grover-based algorithm for the calculations of entanglement entropies apparently did not use the full weight for the important sampling, leading to systematic deviations. In case the ALF code is based on the incomplete weight, the authors should consider to modify the code. If instead the ALF code is indeed using the full weight, this fact should be stressed in the documentation. In either case, it would be appropriate to cite the above reference.

A few minor points:
-there are a few minor typos (e.g., „bellow“ on page 21), which point towards a further spellcheck.
-the figure placements could be reconsidered, e.g., Fig. 2 comes a bit late, being cited already on page 23 for the first time.

Requested changes

See report

---

## Round 8 · Author Response

We would like to thank the referee for their comment concerning the implementation of entanglement entropy.

The method we use is correct and follows precisely the seminal work of T. Grover. Although there are no conceptual errors, it is known that the method proposed by T. Grover suffers from fat tails and spikes in the measurements in the strong coupling regime. In the revised version of the manuscript, we comment on this aspect of the Grover approach for the calculation of the Renyi entropies. It is however a very useful approach for small subsystems in the weak to intermediate coupling regime.

In the meantime, there has been major progress in the calculation of the Renyi entropy. In particular a fermion generalization of the incremental approach has been proposed. In the revised version of the manuscript, we clearly mention the limitations of the implemented calculation of the Renyi entropies. It is beyond the scope of this version to implement the incremental method.

We would also like to thank the referee for pointing out typos in the manuscript.

---

## Round 8 · List of Changes

1. Below Eq. 193. We have added a word of caution for the use of the Grover method to compute Renyi entropies.
  2. Again below Eq. 193. We have added Refs. 146 and 147. Both references document recent progress in the calculation of the Renyi entropy.
  3. We have attempted to catch all the typos.

---

## Editorial Decision

published